

# AutoTerm: A "big data" repository of Greenland glacier termini delineated using deep learning

Enze Zhang[1], Ginny Catania[1,2], and Daniel T. Trugman[3]

[1]The University of Texas at Austin, Institute of Geophysics, TX 78758, USA
[2]The University of Texas at Austin, Department of Geological Sciences, TX 78712, USA
[3]University of Reno, Nevada, Nevada Seismological Laboratory, NV 89557, USA

**Correspondence:** Enze Zhang (enze.zhang@austin.utexas.edu)

**Abstract.** Ice sheet marine margins via outlet glaciers are susceptible to climate change and are expected to respond through retreat, steepening, and acceleration, although with significant spatial heterogeneity. However, research on ice-ocean interactions has continued to rely on decentralized, manual mapping of features at the ice-ocean interface, impeding progress in understanding the response of glaciers and ice sheets to climate change. The proliferation of remote sensing images lays the

foundation for a better understanding of ice-ocean interactions and also necessitates the automation of terminus delineation. While deep learning (DL) techniques have already been applied to automate the terminus delineation, none involve sufficient quality control and automation to enable DL applications to "Big Data" problems in glaciology. Here, we build on established methods to create a fully automated pipeline for terminus delineation that makes several advances over prior studies. First, we leverage existing manually-picked terminus traces (16,440) as training data to significantly improve the generalization of the DL algorithm. Second, we employ a rigorous automated screening module to enhance the data product quality. Third,

we perform a thoroughly automated uncertainty quantification on the resulting data. Finally, we automate several steps in the pipeline allowing data to be regularly delivered to public databases with increased frequency. The automation level of our method ensures the sustainability of terminus data production. Altogether, these improvements produce the most complete and high-quality record of terminus data that exists for the Greenland Ice Sheet (GrIS). Our pipeline has successfully picked

278,239 termini for 295 glaciers in Greenland from Landsat-5, -7, -8, Sentinel-1, and -2 images, spanning from 1984 to 2021 with an average uncertainty of 37 meters. The high sampling frequency and the controlled quality of our terminus data will enable better quantification of ice sheet change and model-based parameterizations of ice-ocean interactions.

## 1   Introduction

The declining mass balance of the world's ice sheets and glaciers represents the largest source of sea level rise occurring since

the 1900s, with losses from mountain glaciers, the Greenland Ice Sheet (GrIS), and the Antarctic Ice Sheet (AIS) representing 41%, 25%, and 4% of total sea level rise respectively (IPCC, 2021). This loss of ice is driven by climate-induced changes in the surface mass balance, which primarily impacts snowfall accumulation and surface melt, and so-called dynamic changes in ice flow that occur as a result of changing ice flux to the ocean. Current work for the two largest ice sheets on Earth suggests that much of the past ice loss was dominated by the enhanced flow of ice as revealed in satellite-derived ice surface





velocities (Mouginot et al., 2019; Rignot et al., 2019). Recent results suggest that outlet glacier dynamics will continue to contribute 50±20% of the total mass loss of the ice sheet through to the end of the century (Choi et al., 2021). While a range of mechanisms can lead to enhanced flow, there is a general consensus that ocean-induced terminus retreat is one of the dominant triggers for this enhanced flow (Catania et al., 2018; King et al., 2020; Hill et al., 2018; Murray et al., 2015; Miles et al., 2016; Cook et al., 2016; Seroussi et al., 2017-06; Miles et al., 2013). Shrinking ocean-terminating glaciers will

not only impact sea level; increased freshwater discharge (via meltwater and icebergs) into the climate-sensitive, convective polar regions also plays a role in global ocean circulation (Böning et al., 2016; Luo et al., 2016; Oltmanns et al., 2018; Pan et al., 2022). Regionally, increasing freshwater discharge and the distribution and transport of sediments and nutrients into the ocean also influences the marine ecosystem (Arrigo et al., 2017; Bhatia et al., 2013; Arendt et al., 2016; Overeem et al., 2017). Further, terminus-derived icebergs have been shown to significantly contribute to fjord circulation, impacting the magnitude,

timing, and spatial distribution of submarine melt at the terminus, which is itself a trigger of glacier retreat (Moon et al., 2018) Thus, understanding and correctly representing changes at the ice sheet marine margin is key to predicting future polar ocean variability and the fate of dependent systems.

In Greenland, the magnitude and timing of terminus-driven dynamic mass loss varies widely between glaciers in part due to differences in glacier geometry (Enderlin et al., 2013; Brough et al., 2019; Bunce et al., 2018; Catania et al., 2018; Felikson

et al., 2017; Bassis and Jacobs, 2013). In addition, regional variability in climate forcing also influences the response of marine-terminating glaciers, as supported by several observation- and modeling-based studies (Holland et al., 2008; Rignot et al., 2016; Straneo and Heimbach, 2013; Cook et al., 2016; Miles et al., 2016; Wood et al., 2021). At present, the research community lacks agreement regarding how to parameterize terminus behavior. This is partly because myriad processes can occur at the ice-ocean boundary, but these processes vary over space and time, both within an individual glacier fjord but also

between glaciers. The research community also suffers from irregular availability and uneven distribution of terminus data, and data that do exist are inconsistent in format, quality, sampling frequency, and availability. This makes it more difficult for terminus data to be used in models (e.g., numerical or machine learning) to test various terminus parameterizations. Together, these factors contribute to an inability to quantify the relationship governing interactions between external and internal controls on glacier termini, which leads to large ranges in published sea-level rise projections over the coming century. For example,

numerical modeling studies project between 5 – 33 cm of sea-level rise contribution from the GrIS by 2100 with discharge from outlet glaciers accounting for 8 – 45% of the total (Aschwanden et al., 2019).

Over the last few decades, the proliferation of new satellite sensors has created an explosion of Earth science data for use by scientists. The sheer volume of data, when coupled with increasing computational capacity and the rapid improvement of deep learning (DL) algorithms, allows scientists to construct exceptional spatio-temporal time series of the changing Earth. This

is particularly valuable for the Earth's cryosphere, which exhibits large, non-linear sensitivity to climate change. Recently, several studies have demonstrated that it is possible to use DL methods to delineate glacier termini (Mohajerani et al., 2019; Zhang et al., 2019; Baumhoer et al., 2019; Cheng et al., 2020; Zhang et al., 2021; Davari et al., 2021; Hartmann et al., 2021; Holzmann et al., 2021; Marochov et al., 2021; Davari et al., 2021; Heidler et al., 2021; Periyasamy et al., 2022) with many generating data products that are of interest to the glaciological community.





While these works represent a significant step forward, making DL algorithms applicable to the total catalog of image data necessitates a level of generalization, rigor, and automation that has not yet been accomplished due to several outstanding challenges. First, applying deep learning to the existing and substantial volume of images requires the network to have a high level of generalization, comparable to the diversity found in all of the images. This diversity is introduced by spatial and temporal coverage of the images and the difference in satellite sensors. Most previous studies applied DL algorithms to thousands of images, with the most complete study generating 22,678 glacier termini (Cheng et al., 2020). However, the number is an order of magnitude less than the number of the total catalog of image data (more than 400,000 in Greenland). The complexity brought by such a large and diverse set of images could fail with existing algorithms. Therefore, generalization of the DL algorithm must be improved before applying it to the total catalog. Secondly, despite its power, DL technology cannot perfectly identify termini for all available images. Most previous studies have no quality control of the automatically picked terminus traces, which can lead to spurious terminus trace results. Only two studies (Zhang et al., 2021; Baumhoer et al., 2019) developed automated quality control techniques, but they have limited applications and are thus insufficient to be applied to the large volume of glaciers. Thirdly, any manual step in the pipeline requires intense effort and significantly slows progress, considering the substantial processing load. This necessitates improved automation in the pipeline that spans from data collection to quality control and quantifying data uncertainties, which previous studies have lacked.

Here, we build on established methods to implement an automated pipeline for terminus delineation that makes several advances over prior studies. First, we leverage existing manually-picked terminus data (Goliber et al., 2022) to use as our training data, which greatly improves generalization of the DL algorithm. Second, we employ a rigorous automated screening module improving on previous methods (Zhang et al., 2021) to refine data quality. Third, we perform a thorough uncertainty quantification on our resulting data in order to provide end-users with quantified estimates of data quality. Finally, we automate multiple steps in the pipeline allowing data (Glacier IDs, termini, and ice/ocean/bedrock masks) to be regularly delivered to public databases with increased and regular frequency. Altogether, these improvements produce the most complete and high-quality record of terminus data that exists for the GrIS, and one that can be updated as new imagery become available.

## 2 Input Data

### 2.1 Remote Sensing Imagery

Our data cover five satellites available on Google Earth Engine (GEE; described in detail below); Landsat-5 to 8 and Sentinel-1 and 2, with a diverse range of image resolutions, repeat cycles, and operation times (Table 1). Note that GEE only contains Landsat-7 images until 2013 over Greenland, although the satellite is still operating as of 2022. As the only satellite operating in winter, Sentinel-1 is essential for analyzing seasonal terminus variations. However, despite the success of Sentinel-1 instruments and their ground processing system in providing open-source data with high geometric accuracy, Sentinel-1 images have several issues. First, apparent georeferencing errors remain between Sentinel-1 and optical images (Ye et al., 2021) thus requiring a georeferencing adjustment for Sentinel-1 that must be automatically applied. Second, the distribution of Sentinel-1



images is not even across Greenland with some glaciers located in between image gaps. Third, as SAR images, Sentinel-1 images are cloud-free but suffer from speckle noise (Bamler, 2000), affecting the image quality.

## 2.2 TermPicks: Manually-digitized terminus dataset

Deep learning methods employ training data to be used to train the algorithm to predict termini in new imagery. Here, we use a manually-picked terminus dataset for Greenland called TermPicks (Goliber et al., 2022), which covers 291 outlet glaciers in Greenland with over 39,000 terminus traces spanning from 1916 to 2019. As the most complete set of manually digitized terminus data for Greenland's outlet glaciers, TermPicks enriches the training set and improves generalization of the network. TermPicks merges several existing Glacier ID files across both published literature and several unpublished sources to properly

identify glaciers and homogenize terminus trace data. TermPicks data have been cleaned to ensure quality, and reformatted specifically for deep learning applications. This dataset covers a wide range of local conditions (e.g., weather, illumination angle, ice mélange strength), glacier orientations, geometries, and satellite sensor differences (e.g., different image textures and pixel value ranges).

## 2.3 Glacier Identifications

Glacier identification is important for data management since Greenland has numerous glaciers. Here, we include 295 outlet glaciers by combining IDs from (Moon and Joughin, 2008) and TermPicks (Goliber et al., 2022). The criteria of these two IDs are as follows: glaciers with velocities larger than $50 \ m \ yr^{-1}$, grounding lines below sea level (ocean terminating), and termini greater than or equal to 1km in width (Moon and Joughin, 2008; Goliber et al., 2022). To be easily referenced with other datasets, we also include glacier naming schemes cataloged by Bjørk et al. (2015) along with the IDs. Glacier IDs need

to be updated continuously because as glaciers retreat, the terminus may diverge into several tributaries and vice versa, as a glacier advances several tributaries merge into one. Since the two ID files are based on the more recent configuration of GrIS outlet glaciers, some glacier termini do not appear in older Landsat-5 and -7 images because at that time they had merged with adjacent tributaries. Thus, we do not include these glaciers in those older images. Although a static Glacier ID is sufficient for current usage, updating the Glacier IDs is an essential step in maintaining the longevity of the pipeline in the future (see section

115 5.4).

## 2.4 Ice/Ocean Mask

Land, ice, and ocean masks serve as important data sources for estimating ice-mass balance through elevation changes. Measuring height differences without considering changes in the position of glacier termini can result in significant spurious changes that can dominate estimates of ice-mass change (Kjeldsen et al., 2020; Hansen et al., 2022). Ice masks delineate the glacier

area so that measured elevation or mass change can be integrated over the glacier domain (and not, for example, over ocean or rock). In addition, these masks are used to remove measurements over open water so that measured elevation or mass changes





never represent, for example, the difference between the height or mass of a glacier and the height or mass of the open water that replaces it when the glacier calves away during retreat.

With the newly generated terminus traces and an original mask, we can update the mask and avoid the spurious changes
caused by using fixed terminus positions. The original mask we use is the 2015 GrIMP ice mask from the Greenland Ice Mapping Project (Howat, 2017). This mask used manual delineation of the ice margin from the panchromatic and pan-sharpened multispectral GrIMP 2015 Image Mosaic (Howat, 2017). The ice mask includes snowfields, identifies the ice sheet margin using visible information, and breaks in surface slope, where it is visually difficult to differentiate the ice margin. The ocean mask is produced similarly but examines only the coastline with the null of the ice and ocean masks being ice-free terrain
(Howat et al., 2014).

## 3 Methods

Our overarching approach is to build an automated pipeline (Fig. 1) for extracting outlet-glacier termini from all available satellite remote sensing images on Google Earth Engine over Greenland using deep learning (DL). Automation requires steps above and beyond terminus delineation, including image collection, pre-processing, quality control, and uncertainty quantifi-
cation (Fig. 1, blue arrows). Additionally, converting the TermPicks terminus data into a training dataset suitable for deep learning highly generalizes the network and ensures the success of extracting glacier termini from new datasets (Fig. 1, black arrows). In the long term, additional efforts are required to maintain the pipeline, such as preparing more training data and updating the ROI (Fig. 1, red arrows). The whole pipeline is built and executed with all software written in Python and Bash.

### 3.1 Automated Data Collection and Pre-processing

As the first step, automating image collection eliminates the time involved in the manual collection of remote sensing imagery. We use Google Earth Engine (GEE) Python API (https://earthengine.google.com/) to automate our search for satellite data with a given region of interest (ROI) for each glacier and use GEE tools (https://github.com/gee-community/gee_tools) to automate data collection. The ROIs require manual preparation to span the range of terminus variations occurring during the study period for each glacier. This is the only manual step in data collection, however it only needs to be done once for each glacier, thus
represents the minimum manual effort. GEE provides a platform for scientific analysis and visualization of geospatial datasets but also hosts a large volume of satellite imagery that goes back more than forty years and stores these in a public data archive (Table 1). The images, ingested on a daily basis, are then made available for global-scale data mining. GEE also provides APIs and other tools to enable the analysis of large datasets. Through the fusion of multiple datasets on GEE, we can provide a publicly-available, densely-sampled terminus position dataset that covers the observational time period and importantly fills
gaps in existing (manual and automatically delineated) terminus data sets. We do not use a cloud filter to maximize the number of available images where termini may be visible. This is because common cloud filters are calculated based on the full image scene but not the area of interest where a termini might be located. Thus, image scenes with high cloud coverage might still



have clear views of glacier termini. Instead, we filter cloud-covered termini with a screening module described in Section 3.4. Overall, we collected ∼430,000 images with a total volume of ∼1 TB spanning the GrIS over a period from 1984-2021.

In addition to automating the data acquisition process, we also automate several data preparation steps before applying DL to delineate glacier termini. First, all satellite images are cropped to the ROI on the GEE platform to save local processing and storage costs. For example, the size of an entire Sentinel-1 scene is about 800 MB, while the size of a cropped image is less than 10 MB, meaning that cropping can decrease costs by a factor of ∼80. Second, we pre-process these cropped images on our local server to normalize image differences between sensors with heterogeneous image textures, resolutions, pixel values, etc. This

normalization is necessary since it will ease the terminus extraction task for the DL algorithm by decreasing the complexity level of the dataset, especially when applying DL to a substantial volume of images. We first use histogram normalization to equalize the pixel value differences between SAR and optical image types with different dynamic ranges and image textures (Zhang et al., 2021). We then normalize the image size to standardize image resolutions. This allows glaciers with various natural sizes to have a similar image size in computer vision, which largely decreases the complexity of delineating glacier

terminus. Specifically, we upsample small images (image width less than 1000 pixels) by an integer value so that their widths are just over 1000 pixels. We do not downsample images of large glaciers to avoid losing spatial information. Moreover, since the images will be subdivided into patches with overlaps before going through the network (Zhang et al., 2019), upsampling the small images allows the network to make multiple predictions over the same area, making the inferences more robust. The effect of size normalization will be discussed in the section 5.1.

## 3.2   Generalizing the Network


The ability of the network to generalize and identify a glacier terminus is primarily determined by the heterogeneity found in the training dataset (LeCun et al., 2015). More precisely, we want the training dataset to reflect the heterogeneity of conditions observed in the real world. To accomplish this, we leverage existing manually-picked terminus data from Greenland using TermPicks (Goliber et al., 2022), which consists of the largest compilation of manually-picked terminus traces covering a

range of satellite sensors and glacier conditions. The TermPicks traces, which are polylines, need to be converted into labeled polygons for generating binary labels (Fig. 2) for each glacier in Greenland. Each labeled polygon contains the glacier terminus, fjord boundary, and an outer boundary that ensures that the polygon covers the corresponding source image. To automate the conversion of terminus traces into polygon labels, we manually create one reference polygon for each glacier. A reference polygon is similar to a polygon label, but its terminus (blue curve in Fig. 2a) is up-glacier from all the TermPicks traces (red

curves in Fig. 2a) for that glacier. This ensures that each reference polygon has two intersections with a TermPicks trace (on either end of the TermPicks trace). We then generate polygon labels by connecting each TermPicks trace between the two intersection points and the reference polygon (e.g., red line in Fig 2b). Then, we pair the converted polygon labels with the GEE collection of satellite images based on date. Finally, we manually abandon training data mismatches between polygon labels and images. This can occur when manually-picked traces do not extend across the fjord, contain erroneous points (Fig.

S1a, b), and/or are offset due to differences in georeferencing (Fig. S1c). After manual checking, we have 16,440 polygon





labels from TermPicks for 249 glaciers. Most of the unused TermPicks traces are due to not being able to match the source image, as we only use the data available on GEE, which has a limited temporal range.

Although TermPicks covers a range of conditions and brings great diversity to the training set, additional training data would improve the accuracy of the network in difficult situations. We identify five conditions that pose distinct challenges: (1) images covered by cloud but where termini are still visible; (2) winter Sentinel-1 images with blurry boundaries due to its coarse resolution, ice mélange and snow cover; (3) images with shadow over the terminus; (4) images with tabular icebergs close to the glacier terminus; and (5) similarities in texture between ice mélange and glacier (Fig. S2). For these types of images, we manually prepare an additional 1,466 training examples. To further increase the diversity of our training set, we perform data augmentation to all the training examples, including rotating images by 90°, 180°, 270°, and image flipping following (Zhang et al., 2019), increasing the training set by a factor of four.

### 3.3 Deep Learning Network

We use DeepLabv3+, a state-of-the-art deep learning algorithm for image segmentation (Chen et al., 2018). DeepLabv3+ combines an encoder-decoder structure with atrous convolution, where the former can obtain sharp object boundaries while the latter senses multi-scale contextual information. These two abilities are helpful for our task since 1) sharp boundaries can improve delineation accuracy, and 2) we integrate remote sensing datasets with different spatial resolutions. This network has been proven to have large learning capability, spatial transferability, and the capability of using multi-sensor remote sensing images (Zhang et al., 2021). The network is trained with a learning rate of 0.005 and an L2 regularization factor of 0.0005, as recommended by (Zhang et al., 2021). To improve the efficiency of network training, we choose the largest batch size (16) and maximize our computational power. The network training takes about a week and consumes 120 GB of memory, across 4 NVIDIA A100 GPU. After the training, terminus picking and post-processing for a single image takes less than a minute.

### 3.4 Automated Screen Module

Despite the power of deep learning technology, it cannot perfectly identify termini for all available images. Moreover, the network is expected to generate erroneous results from images where termini are invisible. These results should be detected and removed. With this in mind, we have developed an automated screening module to assist with quality control. Many previous DL methods applied to terminus delineation do not have any quality control (Mohajerani et al., 2019; Zhang et al., 2019; Cheng et al., 2020). Where it does exist, data screening has been simplistic and not automatically applied. For example, Zhang et al. (2021) only considers the complexity of the terminus shape and removes traces with abnormal complexity (which, in turn, requires a threshold to be established for each glacier), and Baumhoer et al. (2019) only considers outliers that arise in a time series of terminus position change over time.

We develop an automated screening module (Fig. 3) based on quantified outliers in three different categories: 1) terminus length, 2) terminus curvature, and 3) the abnormally large area enclosed by the two temporally closest termini. This latter case refers to outliers in a time series of terminus change. Terminus length is determined by the sum of the piece-wise length along an individual terminus trace. Terminus curvature is computed between two adjacent points for each point along the terminus



and then an average is taken for each terminus trace. Finally, we calculate the area enclosed by two temporally-adjacent termini
to determine the change in glacier area over time. With these three metrics, we calculate the lower ($T_L$) and upper thresholds
($T_U$) for each based on the inter-quartile range:

$$T_L = Q1 - 1.5 \times (Q3 - Q1) \tag{1}$$

$$T_U = Q3 + 1.5 \times (Q3 - Q1) \tag{2}$$

where $Q3$ is the $75^{\text{th}}$ percentile and $Q1$ is the $25^{\text{th}}$ percentile of the data range. Since the Sentinel-1 images suffer from
speckle noise, the results generated from this satellite are of relatively poor quality compared to the other datasets. Therefore,
we calculate the thresholds based on results from Landsat-8 and Sentinel-2 and apply them uniformly to all remaining datasets.
For outliers in terminus length, we remove both the lower and upper thresholds (Eqns. 1 and 2) because we do not anticipate
large changes in terminus length in either direction (bigger or smaller). In contrast, terminus curvature and area change outliers
are only removed with the upper threshold (Eqn. 1). This is because high-quality terminus traces are expected to be smooth with
small curvature and have a time derivative of terminus change that is small at the sampling frequency permitted. Exceptions to
this latter assumption exist when large calving events occur. In that case, if all of the traces are accurate, only one anomalously
large area change will occur over a short period (typically less than a month). To remove incorrect traces and retain traces
informing of large calving events, we examine the change in the terminus area over five consecutive area polygons (in a
moving time window) and remove the first large-area polygon only if more than one large-area change is identified. We then
repeat this screening procedure ten times to maintain the quality of the terminus product (Fig. 3).

### 3.5   Georeferencing Adjustment for Sentinel-1

Location errors occur for Sentinel-1 images along the azimuth direction (Small and Schubert, 2019) introducing error in
georeferencing for this sensor in our data (Fig. S3). Although applications have been made to correct these georeferencing
errors in post-processing (Ye et al., 2021), they have not been widely deployed for public use. Owing to the overlap of multiple
sensors, it is possible to have more than one machine-predicted terminus trace for a single date allowing us to use duplicate
traces to aid in performing a georeferencing adjustment for Sentinel-1. This is done by calculating all of the areas enclosed by
Sentinel-1 traces and comparing these to area enclosed by traces on the same day, but from optical sensors. Then we take the
averaged area difference between these two time series to adjust the georeferencing offset in the retreat time series.

### 3.6   Uncertainty Quantification

Traditional uncertainty quantification for glacier terminus position is conducted by calculating the difference between manually
picked termini and automatically-picked termini (e.g., Cheng et al., 2020). However, the network accuracy likely varies over
time as glaciers experience different conditions (e.g., cloud cover). Uncertainty quantification thus requires significant manual
effort to ensure that the computed uncertainty is representative of such variability. We compute uncertainty two ways. First,





we use duplicate traces (described above) to automatically quantify the uncertainties for each glacier. For this, only the traces
with the highest source image resolution (Table 1) are kept (Sentinel-2 and Landsat-8). We do not use duplicate Sentinel-1
traces because they are used for the georeferencing offset for that sensor and we do not use Landsat-5 or -7 because of the lack
of overlap with other datasets. Uncertainty from duplicate traces is computed by comparing the average area enclosed by the
duplicate Landsat-8 traces and Sentinel-2 traces for the same date. We also divide this area by the piece-wise terminus length
to get the uncertainty in terminus position as a measure of length change. This is done because some data users may prefer to
examine terminus change in length instead of terminus change in area.

We also compute uncertainty through deploying the Monte Carlo (MC) dropout (Gal and Ghahramani, 2016) method, which
has become widely adopted in the uncertainty quantification for DL methods (Abdar et al., 2021). Dropout is a regularization
technique that prevents overfitting of the data ensuring that the model works well with new imagery that is not contained in the
training data. MC dropout yields variants of our DL network by dropping out random subsets of the network's neurons during
prediction (setting their values to zero). These variants make multiple inferences for a single remote sensing image, and the
differences between these inferences can be used to quantify the model uncertainty. One study deployed MC dropout to glacier
terminus delineation (Hartmann et al., 2021); however, instead of quantifying the uncertainties of terminus traces, they use the
multiple inferences of MC dropout as extra information to retrain the network. Here, we deploy MC dropout and use network
variants to pick multiple terminus traces for a single image. By quantifying the average difference between the traces from
the original network and the variants, we measure the uncertainties in the terminus position, providing a different perspective
on uncertainty quantification from duplicate traces. MC dropout requires multiple inferences and is computationally time-
consuming. To strike a balance between computational cost and the reliability of the MC dropout, we randomly chose ten
images from all the sensors and make three inferences for each of them. Thus, in total each glacier will have two measures of
uncertainty: one from duplicate traces and the other estimated by MC dropout.

## 3.7  Ice/Ocean Masks

The newly generated terminus traces are also used to update the GrIMP mask for accurate estimates of ice mass balance.
While we can update masks monthly, we do not expect significant changes in glacier area on this time scale. We thus only
create updated masks annually beginning in 2018 to serve the ICESat-2 community needs for improved accuracy of laser
returns during periods of extensive glacier terminus retreat. To create a new ice mask we first select terminus traces at a time
of minimum ice extent (late Fall) for every glacier. These termini are combined with geometries delineating the edges of outlet
fjords and the edges of static ice margins from GrIMP (Howat et al., 2014) to form a continuous boundary of the ice sheet.
We use the new terminus to update only the ocean mask and consider the bedrock mask to be static. The ice mask is updated
automatically because of the shared ice-ocean boundary with the ocean mask. Practically, we first vectorize the ocean mask
into a shapefile. Second, we crop the shapefile with the glacier ROIs and replace the parts in the ROIs with the newly generated
terminus traces. Then, we convert the updated shapefile to a raster as the new ocean mask. Finally, the residual of the new
ocean mask and original bed mask serve as the new ice mask.



## 4 Results

In addition to terminus delineation, we have successfully automated data collection, pre-processing, data quality control (Fig. 3), uncertainty quantification, the measurement of terminus variation (Fig. 1, blue color), and the derivation of annual

ice/ocean/land masks for Greenland. The improvements in automation enable the pipeline to generate a tremendous amount of terminus trace data continuously with controlled quality and measured uncertainties. As a result, the pipeline is capable of automatically producing new terminus traces from all newly acquired satellite images in Greenland.

### 4.1 Data Quality

Our network is capable of handling different image scales and resolutions, heavy shadowing, ice mélange, light cloud cover,

and Landsat-7 scan-line errors (Fig.4). Thus we can pick the terminus trace whenever it can be clearly seen in an image. Further, our screening module is capable of removing erroneous terminus traces generated from numerous causes (e.g., cloud cover, image resolution Fig. 5). With these removed a time series of terminus variation shows clear signals without spurious changes in terminus position (Fig. 6). Data quality is assessed via success rate and uncertainty. Our success rate is determined by examining how many terminus traces pass the screening module and dividing this by the number of images available for

each glacier. We find an average success rate of 64% (Fig. 7) but this varies temporally and spatially. Such variations are largely caused by the uneven distribution of the training data—glaciers with more training data have higher success rates. We have improved seasonality of terminus position, however the network does struggle to delineate termini in many wintertime Sentinel-1 images because of blurry boundaries and the lack of sufficient training data specifically using Sentinel-1 imagery. For example, we only have 484 Sentinel-1 traces from Termpicks and an additional 936 manually-prepared Sentinel-1 traces

as part of this work. As a result, many more traces from Sentinel-1 images did not pass the screening module (Fig. 8).

The uncertainties are measured in two ways: using duplicate traces and the MC dropout method. The MC dropout measures model uncertainties in neural network parameters, while duplicate traces quantifies the performance difference of the network on various datasets. Using duplicate traces, we find an average uncertainty of ∼37 m with a range from 10 to 204 m (Fig. 9a), comparable to traditional uncertainty quantification techniques for terminus data where uncertainties range from 33 to 108

m (Mohajerani et al., 2019; Zhang et al., 2019; Baumhoer et al., 2019; Cheng et al., 2020). The duplicate trace uncertainty varies between glaciers along with success rates largely because the training data is not evenly distributed for each glacier; glaciers with less training data will have larger uncertainties and lower success rates. Uncertainty also varies across sensor type. Figure 9b–f shows the uncertainties of different satellite sensors from MC dropout. Among the five datasets used, Landsat-8 and Sentinel-2 have the lowest average uncertainties since they have the highest spatial resolution. Landsat-7 images suffer

from the Scan Line Corrector (SLC) failure, which contributes to the uncertainties of the derived results. The reasons for the Landsat-5 uncertainty are twofold. First, Landsat-5 does not have a panchromatic band and thus, its resolution is coarser than other Landsat sensors. Second, floating ice tongues were more prevalent at the time of Landsat-5 data acquisition than they are now (Hill et al., 2018), which challenges the network to accurately delineate ice tongue edges without significant training data. The higher uncertainty of Sentinel-1 images is due to its low image quality, coarse resolution, and the lower volume of





training data derived from this sensor. Figure S4 shows multiple predictions of terminus traces resulting from MC dropout with comparison to the original terminus prediction for two glaciers. Due to the randomness of the network parameter that is shutdown during this calculation, MC dropout makes some predictions noisier (Fig. S4a). Further, we observe that prediction noise is larger when the original terminus predictions significantly deviate from reality (Fig. S4b).

The differences in the two types of uncertainties are caused by their quantification methods and source images. When using
MC dropout to quantify uncertainty, the model is varying, but the input images are fixed, while the situation is reversed when we quantify uncertainty measured by duplicate traces. As a result, the MC dropout uncertainty emphasizes uncertainty in the model itself, while duplicate traces relies on data uncertainty inferred from the difference between Landsat-8 and Sentinel-2 imagery. Additionally, the MC dropout uncertainty permits quantification of uncertainty for each dataset and is thus influenced by the characteristics of the training data as a whole, such as the SLC failure in Landsat-7. On the contrary, the uncertainty
from duplicate traces is more representative of Landsat-7 and Sentinel-2 than other datasets. Moreover, different ways of choosing source images in two types of uncertainties bring discrepancies. The source images for computing the MC dropout are randomly selected, but this is not true for duplicate traces. The dates having duplicate traces from both Landsat-8 and Sentinel-2 images are governed by satellite coverage. Overall, uncertainties from MC dropout and duplicate traces are roughly equivalent, especially for Landsat-8 and Sentinel-2 results since duplicate traces uncertainties are also based on these two
satellites (Fig. S5).

## 4.2 Data Quantity

Using the pipeline, we generated 278,239 glacier termini for 295 glaciers from 433,721 images (Fig. 7). Generally, we find that variations in satellite coverage causes significant spatial variations in image availability. For instance, in the central east Greenland (Glaciers #127 to #138), the relatively low number of images is caused by the shortage of Sentinel-1 images in this
region. There are only ~60 Sentinel-1 images in total for each of these glaciers, while other glaciers have 300-600 images available. We also find the launch of Landsat-8, Sentinel-1, and Sentinel-2 greatly improve the frequency of remote sensing images (Fig. 8) providing ~100 traces per year per glacier for the most recent (>2014) period. Figure 10 shows a heatmap of terminus traces for selected glaciers. The supplementary material provides similar heatmaps for the full record of glaciers (Fig. S6–S10). Importantly, Sentinel-1 images fill data gaps in winter when optical sensors struggle with low light conditions. These
wintertime terminus picks provide near continuous characterization of seasonality of terminus position (Fig. 11).

## 4.3 Ice Mask

In addition to terminus trace data, we also generate three new ice/ocean/bedrock masks for 2018, 2019, and 2020 (Fig. 12). Each newly generated ice mask is provided as a single Geotiff file with black representing the ocean, gray representing the bedrock, and white representing the ice (Fig. 12a). To identify how valuable updates to the ice masks are we compare our
masks with the 2015 GrIMP ice mask product (Howat et al., 2014) for each year (Fig. 12b–d). We find ongoing retreat of most glaciers after 2018 with glaciers in the northwest and southeast of Greenland dominated the retreating. The net area change of





ice extent is 520 km$^2$ for 2018, 660 km$^2$ for 2019, and 72 km$^2$ for 2020. The largest area change was 45.5 km$^2$ at Kjer Glacier, which was previously attached to a nunatak and has now detached from it and diverged into two tributaries (Glacier IDs #28 & #29). The one blue circle shows the advance of Jakobshavn Isbræ, which has been associated to regional cooling of ocean
water (Khazendar et al., 2019; Joughin et al., 2019).

### 4.4 Data Format

AutoTerm contains shapefiles of terminus traces and four supplementary data, including (1) a complete record of uncertainties, (2) identification of glaciers, (3) temporal coverage of terminus traces, (4) time series of terminus variations, and (5) ice masks. The terminus traces of a particular glacier are assembled in a single shapefile with an attribute table showing the metadata
of each trace. The metadata contains the date in YYYY-MM-DD, Glacier ID, source image satellite, and the uncertainty of each trace by averaging the two types of uncertainties provided. The entire record of uncertainties is provided in a spreadsheet containing the duplicate trace uncertainty and MC dropout uncertainties of all five satellites for each glacier. Data end users can choose an average of the two uncertainty measures as a total uncertainty or use one uncertainty value from the spreadsheet based on the prevalence of the data type used. The identification file includes the glacier location, ID, name, and region of
interest. For each glacier, we will provide a figure similar to Figure 8 showing the temporal coverage of terminus traces and a time series figure identical to Figure 6 showing the terminus variation. The temporal coverage and time series figures will be packaged into two KMZ files, respectively. In the KMZ files, the figures are assigned with the locations of their corresponding glaciers. By doing so, we can easily access the information on data gaps and terminus variation, comparing adjacent glaciers. The format of ice masks is described in section 4.3.

## 365 5 Discussion

### 5.1 Methodological Improvements

Building on previous DL-based studies, the major improvements we achieve in this work are 1) increasing the generalization level of the deep learning network to enable more and better quality terminus predictions; 2) deploying size normalization to improve the accuracy of terminus delineation for small glaciers; 3) designing a rigorous automated screening module to control
the data quality; and 4) automating several additional steps in the pipeline such as data collection and uncertainty quantification to allow the data to be regularly delivered.

The substantial generalization improvement we observe is due in large part to converting the TermPicks (Goliber et al., 2022) dataset into a rich training dataset. All previous DL-based studies use training data that is manually prepared by the individual authors with CALFIN having the most training data (1,773 training pairs; Cheng et al. (2020)). Because network generalization
is tied to the diversity of training data, small volumes of training data limit the ability of the network to generalize and thus reduce the accuracy of terminus predictions. Instead, we prepare the training data semi-automatically, and only manually check for mismatches between TermPicks traces and the source images, saving time. Further, TermPicks covers a larger variety of glacier conditions, geometries, and satellite sensor differences. In total, we have 16,440 training pairs from TermPicks and



1,466 training examples prepared manually. This diversity is much more representative of the real world and improves the
success of the network.

Despite numerous studies that have demonstrated the feasibility of using DL algorithms to automate terminus delineation,
there is an additional degree of automation needed to deal with the emerging big data now available on cloud services. Our
automated pipeline saves substantial manual effort, even though we still employ some manual effort, like preparing the regions
of interest. As the volume of images increases, so does the difficulty for the network to succeed on all of them. As a result, the
need for quality control becomes more paramount, particularly given that there are plans for follow-on Landsat missions ex-
tending terminus time series indefinitely into the future. Although we could devote more effort to manually preparing additional
training examples and improving the network accuracy, we opted to build a screening module enabling improved data quality.
This choice results in significant time savings over adding additional training data. Terminus data produced from machine
learning will always have larger uncertainty than manually delineated data since we use manually delineated data as our train-
ing data. The uncertainty of data generated from deep learning has been traditionally quantified by measuring the difference
between automatically picked termini and manually picked ones, which is rigorous but also requires significant manual effort.
Further, how representative such uncertainty is depends on the diversity of conditions covered by manual delineations. As a
result, improved uncertainty estimates come at the cost of labor required to compute them. Our implementation of duplicate
traces and MC dropout provides an estimate of uncertainty automatically while only sacrificing a modest amount of rigor over
manual delineation. For instance, if both duplicated traces are deviated from reality but are close to each other, the uncertainty
would not represent the reality.

Image normalization homogenizes images and thus eases the difficulties of terminus delineation under various conditions
(e.g., weather, illumination, geometry, etc.). In addition to histogram normalization (Zhang et al., 2021), we also conduct size
normalization to deal with the diversity of glacier sizes around Greenland. Although the design of DeepLabV3+ enables the
network to sense multi-scale contextual information, glacier sizes in Greenland vary by an order of magnitude (1–80 km in
width) necessitating size normalization. Since we upsample small images, size normalization is especially useful in increasing
the accuracy of terminus predictions for small glaciers and capturing detailed features in the terminus (Fig. 13).

## 5.2   Advantages of AutoTerm

Owing to the automation level we have achieved, AutoTerm produces terminus data with complete spatial coverage, sub-
seasonal sampling interval (Fig. 10), and full-width terminus morphology. Previous studies on terminus variation either have a
high temporal resolution (Schild and Hamilton, 2013; Kehrl et al., 2017; Fried et al., 2018; Catania et al., 2018) or complete
spatial coverage (Murray et al., 2015; Wood et al., 2021) but not both because of the laborious effort required with manual
terminus delineation. Even with DL-based terminus prediction, the most data available comes from CALFIN (Cheng et al.,
2020), which produced 22,678 terminus traces across 66 Greenland glaciers, limited in part because they only examined
Landsat imagery. Our inclusion of Sentinel-1 data improves the temporal sampling of the terminus data 3-fold over CALFIN,
providing an average sampling frequency of ∼100 traces per year for the most recent (>2014) period (Fig. 6b & 11). These
additional winter terminus traces allow improved accuracy for quantifying seasonality and inter-annual variability (Fig. 11).



Further, our ability to provide full-width terminus trace morphology enables detailed investigation of the specific processes controlling the ice-ocean interface (Murray et al., 2015; Fried et al., 2018; Rignot et al., 2016; Slater et al., 2021).

## 5.3 Limitations


Despite the success in automating the pipeline and producing a massive amount of terminus trace data, our workflow is limited by the immense computational power (120 GB of GPU memory) and long training time (5-7 days) required, which also makes uncertainty quantification challenging. This degree of processing time is due to the extensive volume of training data, which is crucial to generalizing the network and improving model performance. An additional limitation is caused by our assumption

that the screening module provides high quality results. This assumption rests on the choice of thresholds defined by the interquartile range in the screening module. Thus, when most results for a glacier are not credible, the screening module might not be able to clean the results because the random distribution of the terminus attributes leads to improper thresholds. The resulting terminus variation series could be spurious, and additional training data will be required to improve the data quality. A final limitation is that not all the data that can provide terminus trace information is included here. For example, there are

numerous satellite and airborne sensors that are not available on GEE (e.g., air photos, ASTER, and other SAR products). Our workflow is limited to what is available on GEE. As a result, AutoTerm only produces a high sampling frequency with winter traces after 2014.

## 5.4 Future effort required for maintaining the pipeline

Maintaining the longevity of the pipeline is essential as glaciers and ice sheets in our chosen regions undergo rapid and large-

scale changes with time. To continuously produce terminus traces each year in the future, the ROI for each glacier can be automatically updated based on the intersection between the glacier centerline and the most recent terminus trace. With an updated ROI, new images can be collected via GEE and the entire pipeline can be rerun to produce new terminus trace data for that year. Moreover, manually preparing additional training data might be required as the network could fail to pick terminus from new images. The pipeline can alert us of its failure based on the success rate within the screening module. Annually, these

terminus data can be used to calculate updated glacier terminus change data, which in turn informs the need for generation of new land, ice, and ocean masks. We can also update Glacier ID files triggered by the bifurcation or confluence of termini. For example, when a glacier retreats and in doing so, diverges into several tributaries or when an ice shelf collapses and exposes new glacier termini, the existing Glacier IDs (numbers) can be suffixed with letters (abc...) indicating that the origin of each tributary is embedded within the ID. When several tributaries merge into one main terminus, for example through advance,

the ID of the largest tributary will be kept. Lastly, we depend on future community feedback about our products to assist in identifying issues not caught by our screening module. This is because the massive amount of data precludes the ability to guarantee the quality of each individual trace.



## 6   Conclusions

This study builds a fully automated, deep-learning-based pipeline that can continuously produce terminus traces from multi-sensor remote sensing images. We convert a large volume of manually-picked terminus traces to be used as training data, allowing the network to tackle diverse conditions found in "Big Data." In addition to terminus delineation, we automate data collection, quality control, and uncertainty estimation in order to generate a terminus dataset with comprehensive spatial coverage and dense temporal sampling, which we call AutoTerm. AutoTerm covers 295 outlet glaciers in Greenland and contains 278,239 terminus traces with controlled quality and uncertainties. The comprehensiveness of the terminus dataset will benefit the community for conducting a pan-Greenland investigation of terminus variation and model-based parameterizations of ice-ocean interactions. Owing to the transferability of deep learning, the entire pipeline has the potential to be applied to many other outlet glaciers around the world.

*Code and data availability.*   The codes of the pipeline is available at https://github.com/enzezhang/AutoTerm. All the data including terminus traces, inventory, uncertainties, ice mask, and terminus variation are submitted to NSIDC and pending approval. Before approval, data is available at https://doi.org/10.5281/zenodo.7190740. The data will be version controlled through community feedback and manual inspection.

*Author contributions.*   EZ developed the code, performed the data processing, and wrote the manuscript. GC and DT advised EZ and revised the manuscript.

*Competing interests.*   The authors declare that they have no conflict of interest.

*Acknowledgements.*   We acknowledge partial funding for this work from NASA (Grant 80NSSC21K0903) and the Institute for Geophysics Postdoctoral Fellowship at the Jackson School to E. Zhang.





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





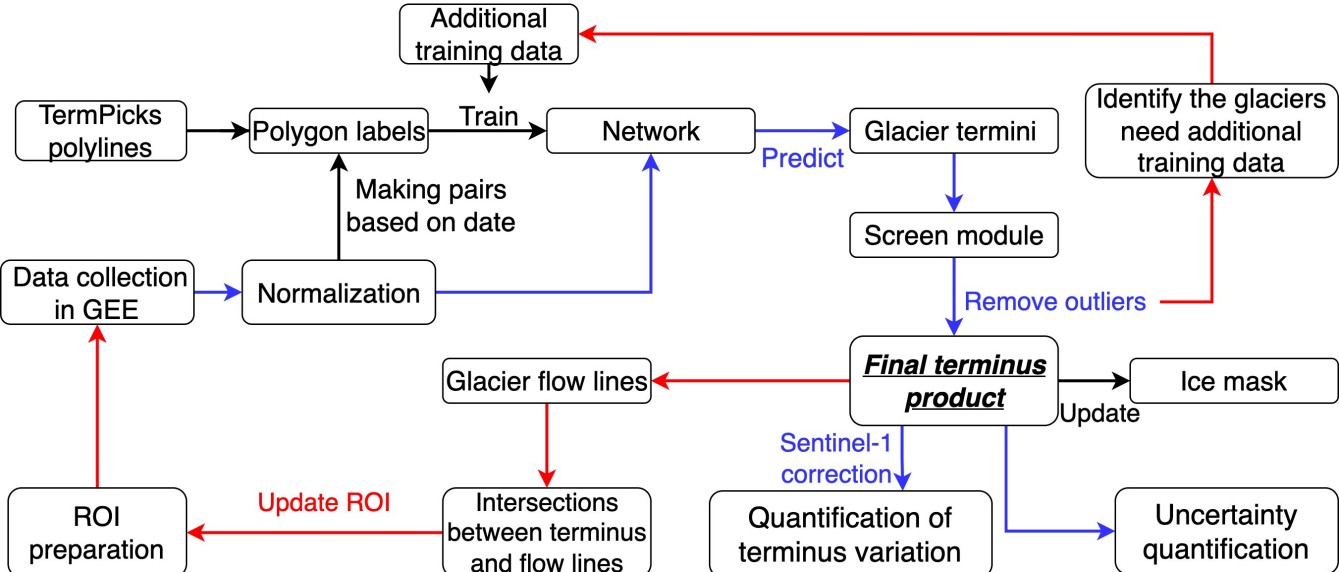

**Figure 1.** Our automated deep learning pipeline. The blue arrows represent the procedures that are fully automated when generating glacier terminus traces. The black arrows represent aspects related to the training data via TermPicks traces and is done semi-automatically. The red arrows represent the procedures in the workflow emplaced to maintain the longevity of producing terminus traces through automation.



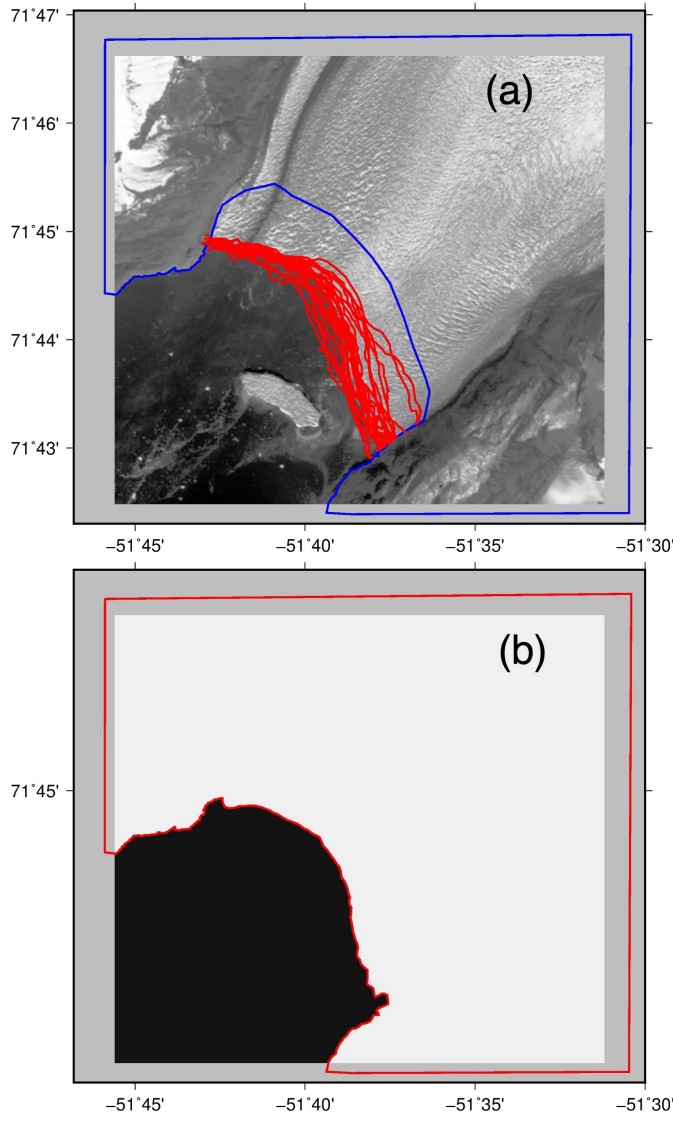

**Figure 2.** An example of converting a polyline into a polygon label and producing a labeled figure. (a) The source image, TermPicks traces (red curve), the reference polygon for this glacier (blue polygon). The terminus of the reference polygon is upglacier from all the TermPicks traces. (b) The red polygon shows a converted polygon label from one of the TermPicks traces, and the binary label image is derived from the polygon.




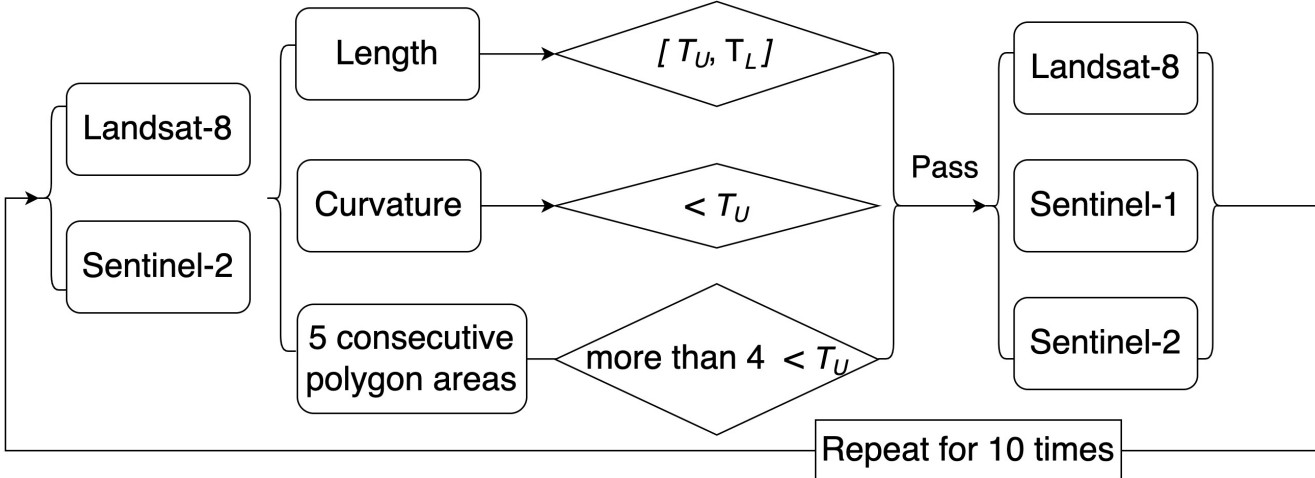

**Figure 3.** The pipeline of the screening module. $T_U$ is the upper threshold and $T_L$ is the lower threshold. Each metric (length, curvature, 5-consecutive areas) has its own threshold. Only the results from optical images are used to calculate the thresholds, and the thresholds are applied uniformly to all the datasets.



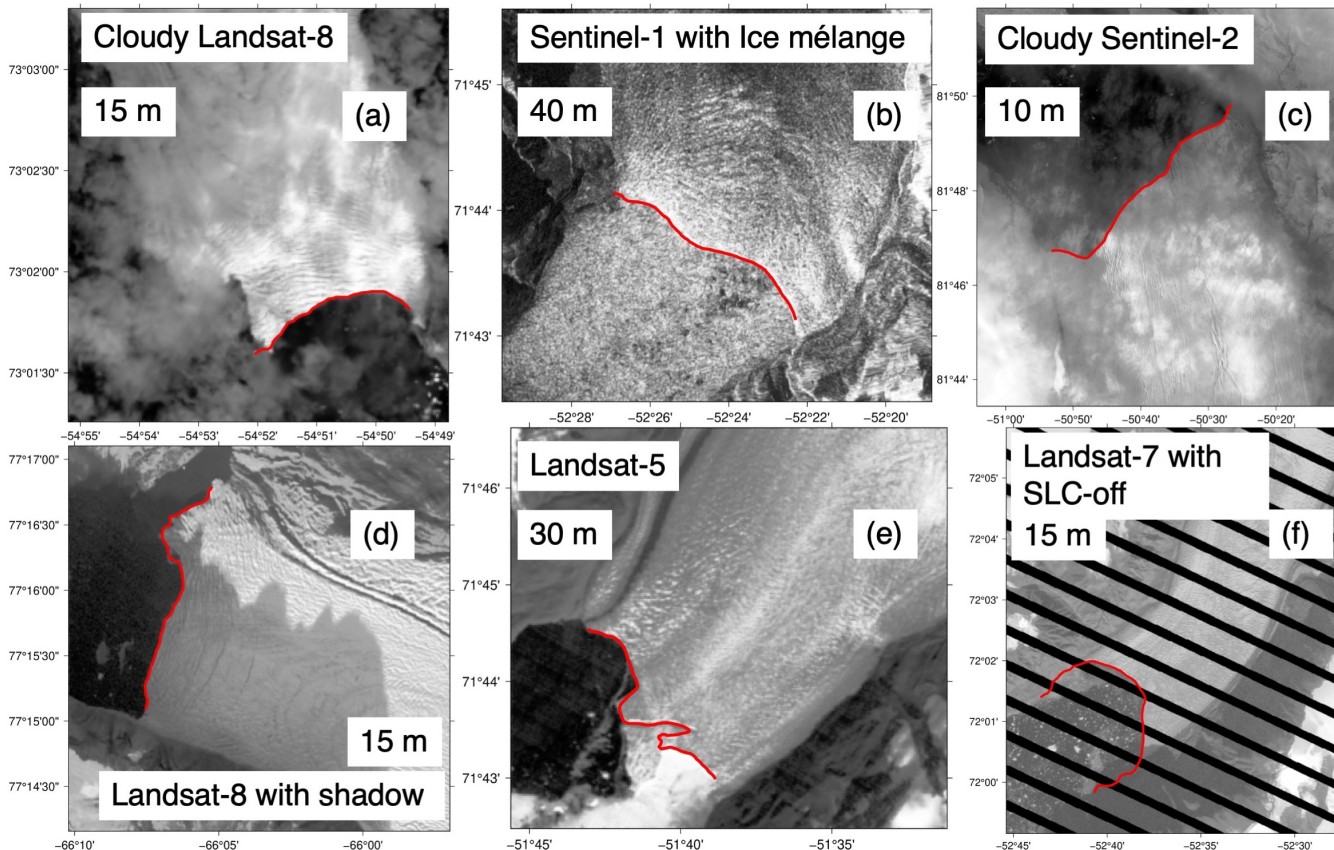

**Figure 4.** Examples of the automatically picked glacier terminus. The network can handle different scales/resolutions, light cloud cover (a & c), ice mélange (b), heavy shadowing (d), complex geometry (e), and Landsat 7 scan-line errors (f).



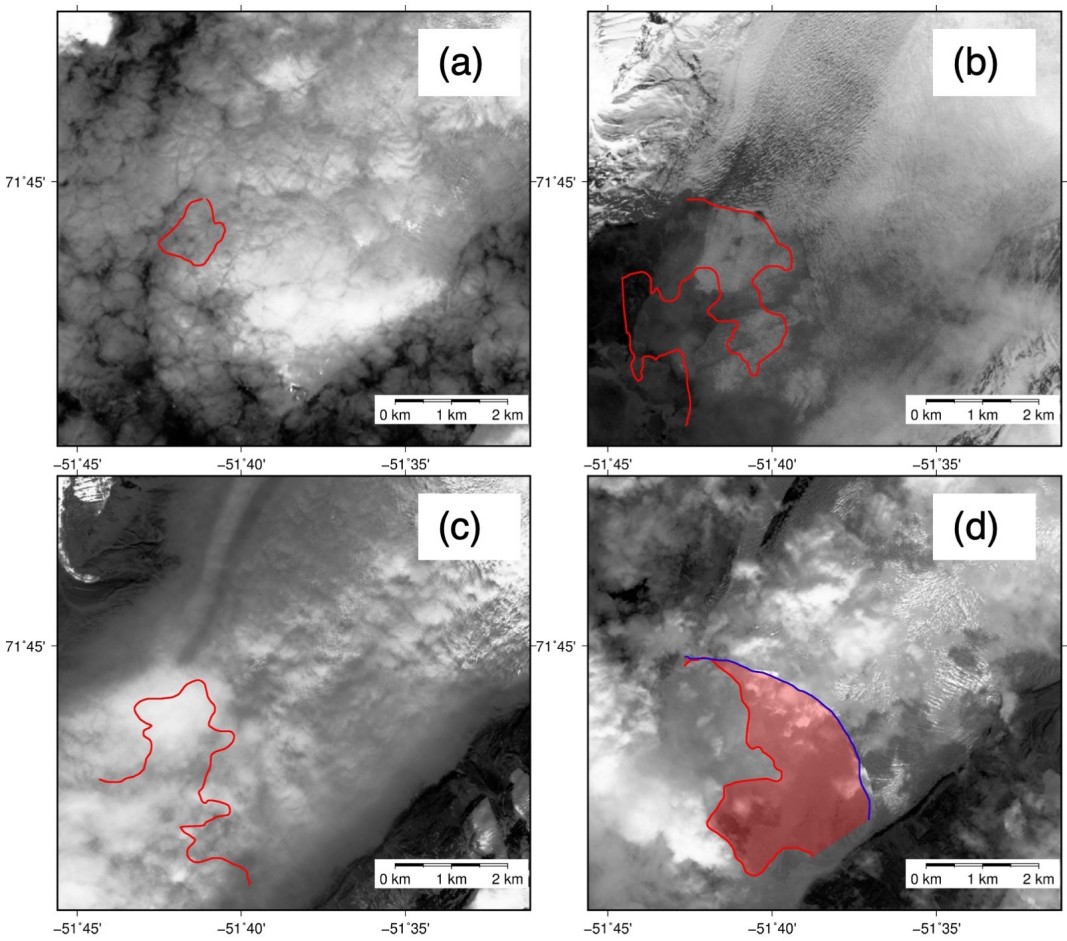

**Figure 5.** Examples of results abandoned for different reasons. (a) Short terminus. (b) & (c) Long and complex terminus. (d) Terminus forms a large polygon with its adjacent picks. The backgrounds are the source images of the wrong picks. The red line in (d) is the wrong pick and the blue curve shows its time-adjacent manual pick.



**Figure 6.** (a) An example of terminus variation over time from our results showing clear seasonal and longer-term signals in terminus change. We highlight the ability of our screening module to detect erroneous traces (red x's). After 2014, seasonal variations are more apparent owing to the addition of wintertime records from Sentinel-1. (b) Detail of (a) over 2013-2021 showing the comparison between our results, manual traces from Termpicks covering 2013-2020 (Goliber et al., 2022) and CALFIN covering 2013-2019 (Cheng et al., 2020). Uncertainties in our results are shown as vertical bars for each terminus trace and are measured by duplicate traces.





**Figure 7.** (a) Total number of images and (b) overall success rate of AutoTerm for each glacier. The ellipse in (a) indicates the Glacier #127 to #138 with relatively low numbers of images. The spatial variations of image numbers are caused by the variations in satellite spatial coverage. The spatial variations in success rates are caused by the uneven distribution of training data. Glaciers with more training data have higher success rates. The boundary of the GrIS is provided by The Generic Mapping Tools (GMT, https://www.generic-mapping-tools.org/).



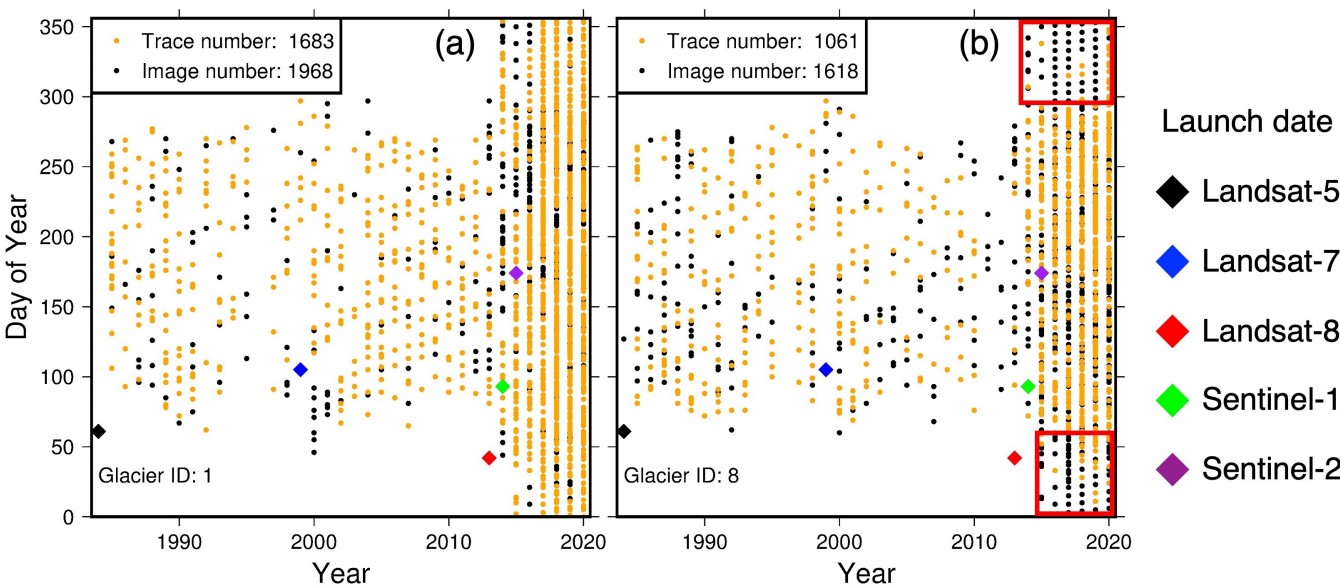

**Figure 8.** Examples of remote sensing image availability (black dots) versus termini successfully predicted for (a) Glacier #1 and (b) Glacier #8. The diamonds show the launch date of the satellites. Since 2014, the launch of Sentinel-1 (green diamond) fills the gap in winters. Due to the blurry boundaries of wintertime Sentinel-1 images, some of these terminus predictions did not pass the screening module (red boxes in b).







**Figure 9.** Terminus trace uncertainties measured by duplicate terminus traces (a) and MC dropout (f) for each glacier. Gray indicates no uncertainty is measured due to data shortage (either no duplicate trace for a or no source image for b–f). The boundary of the GrIS is the same with Fig. 7, provided by GMT.





**Figure 10.** Example heatmap of the number of successful traces predicted in each year for Glaciers #1-50. The full record of annual trace numbers can be found in the supplementary material.







**Figure 11.** An example showing the importance of including Sentinel-1 traces for Glacier ID #164. (a) With Sentinel-1 (green circles), Landsat-8 (red circles), and Sentinel-2 (purple) and (b) with only Landsat-8 and Sentinel-2 data. In (a), we quantify the inter-annual and seasonal variation in terminus position using the singular spectrum analysis method (Zhang et al., 2018).



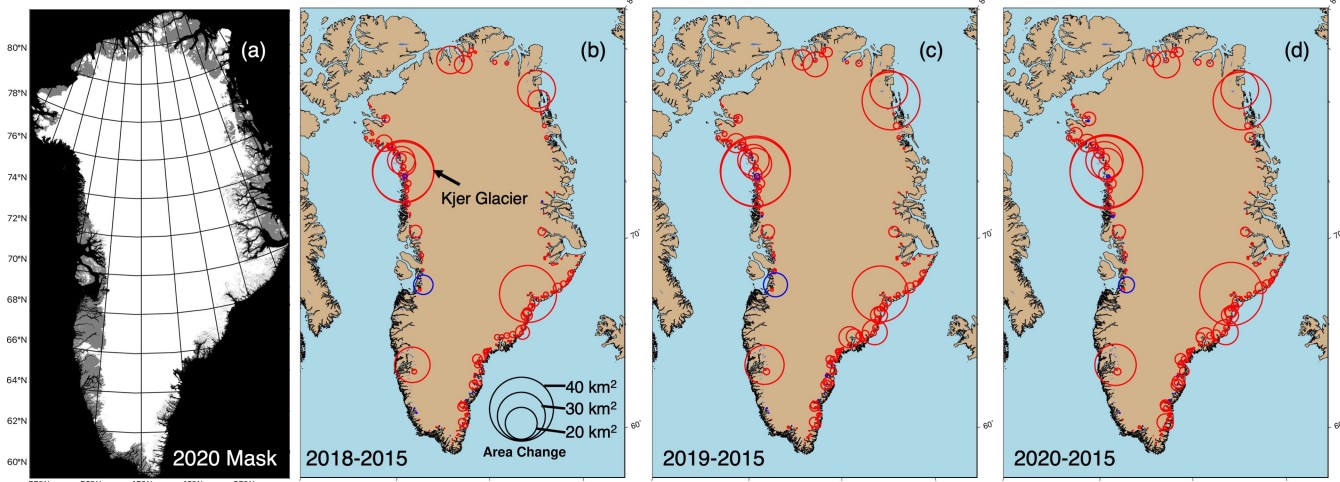

**Figure 12.** An example of an updated ice mask for 2020 (a) and the terminus change between the updated masks and the original 2015 GrIMP ice mask (b–d; 2018-2020). Red circles represent retreating glaciers and blue circles represent advancing glaciers. The size of the circle indicates the difference in area change of each glacier from the original mask. The boundary of the GrIS in (b–d) is the same with Fig. 7, provided by GMT.



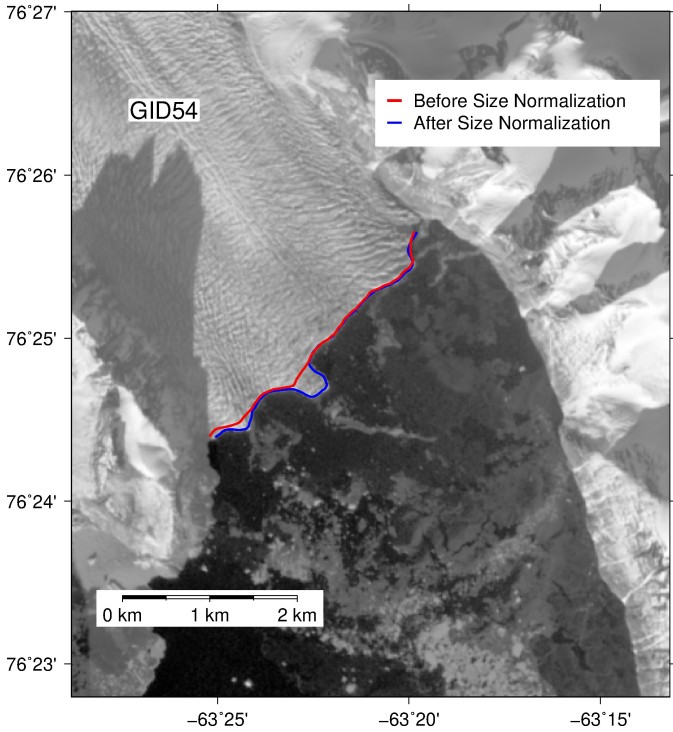

**Figure 13.** An example showing the effect of size normalization for Glacier ID #54. After normalization, delineation of the terminus is more accurate and captures small features.



| Sensor | Coverage | Resolution | Repeat | Time Range | Access | DAAC |
|---|---|---|---|---|---|---|
| Landsat 5 | global | 30 m | 18 d | 1972-2013 | Public | LP |
| Landsat 7 | global | 15 m | 16 d | 1999-2013 | Public | LP |
| Landsat 8 | global | 15 m | 16 d | 2013-present | Public | LP |
| Sentinel-2 | global | 10 m | 10 d | 2015-present | Public | LP |
| Sentinel-1 | global | 10 m | 6-12 d | 2014-present | Public | ASF |

**Table 1.** Satellite missions with publicly available data on Google Earth Engine for terminus extraction.