# Peer review of "AutoTerm: an automated pipeline for glacier terminus extraction using machine learning and a "big data" repository of Greenland glacier termini"

_EGUsphere, 2022_

## Author Comment (AC1)

**General Comments**

This paper presents an automated pipeline in Google Earth Engine for glacier terminus tracing together with a so-derived dataset and updated ice/ocean masks. Such a pipeline is highly needed and of great significance to the community. This extent of automation has not been reached in related works. We thank the authors for their valuable contribution!

While this paper employs a sound deep learning architecture in combination with a promising screening module, I have several major concerns, including the technical correctness of the evaluation protocol and, thus, the validity of the proposed study, as the generalizability of the deep learning network still needs to be proven. Furthermore, comparisons to other studies need to be conducted in a technically correct way, and the reproducibility of the study needs to be ensured by making the assembled training dataset publicly available. Lastly, the structure of the manuscript should be improved upon.

We greatly appreciate the detailed and thorough review by Reviewer 1. We have made our best effort to revise the manuscript based on the referee's comments and suggestions. Below is an item-by-item response to the specific comments by this reviewer.

**Major Concern 1: Evaluation Protocol**

The pipeline has not been properly tested, and hence, we can not yet rely on its output. In my understanding, the authors seem to confuse uncertainty estimation with error assessment. In line 245, they call the calculation of the difference between prediction and ground truth „uncertainty quantification". The authors then claim that comparing to manually picked traces „requires significant manual effort" because it would have to be redone, as „network accuracy likely varies over time as glaciers experience different conditions". Instead, the authors use two different uncertainty quantifications that do not rely on ground truth data. Calculating uncertainties is definitely useful, and the two used ways of calculating the effect of different sources of uncertainty (model inherent and input inherent) look very promising. However, calculating the uncertainty is no substitute for an error assessment. The authors themselves state in line 395: „if both duplicated traces are deviated from reality but are close to each other, the uncertainty would not represent the reality." It is, therefore, indispensable to calculate the deviation of the network's predictions to manually delineated ground truth traces on a test set that is independent of the train set. First, we need to know how well the network is performing at the moment before we apply it to new unseen data and afterward assess whether the network's performance degrades when new sensors are used or other conditions change (called domain shift in machine learning).

We agree with the reviewer that the difference between predictions and ground truth should be called "error", while the difference between duplicate traces should be taken as "uncertainty". We have identified places in the manuscript where this terminology may have been confused and have updated the text. In addition, we have performed a test of the network as follows. We randomly choose 100 traces from TermPicks as a test dataset and use the rest of the TermPicks data to train the network from scratch. After training the network, we apply it to a test dataset and quantify the deviation of the network's prediction to manual delineations in the test dataset. This reveals a test error of 79 meters, which is similar to previous authors (Mohajerani et al., 2019; Zhang et al., 2019; Baumhoer et al., 2019; Cheng et al., 2020). The description of this test is added to the manuscript in Line 214 and Line 316.

Quantifying error based on manual delineation involves a trade-off: the more representative the error is, the more manual effort it takes. Since we aim to produce as large a terminus dataset as possible (with a resulting 278,239 glacier termini), a highly representative error would require too much manual effort, which violates our primary objective to save manual effort. For this reason, we still keep the two automated ways to quantify the uncertainty of the terminus data. We agree that uncertainty and error are not the same.

Although the reviewer states that we cannot rely on our model output, even without the model test we have now performed, we believe our data to be reliable for the following reasons. First, our terminus traces match the remote sensing images (Fig. 4). Second, the time series of terminus variation are in agreement with both Termpicks and CALFIN (Fig. 5). Third, the time series of terminus variations show a clear seasonal signal (refer to the time series data described in section 4.4), which would not be revealed if our terminus traces are unreliable.

Additionally, an experiment should be conducted to determine whether and by how much the error between prediction and ground truth on the test set is reduced when the screening module is applied versus not applied. In this way, the effectiveness of the screening module can be demonstrated. The same holds for the upsampling of small images (it is not sufficient to visualize the results of one sample, as shown in Fig. 13).

The screening module belongs in post-processing, and is thus not related to network inference or training. Instead, the screening module is for detecting outliers in order to improve data quality. Fig. 3 and the red crosses in Fig. 5 demonstrate the effectiveness of our screening module. For these reasons, we did not see it necessary to validate the effectiveness of the screening module on a test set.

We agree with the reviewer that the test error is needed to demonstrate the effectiveness of upsampling as it is a pre-processing procedure. Thus, we have also conducted an upsampling test. For this test, we randomly select 36 images of five small glaciers to be a test set for size normalization. These images are not included in the training set for the independent evaluation of the size normalization's effectiveness. We add a new table to show the test error and uncertainty from duplicate traces with and without size normalization (Table S2). The results show that size normalization can effectively reduce test error and uncertainty. The related description is now added in Line 429.

**Major Concern 2: Generalizability**

The pipeline has to be tested on out-of-sample data (i.e., glaciers not present in the training dataset) and data outside of Greenland to show generalizability to the global scope.

1. Line 451 „Owing to the transferability of deep learning, the entire pipeline has the potential to be applied to many other outlet glaciers around the world"
2. Line 135 „converting the TermPicks terminus data into a training dataset suitable for deep learning highly generalizes the network"

These claims have to be proven on such a test set. As most manually annotated traces available from related work are part of TermPicks and hence, have been used for training, another test set has to be used. For testing on SAR imagery, the dataset provided by Gourmelon et al. could, for example, be used, as it is not incorporated in TermPicks (except Jacobshaven, which probably has overlaps with TermPicks). However, test data for optical imagery might have to be created manually (e.g., from Antarctica or the Russian Arctic). At least, I am unaware of a dataset based on optical imagery that is not incorporated in TermPicks.

The importance of the size of training data in the deep learning field has been well demonstrated. For instance, Sun et al. (2017) showed that the network's performance increases logarithmically based on the volume of training data size. For this reason, we see no need to provide additional proof that our model has the ability to generalize. Further, our study focus is on Greenland alone – not on a global scale. We merely identify the potential for our model to be used at that scale. That said, out of interest we conducted an experiment in which we trained the network with only 1466 training examples prepared manually without including TermPicks. The test error of this network is 315 meters, which is much larger than 79 meters that we have after training with TermPicks. Such an improvement demonstrates the generalization improvements brought by TermPicks. The related description is added in Line 405.

**Major Concern 3: Comparability**

It is not possible to compare the calculated uncertainties of this manuscript to the errors calculated in related works, as done in, e.g., line 304 or line 379. Two totally different metrics are compared here, and studies have been conducted on different datasets. For a valid comparison, the exact same network/pipeline needs to be tested on different datasets, or different networks/pipelines have to be trained, optimized, and tested on the exact same data (a so-called benchmark dataset). Altering both the dataset and the network/pipeline introduces too many changes, and a changed performance could result from either the different dataset (for example, the test set might be easier, and therefore, the performance of an otherwise worse performing network would be better on this test set) or the different network/pipeline.

Concludingly, the claimed improvements 1 and 2 (line 377 „1 increasing the generalization level of the deep learning network to enable more and better quality terminus predictions; 2 deploying size normalization to improve the accuracy of terminus delineation for small glaciers") are not proven.

One way to show the superior terminus prediction performance on SAR imagery could be the use of the benchmark dataset recently proposed by Gourmelon et al. (2022) (i.e., retraining the pipeline on the train set and evaluating it on the test set using the stated metrics). To the best of our knowledge, there is no equivalent benchmark dataset for optical imagery.

Now that we have performed an error estimate we can more accurately compare our network error to that from other studies (Line 315), and find that our network's performance is comparable with other studies, not superior or inferior.

We agree with the review that different networks should be trained and tested on the same benchmark dataset to determine the best. However, such a test is outside of the scope of this study. The objective of this study is not to demonstrate which deep learning network is better but to generate a terminus dataset with spatial coverage and temporal resolution that no prior study has with the addition of further automation in the production pipeline.

For generalization and size normalization, please refer to the response of Major Concern 3 and Major Concern 1, respectively.

**Major Concern 4: Reproducibility**

Please make your complete assembled training data (including the satellite imagery) publicly available, as only in this way the reproducibility of the results is guaranteed. Moreover, please also provide the manually created reference polygons for each glacier.

All our remote sensing images are freely available on Google Earth Engine. The code for collecting the data is also available on GitHub. TermPicks is also a publicly available dataset. The reference polygons and label polygons converted from TermPicks traces have been included in the AutoTerm dataset now (10.5281/zenodo.7527485).

**Major Concern 5: Structure of the manuscript**

The structure of the manuscript has to be improved upon. There is a mix-up between the training and inference of the pipeline, and some information is given twice at different positions in the manuscript. It is hard to tell when the authors write about the newly derived dataset in contrast to the dataset derived from TermPicks for training the network, e.g., in line 295 („We find an average success rate of 64%"), it is unclear on which dataset the success rate was calculated. I would suggest splitting the manuscript into two main parts as follows, but there could also be another better split-up:

1. Training Pipeline: manual delineated dataset creation (TermPicks + additional manual annotations), neural network (architecture), network training (train-validation-test split, learning rate, number of trained epochs, etc.), screening module, error calculation, uncertainty estimation

2. Inference Pipeline: new data acquisition + pre-processing, uncertainty estimation on this newly derived dataset, ice/ocean mask updates

We thank the reviewer for these comments. We decided to structure the manuscript following the order of data processing. We first collect remote sensing images and conduct preprocessing (section 3.1). Second, we generate the training dataset by converting the terminus traces in TermPicks into label polygons and pairing polygons with the remote sensing images. Third, we introduce the network structure and the training progress in section 3.3. Sections 3.4 to 3.7 are post-processing procedures after applying the well-trained network to make inferences on the 433,721 images collected via Google Earth Engine. That said, we notice that the original section titles may have caused some confusion. Thus, to address this comment we changed the title of Section 3.2 from "Generalizing the network" to "Generating training data from TermPicks", and the title of Section 3.3 from "Deep Learning Network" to "The Structure and Training of Deep Learning Network".

Furthermore, we have revised some of the text to make it clearer that we first perform network training and then we apply the well-trained network to all the images collected through Google Earth Engine to generate terminus positions. This is done in the last sentence in Section 3.3 as "*After the training, we apply the well-trained network to the test set for quantifying the test error and to all the images collected via GEE for generating the terminus dataset.*" Line 217.

The success rate is the percentage of terminus traces that pass the screen module. We add one sentence at the end of section 3.4: "*Finally, we estimate the success rate by calculating the percentage of the terminus traces that pass the screening module.*" Line 255.

In section 4.1, the authors first introduce the ‚success rate', which should, however, be introduced in the methods section.

We add one sentence at the end of section 3.4: "*Finally, we estimate the success rate by calculating the percentage of the terminus traces that pass the screening module."* Line 255.

Moreover, paragraph line 188 to 195 should be moved to limitations.

Line 194 to 201 describe the limitations of the input imagery, not the limitations of our work. These points are discussed here because they explain why we needed to prepare additional training data. Therefore, we think this text fits better in Section 3.2, which describes the training data preparation.

An explanation of the two uncertainty measures, as given in lines 319 to 323, should be moved to further at the beginning of the manuscript.

The principle of the two uncertainty measures is described fully in Methods (Section 3.6). Lines 344 to 355 explains the uncertainty results and focus on why these two uncertainties are different. Therefore, we believe lines 344 to 355 fit better in the Results section.

**Major Comments:**

1. It is unclear to me whether the name „AutoTerm" refers to the automated pipeline or the derived dataset, or both.

We thank the reviewer for pointing this out. We think it is a good idea to use AutoTerm reflect both the data and pipeline. We have changed the title of the manuscript to: "AutoTerm: an automated pipeline for glacier terminus extraction using machine learning and a "big data" repository of Greenland glacier termini."

2. The title of the manuscript does not mention the automated pipeline, which is, in my humble regard, the most significant contribution. Hence, I'd argue for a more suitable title, e.g. AutoTerm: an automated Google Earth Engine pipeline for glacier terminus extraction and „big data" repository of Greenland glacier termini

We thank the reviewer's suggestion. We have changed the title of the manuscript to: "AutoTerm: an automated pipeline for glacier terminus extraction using machine learning and a "big data" repository of Greenland glacier termini."

3. It needs to be clarified whether the region of interest that has to be defined for each new glacier has to be a polygon like in figure 2 or whether it can simply be a bounding box.

Each region of interest is a bounding box. The polygon in Figure 2 is a reference polygon only for converting TermPicks traces into a training label polygon, which is described in the Figure caption. We have updated the text where needed to clarify this point. Line 146.

4. Line 163 onwards: „This allows glaciers with various natural sizes to have a similar image size in computer vision, which largely decreases the complexity of delineating glacier terminus." This statement (the second part of it) needs more explanation or a reference.

We rephrased the text here and added a reference: "*We then normalize the image size, which is commonly adopted in the computer vision field for better capturing object features with various physical sizes (Xu et al., 2017).*" Line 167.

5. The normalization of image sizes is not clear to me. Small images are upsampled, but large images are not downsampled. Hence, do they still have different sizes? I would not call this normalization, then. Moreover, the authors extract patches afterward, so the input size is always equal anyways. Additionally, only showing one figure that shows an improvement for one trace is not sufficient evidence that this upsampling generally improves the delineation performance. Please show the improvement in numbers over a complete, independent test set (refer to major concerns 1 and 2).

Different glaciers are not of the same physical size and therefore don't have similar number of pixels that fall within their fjord walls. This can vary from 300 pixels to 3000 pixels. As a result, the deep learning input (image patch) may sometimes cover more than the entire glacier and may alternatively sometimes only cover a part of a glacier.

After normalization, all small glaciers will be covered by images with a width of about 1000 pixels, regardless of their original image width. In other words, the normalization makes glaciers appear to the deep learning network as if they had a similar physical size. We have attempted to clarify this in the text. Line 168.

We agree with the reviewer that test error is needed to demonstrate the effectiveness of upsampling as it is a pre-processing procedure. We randomly select 36 images of five small glaciers that are beyond the training set as the test set for size normalization. We add a new table to show the test error and uncertainty from duplicate traces with and without size normalization (Table S2). The results show that size normalization can effectively reduce test error and uncertainty. The related description is now added in Line 428.

6. Section 3.3:

   1. „encoder-decoder structure [...] can obtain sharp object boundaries": Actually, an encoder-decoder structure without skip connections would most probably not recover any details and, therefore, no sharp object boundaries. In Chen et al. (2018), they use a more sophisticated method to obtain the sharp boundaries: „A fully connected CRF [conditional random field] is then applied to refine the segmentation result and better capture the object boundaries."

      We realize that there is a mistake in our citation. The original DeepLab use the CRF to refine the segmentation results (Chen et al., 2018a, DeepLab: Semantic Image Segmentation with Deep Convolutional Nets, Atrous Convolution and Fully Connected CRFs). Later on, the same author improved the DeepLab and developed DeepLabV3+ (Chen et al., 2018b; Encoder-Decoder with Atrous Separable Convolution for Semantic Image Segmentation), which is the network we use in this manuscript. The DeepLabV3+ adds a simple yet effective decoder module to refine the segmentation results especially along object boundaries (Chen et al., 2018b).

   2. „atrous convolution [...] senses multi-scale contextual information": It is not the atrous convolutions alone that make recognition of multi-scale contextual information possible, but the combination of atrous convolutions in ASPP. "Atrous convolution allows us to explicitly control the resolution at which feature responses are computed within Deep Convolutional Neural Networks. It also allows us to effectively enlarge the field of view of filters to incorporate larger context without increasing the number of parameters or the amount of computation. Second, we propose atrous spatial pyramid pooling (ASPP) to robustly segment objects at multiple scales." (Chen et al., 2018)

      We thank the reviewer's comments and changed "atrous convolution" to "atrous spatial pyramid pooling".

   3. „multi-scale contextual information [...] [is] helpful for our task since [...] we integrate remote sensing datasets with different spatial resolutions": Multi-scale refers to how many pixels a neuron is able to see (effective receptive field) and not how much square meters one pixel can see. Hence, multi-scale contextual information helps when the calving front covers many versus only a few pixels. Thus, it helps only indirectly with different spatial resolutions of the dataset.

      We agree with the reviewer's comment. Multi-scale refers to the network's ability to sense various portions of the images but not sense images with different resolutions. We rephrase the sentence as "*Sharp boundaries can improve delineation accuracy, and sensing multi-scale information helps indirectly when we integrate remote sensing datasets with different spatial resolutions.*" Line 205

   4. „This network has been proven to have large learning capability, spatial transferability [...]": These are quite big claims based on a train set of two glaciers and a test set of one glacier that are all located in Greenland (Zhang et al., 2021).

      The large learning capability has been demonstrated by the paper of DeepLabV3+ (Chen et al., 2018b). Zhang et al. (2021) applied the network to a glacier beyond the training set, showing the network's spatial transferability. In this work, we apply the network to 295 glaciers in Greenland and generate 278,239 glacier termini, of which 17,906 terminus traces are from the

training set. This means that 94% of our results are beyond the training set, which further demonstrates the learning capability and spatial transferability of the network.

5. „The network is trained with a learning rate of 0.005 [...] as recommended by (Zhang et al., 2021)“: The optimal learning rate for training is highly dependent on the dataset as well as on the batch size (not just the model). Hence, the learning rate has to be treated as a hyperparameter, which has to be optimized on a validation set (not the test set). A sub-optimal learning rate can lead to significantly longer training times until convergence or no convergence at all.

We agree with the reviewer and have tried to train the network with learning rates of $5×10^{-3}$, $2×10^{-3}$, and $1×10^{-3}$. The validation losses for these three learning rates are 0.023, 0.020, and 0.024, respectively. Overall, the validation losses are measured using binary cross entropy and are similar to each other. We chose the learning rate ($2×10^{-3}$) with the lowest validation error. We rephrased the sentence as: "*Based on the learning rate in Chen et al. (2018b) and Zhang et al. (2021), we train the network with learning rates of $5×10^{-3}$, $2×10^{-3}$, and $1×10^{-3}$, and choose $2×10^{-3}$ owing to its lowest validation loss.*" Line 210.

6. „we choose the largest batch size (16)“: This should be „largest possible batch size (16) on an A100 GPU with 40/80 GB GPU memory“. Please specify whether your A100s have 40 or 80 GB GPU memory.

The sentence is rephrased as follows: "*we choose the largest possible batch size (16) on four A100 GPUs with 160 GB GPU memory in total. We set the batch size to a power of two to take full advantage of GPU processing (Kandel and Castelli, 2020)*" Line 212.

7. What exactly is meant by "maximize our computational power“ in line 204?

It means using as much GPU memory as possible. The sentence is now rephrased as "*we choose the largest possible batch size (16) on four A100 GPUs with 160 GB GPU memory in total.*" Line 212.

8. „The network training takes about a week“: This is quite long and might be due to a sub-optimal learning rate. Please specify not only the training time but also the number of trained iterations over the complete augmented dataset. Also, specify your train-test-split and your metrics for evaluation (refer to major concern 1). Moreover, did you use an early stopping criterion? You might have overfitted during this long training time.

The long training time is caused by the large training dataset. We have 17906 training examples, and the augmentation increases the training set by a factor of four. The training takes more than 600,000 iterations. Among all the training data, we select 5% of images randomly as validation datasets to conduct early stopping. The training will be stopped when the validation error stops decreasing for 3 consecutive epochs. We add these details in Line 214.

7. Line 210 „do not have any quality control“: At least Cheng et al. have manual control. So, maybe rephrase it to „do not have any automated quality control“.

We thank the reviewer for pointing that out. Cheng et al. (2021) have an automated data screening based on the deviations of two classifications of the network.

We have changed the related text as: "*Many previous DL methods applied to terminus delineation do not have quality control (Mohajerani et al., 2019; Zhang et al., 2019). Where it does exist, data*

*screening has been simplistic and not automatically applied. For example, Zhang et al. (2021) only considers the complexity of the terminus shape and removes traces with abnormal complexity (which, in turn, requires a threshold to be established for each glacier), Baumhoer et al. (2019) only considers outliers that arise in a time series of terminus position change over time, and Gourmelon et al. (2022) remove the outliers based on terminus length. Cheng et al. (2021) however did design an automated data screening based on the deviation of two classifications from the network. Our screening module is based on using the physical properties of glacier termini.* ” Line 224.

8. Line 215 onwards: Please mention that the screening builds on top of existing works here (Zhang et al. 2021 – terminus curvature screening, Baumhoer et al. 2019 – time series outliers, Gourmelon et al. 2022 – removal of too short termini predictions), but goes one step further, i.e., doesn't use any manual intervention or prior knowledge of the data.

The related sentences are revised as follows: “*Based on the previous works (Baumhoer et al. 2019 , Zhang et al. 2021, Gourmelon et al. 2022), we develop an automated screening module that forgoes any manual intervention or prior knowledge of the data.*“ Line 232.

9. Line 217 „Terminus length is determined by the sum of the piece-wise length along an individual terminus trace“. Please explain in more detail. This, at least for me, is hard to understand.

Each terminus trace is an ordered set of points. The length is the sum of the length between the two closest points. The description refers to how we calculate the terminus length, which might be confusing to readers. Terminus length is just the physical length of the glacier terminus. As a result of this confusion, we decided to remove this sentence.

10. Line 218 „Terminus curvature is computed between two adjacent points for each point along the terminus and then an average is taken for each terminus trace.“ This is also not completely clear to me. I think an equation would help.

We have rephrased the related text as: “*Terminus curvature is computed among every three adjacent points along the terminus based on Peijin Zhang's work (https://github.com/peijin94/PJCurvature), and then an average is taken for each terminus traces.*”  Line 235.

11. Line 224 „percentile of the data range“: Do you refer to the data range of the generated training data? Is this computed per glacier? Per satellite? The validity of these thresholds needs to be checked on an independent test set (refer to major concerns 1 and 2).

For each glacier, we will calculate the thresholds based on the termini from the same satellite. For instance, we will calculate thresholds for Sentinel-2 traces of GID2 and Landsat-8 traces of GID2 separately. We add one sentence in Line 242 to provide additional clarification: “*The thresholds are calculated automatically based on the results of the same glacier and same satellite.*”

The screening module belongs to the post-processing, which is not related to network inference or training. The screening module is for detecting outliers and maintaining the data quality. Fig. 3 and the red crosses in Fig. 5 demonstrate the effectiveness of our screening module. So, we did not validate the effectiveness of the screening module on a test set.

12. Line 227 „For outliers in terminus length, we remove both the lower and upper thresholds (Eqns. 1 and 2) because we do not anticipate large changes in terminus length in either direction (bigger or smaller).“ As far as I understood, these thresholds were calculated on data for Greenland. Hence, the optimal thresholds for, e.g., Antarctica, might completely deviate from the ones calculated for

Greenland. This might hinder the global applicability of the pipeline (refer to major concerns 1 and 2). This should be added to the limitations.

We agree with the reviewer that these thresholds differ for Greenland and Antarctica glaciers. They are different among glaciers in Greenland. However, these thresholds are determined **automatically** based on the distribution of termini from each satellite and each glacier. Therefore, we believe it is feasible to apply the method globally. We have clarified this in the text. Line 242.

13. Line 235 „We then repeat this screening procedure ten times to maintain the quality of the terminus product": What screening procedure is meant here exactly now? All three or just the one with large areas? And does the outcome change when the screening procedure is done several times? If yes, please explain why.

The pipeline of the screening module is shown in Figure 3. For the first time, the inter-quartile range quantifies thresholds based on the distribution of terminus features such as length, and we remove the outlier traces. Such removal changes the distribution of terminus features, and we will have new thresholds for the second time and detect new outliers. We keep doing this ten times or until we don't find any more outliers. We have clarified this in the text. Line 254.

14. Line 245 „Traditional uncertainty quantification for glacier terminus position is conducted by calculating the difference between manually picked termini and automatically-picked termini." This is not uncertainty quantification but an error assessment (see major concerns 1).

We agree with the reviewer's comments and have revised the related sentence. See our response to major concern 1.

15. Line 262 „instead of quantifying the uncertainties of terminus traces, [Hartmann et al.] use the multiple inferences of MC dropout as extra information to retrain the network. " This is not quite correct. Hartmann et al. use the model uncertainty on one specific input as additional information for a second network with dropout. This second network then outputs several predictions again from which uncertainties could be calculated - but instead, to make it more robust, the predictions are averaged to eliminate this uncertainty.

We thank the reviewer's comments and rephrase the sentence as "*Hartmann et al. (2021) applied MC dropout to glacier terminus delineation and built a two-stage approach. They used the uncertainty of the first network as additional information to train the second network. The multiple outputs of the second network are averaged to eliminate the uncertainty and get the final prediction.*" Line 283.

16. Line 267 „To strike a balance between computational cost and the reliability of the MC dropout, we randomly chose ten images from all the sensors and make three inferences for each of them": It is not quite clear to me. Are ten images of each sensor taken? „in total each glacier will have two measures of uncertainty" – So, also ten images of each glacier?

Our original description is somewhat misleading. For each glacier, we will randomly select ten images for each sensor, and we will have 50 images in total. Using the ten images from the same sensor, we conduct MC dropout to quantify one uncertainty for that sensor. We have two measures of uncertainty, one is from duplicate traces, and the other is from MC dropout. The ten images from the same sensor are only for quantifying MC dropout uncertainty. We rephrase the text as "*To strike a balance between computational cost and the reliability of the MC dropout, we randomly chose ten images from each of the five sensors and made three inferences for each of the images. Thus, in total, each glacier will have six measures of uncertainty: one from duplicate traces and the other five estimated by MC dropout for each sensor.*" Line 289.

17. The results do, at some points, not validate the conclusions. No correlation was calculated (or it was not stated in the manuscript), and even a correlation would not necessarily induce causality. Please rephrase the conclusions to hypotheses.

  1. Line 307 „glaciers with less training data will have larger uncertainties and lower success rates"

     Rephrased as "*glaciers with less training data will probably have larger uncertainties and lower success rates*" Line 331.

  2. Line 309 „since they have the highest spatial resolution"

     Rephrased as "*Among the five datasets used, Landsat-8 and Sentinel-2 have the lowest average uncertainties, probably because they have the highest spatial resolution.*" Line 333.

  3. Line 310 onwards „The reasons for the Landsat-5 uncertainty are twofold [...]"

     Rephrased as: "*The reasons for the Landsat-5 uncertainty might be twofold.*" Line 335.

  4. Line 314 „The higher uncertainty of Sentinel-1 images is due to its low image quality, coarse resolution, and the lower volume of training data derived from this sensor."

     Rephrased as "*The higher uncertainty of Sentinel-1 images could be due to its low image quality, coarse resolution, and the lower volume of training data derived from this sensor.*" Line 338.

18. Line 325 „the uncertainty from duplicate traces is more representative of Landsat-7 and Sentinel-2 than other datasets" - Is it not only representative of these two datasets, as it was only calculated for these?

    For each glacier, we average the uncertainties from all duplicated traces and use the mean to represent the uncertainty of that glacier. Each glacier has one value of uncertainty from duplicate traces, and that value is more representative of Landsat-8 and Sentinel-2 as the value comes from these two satellites. We have added two sentences to make the description clear:

    Line 274: "*For each glacier, we average the uncertainties from all duplicated traces and use the mean to represent the uncertainty of that glacier.*"

    Line 291: "*Thus, in total, each glacier will have six measures of uncertainty: one from duplicate traces and the other five estimated by MC dropout for each sensor.*"

19. Line 308 „Among the five datasets used, Landsat-8 and Sentinel-2 have the lowest average uncertainties": Please give the exact numbers here. A table showing the different values for different data subsets would be good.

    The numbers are shown in Figure 8. The missing information is now added to the figure caption.

20. Line 355 „The metadata contains the date in YYYY-MM-DD, Glacier ID, source image satellite, and the uncertainty of each trace by averaging the two types of uncertainties provided": I thought the uncertainties were not available for every single trace, as they were only calculated for some of them due to computational limitations? Please clarify.

    Each glacier has six measures of uncertainty: one from duplicate traces and the other five estimated by MC dropout for each sensor. The uncertainty of each trace is estimated by averaging the uncertainty

from duplicate traces and the MC dropout uncertainty of its satellite sensor. We added one sentence to clarify in Line 291: "*Thus, in total, each glacier will have six measures of uncertainty: one from duplicate traces and the other five estimated by MC dropout for each sensor.*"

Line 423: „additional training data will be required to improve the data quality": or an improved network/pipeline.

Revised as suggested.

21. Line 434: „The pipeline can alert us of its failure based on the success rate within the screening module.": With your limitation that the screening module might not provide valid results for glaciers with few training examples, this alert might not trigger.

When most of the results are of good quality, the terminus features, such as length, will have a Gaussian-like distribution, where most of the terminus lengths are within the thresholds determined by the screening module, and the success rate will be high. For glaciers with few training examples, their results might be of poor quality. In that case, the distribution of the terminus features will be relatively scattered because many terminus lengths could be unreasonably long or short. As a result, the screening module will detect many traces as outliers, and these glaciers will have a low success rate, which can be used as an alert. We have clarified the text to point this out more readily: "*The network's failure will result in many termini not passing the screening. The pipeline can use the low success rates to alert us to prepare more training data for the corresponding glaciers.*" Line 464.

22. Please revise the color scheme of your figures, as red and green should not appear in the same plot (https://www.the-cryosphere.net/submission.html#figurestables).

Thank you for bringing this to our attention. We have modified the color schemes of Figure 6, Figure 8, and Figure 11, and checked figures through https://www.color-blindness.com/coblis-color-blindness-simulator/.

23. Figure 9: Is the number on the bottom left the average? Moreover, it would be good to state between which sensors the duplicates were calculated in the description of the figure.

Yes. We have added the missing information in the figure captions.

24. Figure S5: Visually, this does not appear to be a linear relationship. Have you done a correlation test?

Yes, the uncertainties from duplicate traces and MC dropout do not show a linear relationship. The differences in the two types of uncertainties are caused by their quantification methods and source images. We explained the details of the differences in the last paragraph of section 4.1, Data Quality.

**Specific Comments:**

1. Line 56 onwards: Heidler et al. 2022 (Deep Active Contour Models for Delineating Glacier Calving Fronts), Loebel et al. 2022 (Extracting glacier calving fronts by deep learning: the benefit of multi-spectral, topographic and textural input features), Gourmelon et al. 2022 (Calving fronts and where to find them: a benchmark dataset and methodology for automatic glacier calving front extraction from synthetic aperture radar imagery), and Davari et al. 2022 (Pixelwise Distance Regression for Glacier Calving Front Detection and Segmentation) are missing.

We thank the reviewer's comments, and all the missing citations are included.

2. Line 188: „Although TermPicks covers a range of conditions and brings great diversity to the training set, additional training data would improve the accuracy of the network in difficult situations." Please rephrase more cautiously (e.g., „... would presumably improve ..."), as you have no hard evidence that further training data would really improve the accuracy in this situation.

   The sentence is revised as suggested.

3. Line 205: GPU -> GPUs

   Revised as suggested.

4. Line 220 „With these three metrics, we calculate the lower (TL) and upper thresholds (TU) for each based on the inter-quartile range:" - The sentence structure is hard to follow. So, you compute the thresholds for each individual criterion?

   Yes. The sentence is rephrased as "*For each of the three metrics, we calculate the lower (TL) and upper thresholds (TU) based on the inter-quartile range:*"

5. Line 417 „120 GB of GPU memory": I guess you mean 120GB RAM? There are only a 40GB and an 80GB A100 version as far as I know, and 4 (=number of GPUs) times 40 GB is already 160 GB.

   I mean 120 GB of GPU memory. We have four A100 GPUs with a total memory of 160 GB, and we use 120 GB of memory. We didn't use all the GPU memory since we wanted the batch size to be a power of 2. When setting the batch size to 16, our network will need 120 GB of GPU memory.

6. Line 444: Remove the word „fully", as you still have some manual steps like defining the region of interest.

   The region of interest is only manually defined once at the beginning, and it won't interrupt the pipeline for continuous producing termini. Also, the region of interest will be **automatically** defined by the intersection between the terminus and the flowline in the future. So, we still think our pipeline is fully automated.

7. Table 1 includes abbreviations that were not introduced.

   The information in the last column is not necessary so it was removed.

8. Figure S2: Please name the conditions in the figure's description as well, referencing (a) to (e).

   The names of the conditions are included in the figure caption.

**Reference**

Baumhoer, C. A., Dietz, A. J., Kneisel, C., and Kuenzer, C.: Automated Extraction of Antarctic Glacier and Ice Shelf Fronts from Sentinel-1 Imagery Using Deep Learning, Remote Sensing, 11, 2529 – 22, https://doi.org/10.3390/rs11212529, 2019.

L. -C. Chen, G. Papandreou, I. Kokkinos, K. Murphy and A. L. Yuille, "DeepLab: Semantic Image Segmentation with Deep Convolutional Nets, Atrous Convolution, and Fully Connected CRFs," in IEEE Transactions on Pattern Analysis and Machine Intelligence, vol. 40, no. 4, pp. 834-848, 1 April 2018a, doi: 10.1109/TPAMI.2017.2699184.

Chen, L.-C., Zhu, Y., Papandreou, G., Schroff, F., and Adam, H.: Encoder-Decoder with Atrous Separable Convolution for Semantic Image Segmentation, in: Proceedings of the European Conference on Computer Vision (ECCV), 2018b.

Cheng, D., Hayes, W., Larour, E. Y., Mohajerani, Y., Wood, M. H., Velicogna, I., and Rignot, E. J.: Calving Front Machine (CALFIN): Glacial Termini Dataset and Automated Deep Learning Extraction Method for Greenland, 1972-2019, The Cryosphere, 2020, 1 – 17, https://doi.org/10.5194/tc-2020-231, 2020.

Gal, Y. and Ghahramani, Z.: Dropout as a Bayesian Approximation: Representing Model Uncertainty in Deep Learning, Proceedings of the 33rd International Conference on Machine Learning, 48, 2016.

Gourmelon, N., Seehaus, T., Braun, M., Maier, A., and Christlein, V.: Calving fronts and where to find them: a benchmark dataset and methodology for automatic glacier calving front extraction from synthetic aperture radar imagery, Earth System Science Data, 14, 4287–4313, https://doi.org/10.5194/essd-14-4287-2022, 2022.

Hartmann, A., Davari, A., Seehaus, T., Braun, M., Maier, A., and Christlein, V.: Bayesian U-Net for Segmenting Glaciers in Sar Imagery, 2021 IEEE International Geoscience and Remote Sensing Symposium IGARSS, 00, 3479–3482, https://doi.org/10.1109/igarss47720.2021.9554292, 2021

Kandel, I., & Castelli, M. (2020). The effect of batch size on the generalizability of the convolutional neural networks on a histopathology dataset. ICT express, 6(4), 312-315.

Mohajerani, Y., Wood, M. H., Velicogna, I., and Rignot, E. J.: Detection of Glacier Calving Margins with Convolutional Neural Networks: A Case Study, Remote Sensing, 11, 74 – 13, https://doi.org/10.3390/rs11010074, 2019.

Sun, C., Shrivastava, A., Singh, S., & Gupta, A. (2017). Revisiting unreasonable effectiveness of data in deep learning era. In Proceedings of the IEEE international conference on computer vision (pp. 843-852).

Xu M, Papageorgiou DP, Abidi SZ, Dao M, Zhao H, Karniadakis GE. A deep convolutional neural network for classification of red blood cells in sickle cell anemia. PLoS computational biology. 2017 Oct 19;13(10):e1005746.

Zhang, E., Liu, L., and Huang, L.: Automatically delineating the calving front of Jakobshavn Isbræ from multitemporal TerraSAR-X images: a deep learning approach, The Cryosphere, 13, 1729 – 1741, https://doi.org/10.5194/tc-13-1729-2019, 2019.

Zhang, E., Liu, L., Huang, L., and Ng, K. S.: An automated, generalized, deep-learning-based method for delineating the calving fronts of Greenland glaciers from multi-sensor remote sensing imagery, Remote Sensing of Environment, 254, 112 265, https://doi.org/10.1016/j.rse.2020.112265, 2021.

---

## Author Comment (AC2)

**General Comments:**

Presented in this manuscript is an automated data processing pipeline for extracting glacier termini positions, and the associated dataset that consists of data spanning 295 Greenlandic glaciers over period 1984-2021. The dataset consists of 278,239 glacier termini for 295 glaciers, and includes ice/ocean masks for the years 2018-2020. The pipeline consists of a Google Earth Engine based downloader , combined with a deep neural network to extract termini locations from the subsetted and preprocessed satellite imagery. The literature review covers most of the existing work in the field. The deep learning methodology also incorporates the greatest diversity of sensors (Landsat 5-8, Sentinel 1 & 2) and sensor types (both optical and SAR), which is a novel development. The methodology is quality controlled by assessing its performance on two uncertainty quantification metrics.

In summary, the study represents a significant contribution to the cryosphere and scientific community, by providing a new glacial termini dataset for Greenland, and an automated deep learning based pipeline for automated glacial feature extraction. However, there are certain comments to be addressed regarding the dataset and the manuscript before acceptance at the editor's discretion, as detailed below.

We greatly appreciate the detailed review and constructive comments by Reviewer 2. We have made our best effort to revise the manuscript based on the referee's comments and suggestions. In the following, we made an item-by-item response to the specific comments by the referee.

**Major Comments:**

- A primary concern to be noted is the lack of certain validation metrics that are commonly used in works such as this. Previous studies use the same established validation metrics (average area/distance between predicted and observed termini, or Mean distance error) to ensure ease of comparison. This measure is used in existing works such as Mohajerani et al. (2019), Baumhoer et al. (2019), Cheng et al. (2021), Heidler et al. (2021), Gourmelon et al. (2022), Loebel et al. (2022), and specifically Zhang et al. (2019, 2021). The average uncertainty of 37m, which is calculated using the average distance between duplicate picks from Landsat-8 and Sentinel-2, is somewhat misleading given this context, and the lack of such mean distance error calculation with respect to the ground truth should be addressed. Use of existing validation sets (Cheng et al. (2021), TermPicks/Goliber et al. (2022), and specifically Gourmelon et al. (2022)) would be advisable, as this would allow a fair comparison of this method with existing studies on established measures.

    We agree with the reviewer that using average uncertainty to compare with the measure of uncertainty defined in other related works is somewhat misleading. Following our response to comments from Reviewer 1, we have built a test dataset by randomly choosing 100 traces from TermPicks and used the rest of the TermPicks dataset to train the network from scratch. After training the network, we apply it to the test dataset and quantify the mean distance error between the network's predictions and the manual delineations. The test error of the network is 79 meters, which is now used to compare with others.

- A related concern to be noted is the biases inherent in the chosen validation metrics. One validation metric (average distance between duplicate picks from Landsat-8 and Sentinel-2) is biased towards lower/better values, since it is only calculated on higher resolution images, and doesn't measure the method's performance with respect to manual delineated observations that function as the ground truth. Furthermore, this uncertainty quantification cannot be calculated across the entire dataset, so its use as a metric to gauge the quality of the dataset is questionable.

    We only use duplicated Landsat-8 and Sentinel-2 since (i) duplicate Sentinel-1 traces are used for the georeferencing offset, and (ii) Landsat-5 or -7 lacks overlap with other datasets. We have clarified

this point in the text. We now build a test set to quantify the overall error by measuring the deviation between the network's predictions and manual delineations. Since this comment is similar to Major Concern 1 from Reviewer 1, we respond with the same comment as we did there: Quantifying error based on manual delineation involves a trade-off: the more representative the error is, the more manual effort it takes. Since we aim to produce as large a terminus dataset as possible (with a resulting 278,239 glacier termini), a highly representative error would require too much manual effort, which violates our primary objective to save manual effort. For this reason, we still keep the two automated ways to quantify the uncertainty of the terminus data. We agree that uncertainty and error are not the same.

- The data itself has a few issues that require reevaluation of the automated screening module. Within the provided dataset, there are fronts that are closed loops, make large spatio-temporal jumps, or are otherwise erroneous. Additionally, there is a non-negligible number of glaciers with termini that are cutoff by the boundaries of the ROI, which should be expanded and/or otherwise addressed.

  Without specific time/location identification of these issues, it is difficult to address this comment. Perhaps the reviewer is referring to Figure 3, which shows pre-screened results. Our aim with this figure is to demonstrate some of the issues that we built the screening module to detect. Regarding the ROIs, without a clear identification of the glaciers/times with these issues, it is hard to address this comment. Our ROIs are prepared manually at the beginning of the entire process, and we carefully choose the ROIs to make them cover the glacier termini over their entire image acquisition period.

- While the primary contributions of this study are the data processing pipeline and dataset, there is value in providing some analysis of the results, such as commenting on the general/regional area change trends (as shown for individual glaciers in the supplement, and to a degree in Figure 6), volume loss (when integrated with velocity datasets, though this may be out of scope), or correlations with temperatures/other measurements.

  The main objective of this study is to build a fully automated pipelined that can continuously produce terminus traces and generate a huge terminus trace dataset. We agree with the reviewer that scientific investigation is important and interesting. However, it is out of the scope of this study and will be accomplished in future works that leverage our data compilation.

- The integration of figures in the manuscript could be better handled. Specifically, few figures are referenced within the manuscript (6, 8, 9, and 10 being the exceptions).

  We merge Figure 3 and Figure 5 together as they both show the screening module. We believe that the rest of the figures in the manuscript have distinct and relevant purposes, and we have double-checked that they are all referenced in the manuscript.

- It would be in the best interests of the community for the TermPicks derived training data to be released for ease of use for future projects.

  Thank you for this suggestion. The reference polygons and label polygons converted from TermPicks traces have been included in the AutoTerm dataset now (10.5281/zenodo.7527485).

- The training & pre/postprocessing of the network can be elaborated upon. The learning rate/regularization factors are less important/useful than information such as the optimizer used, number of epochs trained on, the total number of images trained on, loss function used, vectorization algorithm, and data augmentations used (i.e., if no data augmentations were used, why not, and if so, what were they).

We add the missing information in Line 208: "*To train the network, we use binary cross entropy as the loss function and stochastic gradient descent method as the optimizer with an L2 regularization factor of $5\times10^{-4}$, as recommended by Zhang et al. (2021). Based on the learning rate in Chen et al. (2018b) and Zhang et al. (2021), we train the network with learning rates of $5\times10^{-3}$, $2\times10^{-3}$, and $1\times10^{-3}$, and choose $2\times10^{-3}$ owing to its lowest validation loss.*"

We adopt similar post-processing procedures with Zhang et al. (2019) that vectorize deep learning output to generate terminus traces. It is now added in Line 139.

The information about data augmentation can be found in Line 199.

**Specific Comments:**

**P2 L58:** I would recommend adding Gourmelon et al. (2022) and Loebel et al. (2022) to this list.

We thank the reviewer for pointing this out and have added these references to the reference list.

**P3 L70-71, P7 L210:** There are automated verification steps in Cheng et al. (2021), which includes filtering out unconfident predictions from the DL classifier.

We thank the reviewer for pointing that out. Cheng et al. (2021) has an automated data screening based on the deviations of two classifications of the network.

We have changed the related text as: "*Many previous DL methods applied to terminus delineation do not have quality control (Mohajerani et al., 2019; Zhang et al., 2019). Where it does exist, data screening has been simplistic and not automatically applied. For example, Zhang et al. (2021) only considers the complexity of the terminus shape and removes traces with abnormal complexity (which, in turn, requires a threshold to be established for each glacier), Baumhoer et al. (2019) only considers outliers that arise in a time series of terminus position change over time, and Gourmelon et al. (2022) remove the outliers based on terminus length. Cheng et al. (2021) however did design an automated data screening based on the deviation of two classifications from the network. Our screening module is based on using the physical properties of glacier termini.*" Line 224.

**P8 L225:** Could a detail/edge preserving speckle filter be applied? Or other types of Sentinel-1 processing steps to reduce speckle noise?

Considering the coarse resolution of the Sentinel-1 images, we did not apply the speckle filter to avoid blurry images. Also, glacier termini are still observable from the original Sentinel-1 images, even with speckle noise.

**P11 L341:** Is there a limitation (such as spatial coverage gaps) restricting ice mask generation to 2018-2020, or could they be made for other years?

They could be made for other years. For certain years, some glaciers might lack terminus traces, but there are no significant spatial coverage gaps in general. We only create updated masks annually beginning in 2018 to serve the ICESat-2 community needs for improved accuracy of laser returns during periods of extensive glacier terminus retreat. We have clarified this in Line 295.

**P21 Figure 1:** The flowchart is a not straight forward to follow. Perhaps consider separating the training/inference flowcharts, or organizing it in a more linear fashion.

The figure is composed of three parts: network training (black arrow), terminus inference (blue arrow), and longevity maintenance (red arrow). We chose a figure design to separate these procedures. Moreover, our figure

was designed to make good use of the figure space. Based on this comment, we have revised the figure to make the figure clearer by highlighting the training and inference.

**P26 Figure 6:** The color of the uncertainty bars and your results are the same (both are black). This makes the figure hard to interpret. Additionally, consider using colorblind friendly color schemes.

We have changed the color schemes and removed the uncertainty bars in the figure, as this figure is mainly for demonstrating the improved temporal resolution of our results. We also modified the color scheme of Figure 8 and Figure 11 and checked the figure through https://www.color-blindness.com/coblis-color-blindness-simulator/ following comments from Reviewer 1.

**P31 Figure 11:** Are the uncertainty bars for all of GID164's picks the same size?

Yes. The uncertainty bar is measured by using duplicate traces. Termini of the same glacier have the same uncertainty valued from duplicate traces. We added the description of the uncertainty bar in the figure caption. We have added two sentences to make the description clear:

Line 274: "*For each glacier, we average the uncertainties from all duplicated traces and use the mean to represent the uncertainty of that glacier.*"

Line 291: "*Thus, in total, each glacier will have six measures of uncertainty: one from duplicate traces and the other five estimated by MC dropout for each sensor.*"

**Reference**

Baumhoer, C. A., Dietz, A. J., Kneisel, C., and Kuenzer, C.: Automated Extraction of Antarctic Glacier and Ice Shelf Fronts from Sentinel-1 Imagery Using Deep Learning, Remote Sensing, 11, 2529 – 22, https://doi.org/10.3390/rs11212529, 2019.

Cheng, D., Hayes, W., Larour, E. Y., Mohajerani, Y., Wood, M. H., Velicogna, I., and Rignot, E. J.: Calving Front Machine (CALFIN): Glacial Termini Dataset and Automated Deep Learning Extraction Method for Greenland, 1972-2019, The Cryosphere, 2020, 1 – 17, https://doi.org/10.5194/tc-2020-231, 2020.

Gal, Y. and Ghahramani, Z.: Dropout as a Bayesian Approximation: Representing Model Uncertainty in Deep Learning, Proceedings of the 33rd International Conference on Machine Learning, 48, 2016.

Gourmelon, N., Seehaus, T., Braun, M., Maier, A., and Christlein, V.: Calving fronts and where to find them: a benchmark dataset and methodology for automatic glacier calving front extraction from synthetic aperture radar imagery, Earth System Science Data, 14, 4287–4313, https://doi.org/10.5194/essd-14-4287-2022, 2022.

Loebel, E., Scheinert, M., Horwath, M., Heidler, K., Christmann, J., Phan, L. D., Humbert, A., and Zhu, X. X.: Extracting Glacier Calving Fronts by Deep Learning: The Benefit of Multispectral, Topographic, and Textural Input Features, IEEE Transactions on Geoscience and Remote Sensing, 60, 1–12, https://doi.org/10.1109/TGRS.2022.3208454, 2022.

Mohajerani, Y., Wood, M. H., Velicogna, I., and Rignot, E. J.: Detection of Glacier Calving Margins with Convolutional Neural Networks: A Case Study, Remote Sensing, 11, 74 – 13, https://doi.org/10.3390/rs11010074, 2019.

Zhang, E., Liu, L., and Huang, L.: Automatically delineating the calving front of Jakobshavn Isbræ from multitemporal TerraSAR-X images: a deep learning approach, The Cryosphere, 13, 1729 – 1741, https://doi.org/10.5194/tc-13-1729-2019, 2019.

Zhang, E., Liu, L., Huang, L., and Ng, K. S.: An automated, generalized, deep-learning-based method for delineating the calving fronts of Greenland glaciers from multi-sensor remote sensing imagery, Remote Sensing of Environment, 254, 112 265, https://doi.org/10.1016/j.rse.2020.112265, 2021.

---

## Referee Report (RR1)

**Response to Reviewer 1's comments**
**General Comments**
This paper presents an automated pipeline in Google Earth Engine for glacier terminus tracing together with a so-derived dataset and updated ice/ocean masks. Such a pipeline is highly needed and of great significance to the community. This extent of automation has not been reached in related works. We thank the authors for their valuable contribution!

While this paper employs a sound deep learning architecture in combination with a promising screening module, I have several major concerns, including the technical correctness of the evaluation protocol and, thus, the validity of the proposed study, as the generalizability of the deep learning network still needs to be proven. Furthermore, comparisons to other studies need to be conducted in a technically correct way, and the reproducibility of the study needs to be ensured by making the assembled training dataset publicly available. Lastly, the structure of the manuscript should be improved upon.

We greatly appreciate the detailed and thorough review by Reviewer 1. We have made our best effort to revise the manuscript based on the referee's comments and suggestions. Below is an item-by-item response to the specific comments by this reviewer.

**Major Concern 1: Evaluation Protocol**
The pipeline has not been properly tested, and hence, we can not yet rely on its output. In my understanding, the authors seem to confuse uncertainty estimation with error assessment. In line 245, they call the calculation of the difference between prediction and ground truth „uncertainty quantification". The authors then claim that comparing to manually picked traces „requires significant manual effort" because it would have to be redone, as „network accuracy likely varies over time as glaciers experience different conditions". Instead, the authors use two different uncertainty quantifications that do not rely on ground truth data. Calculating uncertainties is definitely useful, and the two used ways of calculating the effect of different sources of uncertainty (model inherent and input inherent) look very promising. However, calculating the uncertainty is no substitute for an error assessment. The authors themselves state in line 395: „if both duplicated traces are deviated from reality but are close to each other, the uncertainty would not represent the reality." It is, therefore, indispensable to calculate the deviation of the network's predictions to manually delineated ground truth traces on a test set that is independent of the train set. First, we need to know how well the network is performing at the moment before we apply it to new unseen data and afterward assess whether the network's performance degrades when new sensors are used or other conditions change (called domain shift in machine learning).

We agree with the reviewer that the difference between predictions and ground truth should be called "error", while the difference between duplicate traces should be taken as "uncertainty". We have identified places in the manuscript where this terminology may have been confused and have updated the text. In addition, we have performed a test of the network as follows. We randomly choose 100 traces from TermPicks as a test dataset and use the rest of the TermPicks data to train the network from scratch. After training the network, we apply it to a test dataset and quantify the deviation of the network's prediction to manual delineations in the test dataset. This reveals a test error of 79 meters, which is similar to previous authors (Mohajerani et al., 2019; Zhang et al., 2019; Baumhoer et al., 2019; Cheng et al., 2020). The description of this test is added to the manuscript in Line 214 and Line 316.

Thank you for performing this test.
Please add more information in section 3.3 about the evaluation on the test set (train-test split – which images were picked for the test set exactly – this information ensures reproducibility; how exactly the error metric is calculated; etc.). Moreover, a split of the error on the test set between sensors would give additional valuable insights.

Quantifying error based on manual delineation involves a trade-off: the more representative the error is, the more manual effort it takes. Since we aim to produce as large a terminus dataset as possible (with a resulting 278,239 glacier termini), a highly representative error would require too much manual effort, which violates our primary objective to save manual effort. For this reason, we still keep the two automated ways to quantify the uncertainty of the terminus data. We agree that uncertainty and error are not the same.

I'm not sure that I am understanding the authors correctly, but in my understanding, this trade-off between the representativeness of the error and the manual effort is not correct. If the test set is small, it needs to be chosen with greater care such that the error will be representative, i.e., the test set should cover the possible variability of the data the network will see.

What I understand from the author's second sentence is: They trade off quality assessment for quantity. As the authors do provide not only the dataset but also advertise their pipeline for future use, the quality assessment needs to be thorough. Still, keeping the uncertainty quantification is a great bonus.

Although the reviewer states that we cannot rely on our model output, even without the model test we have now performed, we believe our data to be reliable for the following reasons. First, our terminus traces match the remote sensing images (Fig. 4). Second, the time series of terminus variation are in agreement with both Termpicks and CALFIN (Fig. 5). Third, the time series of terminus variations show a clear seasonal signal (refer to the time series data described in section 4.4), which would not be revealed if our terminus traces are unreliable.

Fig. 4 shows only six example traces, and in my regard, checking all 278,239 termini visually manually is also some manual effort, as even if the quality of each trace could be checked in one second, checking all traces would still require at least 10 days. I do not know how many and which images the authors checked, making the assessment not reproducible and subjective. Fig. 5, on the other hand, is a very nice analysis and indicates that there is probably no systemic error in the produced data. Still, this is just a rough hint at the quality and can not replace the test on a test set, which I would like to thank the authors for now providing.

Additionally, an experiment should be conducted to determine whether and by how much the error between prediction and ground truth on the test set is reduced when the screening module is applied versus not applied. In this way, the effectiveness of the screening module can be demonstrated. The same holds for the upsampling of small images (it is not sufficient to visualize the results of one sample, as shown in Fig. 13).

The screening module belongs in post-processing, and is thus not related to network inference or training. Instead, the screening module is for detecting outliers in order to improve data quality. Fig. 3 and the red crosses in Fig. 5 demonstrate the effectiveness of our screening module. For these reasons, we did not see it necessary to validate the effectiveness of the screening module on a test set.

The screening module is part of the complete pipeline which the authors propose. Hence, its effectiveness should, in my regard, be demonstrated in a thorough way as well. This can be done by once using the trained pipeline with and once without the module and reporting the difference in the error metric. Moreover, please also report the differences in the error metric for each sensor, as different sensors are handled differently by the screening module.

We agree with the reviewer that the test error is needed to demonstrate the effectiveness of upsampling as it is a pre-processing procedure. Thus, we have also conducted an upsampling test. For this test, we randomly select 36 images of five small glaciers to be a test set for size normalization. These images are not included in the training set for the independent evaluation of the size normalization's effectiveness. We add a new table to show the test error and uncertainty from duplicate traces with and without size normalization (Table S2). The results show that size normalization can effectively reduce test error and uncertainty. The related description is now added in Line 429.

Great work! Thank you! Just the formulation „36 images of five small glaciers that are beyond the training set as the test set" is hard to follow. Please rewrite as done above.

**Major Concern 2: Generalizability**
The pipeline has to be tested on out-of-sample data (i.e., glaciers not present in the training dataset) and data outside of Greenland to show generalizability to the global scope.
> 1. Line 451 „Owing to the transferability of deep learning, the entire pipeline has the potential to be applied to many other outlet glaciers around the world"
> 2. Line 135 „converting the TermPicks terminus data into a training dataset suitable for deep learning highly generalizes the network"

These claims have to be proven on such a test set. As most manually annotated traces available from related work are part of TermPicks and hence, have been used for training, another test set has to be used. For testing on SAR imagery, the dataset provided by Gourmelon et al. could, for example, be used, as it is not incorporated in TermPicks (except Jacobshaven, which probably has overlaps with TermPicks). However, test data for optical imagery might have to be created manually (e.g., from Antarctica or the Russian Arctic). At least, I am unaware of a dataset based on optical imagery that is not incorporated in TermPicks.

The importance of the size of training data in the deep learning field has been well demonstrated. For instance, Sun et al. (2017) showed that the network's performance increases logarithmically based on the volume of training data size. For this reason, we see no need to provide additional proof that our model has the ability to generalize.

The authors are correct that there is a relationship between dataset size and the performance of a network. This, however, does not mean that every network that is trained with a sufficiently large dataset will have adequate performance. This must be proven on a test set.

Further, our study focus is on Greenland alone – not on a global scale. We merely identify the potential for our model to be used at that scale. That said, out of interest we conducted an experiment in which we trained the network with only 1466 training examples prepared manually without including TermPicks. The test error of this network is 315 meters, which is much larger than 79 meters that we have after training with TermPicks. Such an improvement demonstrates the generalization improvements brought by TermPicks. The related description is added in Line 405.

It seems I have misunderstood what the authors meant by „generalizability". I assumed the meaning was how well the model would predict the termini of images from the target distribution (i.e., all glaciers with termini around the world). However, it seems the authors merely meant how well the model would predict the termini of images from the distribution of data used to train the model (i.e., glaciers that are included in the train set but from future time points given that no other variables significantly change).
I apologize for the confusion. With the evaluation on the test set, the generalizability is proven, and the additional test with TermPicks also nicely proves the sentence from line 135.
However, it has to be stated more clearly (and also in the abstract) that the pipeline is only tested for glaciers in Greenland and not on a global scale and, therefore, can only be applied to Greenland without further precautions!

**Major Concern 3: Comparability**
It is not possible to compare the calculated uncertainties of this manuscript to the errors calculated in related works, as done in, e.g., line 304 or line 379. Two totally different metrics are compared here, and studies have been conducted on different datasets. For a valid comparison, the exact same network/pipeline needs to be tested on different datasets, or different networks/pipelines have to be trained, optimized, and tested on the exact same data (a so-called benchmark dataset). Altering both the dataset and the network/pipeline introduces too many changes, and a changed performance could result from either the different dataset (for example, the test set might be easier, and therefore, the performance of an otherwise worse performing network would be better on this test set) or the different network/pipeline.
Concludingly, the claimed improvements 1 and 2 (line 377 „1 increasing the generalization level of the deep learning network to enable more and better quality terminus predictions; 2 deploying size normalization to improve the accuracy of terminus delineation for small glaciers") are not proven.
One way to show the superior terminus prediction performance on SAR imagery could be the use of the benchmark dataset recently proposed by Gourmelon et al. (2022) (i.e., retraining the pipeline on the train set and evaluating it on the test set using the stated metrics). To the best of our knowledge, there is no equivalent benchmark dataset for optical imagery.

Now that we have performed an error estimate we can more accurately compare our network error to that from other studies (Line 315), and find that our network's performance is comparable with other studies, not superior or inferior.

Please ensure that this is reflected correctly everywhere in the manuscript (e.g., lines 403-407 suggest that your performance is superior).

We agree with the review that different networks should be trained and tested on the same benchmark dataset to determine the best. However, such a test is outside of the scope of this study. The objective of this study is not to demonstrate which deep learning network is better but to generate a terminus dataset with spatial coverage and temporal resolution that no prior study has with the addition of further automation in the production pipeline.

For generalization and size normalization, please refer to the response of Major Concern 3 and Major Concern 1, respectively.

With the evaluation on the test set, this is addressed satisfactorily.
I understand that a rigorously correct comparison is outside the scope of this study.

**Major Concern 4: Reproducibility**
Please make your complete assembled training data (including the satellite imagery) publicly available, as only in this way the reproducibility of the results is guaranteed. Moreover, please also provide the manually created reference polygons for each glacier.

All our remote sensing images are freely available on Google Earth Engine. The code for collecting the data is also available on GitHub. TermPicks is also a publicly available dataset. The reference polygons and label polygons converted from TermPicks traces have been included in the AutoTerm dataset now (10.5281/zenodo.7527485).

Thank you for addressing this issue. Recreating a dataset might have produced some slightly different data, which might again have influenced the training of a network.

**Major Concern 5: Structure of the manuscript**
The structure of the manuscript has to be improved upon. There is a mix-up between the training and inference of the pipeline, and some information is given twice at different positions in the manuscript. It is hard to tell when the authors write about the newly derived dataset in contrast to the dataset derived from TermPicks for training the network, e.g., in line 295 (,,We find an average success rate of 64%"), it is unclear on which dataset the success rate was calculated. I would suggest splitting the manuscript into two main parts as follows, but there could also be another better split-up:
> 1. Training Pipeline: manual delineated dataset creation (TermPicks + additional manual annotations), neural network (architecture), network training (train-validation-test split, learning rate, number of trained epochs, etc.), screening module, error calculation, uncertainty estimation
> 2. Inference Pipeline: new data acquisition + pre-processing, uncertainty estimation on this newly derived dataset, ice/ocean mask updates

We thank the reviewer for these comments. We decided to structure the manuscript following the order of data processing. We first collect remote sensing images and conduct preprocessing (section 3.1). Second, we generate the training dataset by converting the terminus traces in TermPicks into label polygons and pairing polygons with the remote sensing images. Third, we introduce the network structure and the training progress in section 3.3. Sections 3.4 to 3.7 are post-processing procedures after applying the well-trained network to make inferences on the 433,721 images collected via Google Earth Engine. That said, we notice that the original section titles may have caused some confusion. Thus, to address this comment we changed the title of Section 3.2 from "Generalizing the network" to "Generating training data from TermPicks", and the title of Section 3.3 from "Deep Learning Network" to "The Structure and Training of Deep Learning Network".
Furthermore, we have revised some of the text to make it clearer that we first perform network training and then we apply the well-trained network to all the images collected through Google Earth Engine to generate terminus positions. This is done in the last sentence in Section 3.3 as "*After the training, we apply the well-trained network to the test set for quantifying the test error and to all the images collected via GEE for generating the terminus dataset.*" Line 217.
The success rate is the percentage of terminus traces that pass the screen module. We add one sentence at the end of section 3.4: "*Finally, we estimate the success rate by calculating the percentage of the terminus traces that pass the screening module.*" Line 255.

Thank you. I think adding a paragraph like ,,The rest of the paper is structured as follows ..." (as is done above) to the end of the introduction would aid the reader.
Please also provide the success rate for the test set.

In section 4.1, the authors first introduce the ,success rate', which should, however, be introduced in the methods section.

We add one sentence at the end of section 3.4: "*Finally, we estimate the success rate by calculating the percentage of the terminus traces that pass the screening module.*" Line 255.

Thank you.

Moreover, paragraph line 188 to 195 should be moved to limitations.

Line 194 to 201 describe the limitations of the input imagery, not the limitations of our work. These points are discussed here because they explain why we needed to prepare additional training data. Therefore, we think this text fits better in Section 3.2, which describes the training data preparation.

I understand. Does your pipeline still have difficulties with these kinds of images? If so, please additionally add a few sentences about it in the limitations.

An explanation of the two uncertainty measures, as given in lines 319 to 323, should be moved to further at the beginning of the manuscript.

The principle of the two uncertainty measures is described fully in Methods (Section 3.6). Lines 344 to 355 explains the uncertainty results and focus on why these two uncertainties are different. Therefore, we believe lines 344 to 355 fit better in the Results section.

I understand. Thank you.

**Major Comments:**
1. It is unclear to me whether the name „AutoTerm" refers to the automated pipeline or the derived dataset, or both.
We thank the reviewer for pointing this out. We think it is a good idea to use AutoTerm reflect both the data and pipeline. We have changed the title of the manuscript to: "AutoTerm: an automated pipeline for glacier terminus extraction using machine learning and a "big data" repository of Greenland glacier termini."
Yes, thank you.

2. The title of the manuscript does not mention the automated pipeline, which is, in my humble regard, the most significant contribution. Hence, I'd argue for a more suitable title, e.g. AutoTerm: an automated Google Earth Engine pipeline for glacier terminus extraction and „big data" repository of Greenland glacier termini
We thank the reviewer's suggestion. We have changed the title of the manuscript to: "AutoTerm: an automated pipeline for glacier terminus extraction using machine learning and a "big data" repository of Greenland glacier termini."
Thank you.

3. It needs to be clarified whether the region of interest that has to be defined for each new glacier has to be a polygon like in figure 2 or whether it can simply be a bounding box.
Each region of interest is a bounding box. The polygon in Figure 2 is a reference polygon only for converting TermPicks traces into a training label polygon, which is described in the Figure caption. We have updated the text where needed to clarify this point. Line 146.
Thank you.

4. Line 163 onwards: „This allows glaciers with various natural sizes to have a similar image size in computer vision, which largely decreases the complexity of delineating glacier terminus." This statement (the second part of it) needs more explanation or a reference.
We rephrased the text here and added a reference: "*We then normalize the image size, which is commonly adopted in the computer vision field for better capturing object features with various physical sizes (Xu et al., 2017).*" Line 167.
Thank you.

5. The normalization of image sizes is not clear to me. Small images are upsampled, but large images are

not downsampled. Hence, do they still have different sizes? I would not call this normalization, then. Moreover, the authors extract patches afterward, so the input size is always equal anyways. Additionally, only showing one figure that shows an improvement for one trace is not sufficient evidence that this upsampling generally improves the delineation performance. Please show the improvement in numbers over a complete, independent test set (refer to major concerns 1 and 2).

Different glaciers are not of the same physical size and therefore don't have similar number of pixels that fall within their fjord walls. This can vary from 300 pixels to 3000 pixels. As a result, the deep learning input (image patch) may sometimes cover more than the entire glacier and may alternatively sometimes only cover a part of a glacier.
After normalization, all small glaciers will be covered by images with a width of about 1000 pixels, regardless of their original image width. In other words, the normalization makes glaciers appear to the deep learning network as if they had a similar physical size. We have attempted to clarify this in the text. Line 168.
Thank you, I understand now that you meant the physical size. So, what is the mean physical size of one pixel now? Moreover, how is upsampling performed? Linear interpolation or a nearest neighbor interpolation?

We agree with the reviewer that test error is needed to demonstrate the effectiveness of upsampling as it is a pre-processing procedure. We randomly select 36 images of five small glaciers that are beyond the training set as the test set for size normalization. We add a new table to show the test error and uncertainty from duplicate traces with and without size normalization (Table S2). The results show that size normalization can effectively reduce test error and uncertainty. The related description is now added in Line 428.
Thank you.

6. Section 3.3:

    1. „encoder-decoder structure [...] can obtain sharp object boundaries": Actually, an encoder decoder structure without skip connections would most probably not recover any details and, therefore, no sharp object boundaries. In Chen et al. (2018), they use a more sophisticated method to obtain the sharp boundaries: „A fully connected CRF [conditional random field] is then applied to refine the segmentation result and better capture the object boundaries."
    We realize that there is a mistake in our citation. The original DeepLab use the CRF to refine the segmentation results (Chen et al., 2018a, DeepLab: Semantic Image Segmentation with Deep Convolutional Nets, Atrous Convolution and Fully Connected CRFs). Later on, the same author improved the DeepLab and developed DeepLabV3+ (Chen et al., 2018b; Encoder-Decoder with Atrous Separable Convolution for Semantic Image Segmentation), which is the network we use in this manuscript. The DeepLabV3+ adds a simple yet effective decoder module to refine the segmentation results especially along object boundaries (Chen et al., 2018b).
    I see, thank you for correcting your reference.

    2. „atrous convolution [...] senses multi-scale contextual information": It is not the atrous convolutions alone that make recognition of multi-scale contextual information possible, but the combination of atrous convolutions in ASPP. "Atrous convolution allows us to explicitly control the resolution at which feature responses are computed within Deep Convolutional Neural Networks. It also allows us to effectively enlarge the field of view of filters to incorporate larger context without increasing the number of parameters or the amount of computation. Second, we propose atrous spatial pyramid pooling (ASPP) to robustly segment objects at multiple scales." (Chen et al., 2018)
    We thank the reviewer's comments and changed "atrous convolution" to "atrous spatial pyramid pooling".
    Thank you.

    3. „multi-scale contextual information [...] [is] helpful for our task since [...] we integrate remote sensing datasets with different spatial resolutions": Multi-scale refers to how many pixels a neuron is able to see (effective receptive field) and not how much square meters one pixel can see. Hence, multi-scale contextual information helps when the calving front covers many versus only a few pixels. Thus, it helps only indirectly with different spatial resolutions of the dataset.
    We agree with the reviewer's comment. Multi-scale refers to the network's ability to sense various portions of the images but not sense images with different resolutions. We rephrase the sentence as

"*Sharp boundaries can improve delineation accuracy, and sensing multi-scale information helps indirectly when we integrate remote sensing datasets with different spatial resolutions.*" Line 205
Thank you.

4. „This network has been proven to have large learning capability, spatial transferability [...]": These are quite big claims based on a train set of two glaciers and a test set of one glacier that are all located in Greenland (Zhang et al., 2021).
The large learning capability has been demonstrated by the paper of DeepLabV3+ (Chen et al., 2018b). Zhang et al. (2021) applied the network to a glacier beyond the training set, showing the network's spatial transferability. In this work, we apply the network to 295 glaciers in Greenland and generate 278,239 glacier termini, of which 17,906 terminus traces are from the training set. This means that 94% of our results are beyond the training set, which further demonstrates the learning capability and spatial transferability of the network.
As this sentence only references Zhang et al. (2021), the authors are not reporting on the present study. Please call it „network architecture" and not „network" to make clear what you are referring to and add the reference to Chen et al., 2018b as well.

5. „The network is trained with a learning rate of 0.005 [...] as recommended by (Zhang et al., 2021)": The optimal learning rate for training is highly dependent on the dataset as well as on the batch size (not just the model). Hence, the learning rate has to be treated as a hyperparameter, which has to be optimized on a validation set (not the test set). A sub-optimal learning rate can lead to significantly longer training times until convergence or no convergence at all.
We agree with the reviewer and have tried to train the network with learning rates of $5\times10_{-3}$, $2\times10_{-3}$, and $1\times10_{-3}$. The validation losses for these three learning rates are 0.023, 0.020, and 0.024, respectively. Overall, the validation losses are measured using binary cross entropy and are similar to each other. We chose the learning rate ($2\times10_{-3}$) with the lowest validation error. We rephrased the sentence as: "*Based on the learning rate in Chen et al. (2018b) and Zhang et al. (2021), we train the network with learning rates of $5\times10_{-3}$, $2\times10_{-3}$, and $1\times10_{-3}$, and choose $2\times10_{-3}$ owing to its lowest validation loss.*" Line 210.
Thank you very much.

6. „we choose the largest batch size (16)": This should be „largest possible batch size (16) on an A100 GPU with 40/80 GB GPU memory". Please specify whether your A100s have 40 or 80 GB GPU memory.
The sentence is rephrased as follows: "*we choose the largest possible batch size (16) on four A100 GPUs with 160 GB GPU memory in total. We set the batch size to a power of two to take full advantage of GPU processing (Kandel and Castelli, 2020)*" Line 212.
Thank you very much.

7. What exactly is meant by "maximize our computational power" in line 204?
It means using as much GPU memory as possible. The sentence is now rephrased as "*we choose the largest possible batch size (16) on four A100 GPUs with 160 GB GPU memory in total*." Line 212.
Thank you.

8. „The network training takes about a week": This is quite long and might be due to a sub-optimal learning rate. Please specify not only the training time but also the number of trained iterations over the complete augmented dataset. Also, specify your train-test-split and your metrics for evaluation (refer to major concern 1). Moreover, did you use an early stopping criterion? You might have overfitted during this long training time.
The long training time is caused by the large training dataset. We have 17906 training examples, and the augmentation increases the training set by a factor of four. The training takes more than 600,000 iterations. Among all the training data, we select 5% of images randomly as validation datasets to conduct early stopping. The training will be stopped when the validation error stops decreasing for 3 consecutive epochs. We add these details in Line 214.
Thank you very much. I assume by iteration, inputting one batch into the network and backpropagating its loss to update the weights is meant? Please instead provide the times the network has seen the train set („iterations over the train set"), which is normally defined as the number of epochs. If my assumption is correct, that would be (600,000 * 16) / (17906 * 4).

Out of curiosity, is the checkpoint saved after each batch?

7. Line 210 „do not have any quality control": At least Cheng et al. have manual control. So, maybe rephrase it to „do not have any automated quality control".
We thank the reviewer for pointing that out. Cheng et al. (2021) have an automated data screening based on the deviations of two classifications of the network.
We have changed the related text as: "*Many previous DL methods applied to terminus delineation do not have quality control (Mohajerani et al., 2019; Zhang et al., 2019). Where it does exist, data screening has been simplistic and not automatically applied. For example, Zhang et al. (2021) only considers the complexity of the terminus shape and removes traces with abnormal complexity (which, in turn, requires a threshold to be established for each glacier), Baumhoer et al. (2019) only considers outliers that arise in a time series of terminus position change over time, and Gourmelon et al. (2022) remove the outliers based on terminus length. Cheng et al. (2021) however did design an automated data screening based on the deviation of two classifications from the network. Our screening module is based on using the physical properties of glacier termini.*"
Line 224.
Thank you.

8. Line 215 onwards: Please mention that the screening builds on top of existing works here (Zhang et al. 2021 – terminus curvature screening, Baumhoer et al. 2019 – time series outliers, Gourmelon et al. 2022 – removal of too short termini predictions), but goes one step further, i.e., doesn't use any manual intervention or prior knowledge of the data.
The related sentences are revised as follows: "*Based on the previous works (Baumhoer et al. 2019 , Zhang et al. 2021, Gourmelon et al. 2022), we develop an automated screening module that forgoes any manual intervention or prior knowledge of the data.*" Line 232.
Thank you.

9. Line 217 „Terminus length is determined by the sum of the piece-wise length along an individual terminus trace". Please explain in more detail. This, at least for me, is hard to understand.
Each terminus trace is an ordered set of points. The length is the sum of the length between the two closest points. The description refers to how we calculate the terminus length, which might be confusing to readers. Terminus length is just the physical length of the glacier terminus. As a result of this confusion, we decided to remove this sentence.
Thank you.

10. Line 218 „Terminus curvature is computed between two adjacent points for each point along the terminus and then an average is taken for each terminus trace." This is also not completely clear to me. I think an equation would help.
We have rephrased the related text as: "*Terminus curvature is computed among every three adjacent points along the terminus based on Peijin Zhang's work (https://github.com/peijin94/PJCurvature), and then an average is taken for each terminus traces.*" Line 235.
Thank you.

11. Line 224 „percentile of the data range": Do you refer to the data range of the generated training data? Is this computed per glacier? Per satellite? The validity of these thresholds needs to be checked on an independent test set (refer to major concerns 1 and 2).
For each glacier, we will calculate the thresholds based on the termini from the same satellite. For instance, we will calculate thresholds for Sentinel-2 traces of GID2 and Landsat-8 traces of GID2 separately. We add one sentence in Line 242 to provide additional clarification: "*The thresholds are calculated automatically based on the results of the same glacier and same satellite.*"
Thank you. It is also now clear that you calculate these metrics on outputs of the network. So the screening can only be done if the pipeline is applied to a time series of images? Or do you store a rolling average of each glacier and each satellite in your pipeline?

The screening module belongs to the post-processing, which is not related to network inference or training. The screening module is for detecting outliers and maintaining the data quality. Fig. 3 and

the red crosses in Fig. 5 demonstrate the effectiveness of our screening module. So, we did not validate the effectiveness of the screening module on a test set.
Fig. 3 and Fig. 5 are insufficient proof of the effectiveness. The screening module is part of the complete pipeline which the authors propose. Hence, its effectiveness should, in my regard, be demonstrated in a thorough way as well.

12. Line 227 „For outliers in terminus length, we remove both the lower and upper thresholds (Eqns. 1 and 2) because we do not anticipate large changes in terminus length in either direction (bigger or smaller)." As far as I understood, these thresholds were calculated on data for Greenland. Hence, the optimal thresholds for, e.g., Antarctica, might completely deviate from the ones calculated for Greenland. This might hinder the global applicability of the pipeline (refer to major concerns 1 and 2). This should be added to the limitations.
We agree with the reviewer that these thresholds differ for Greenland and Antarctica glaciers. They are different among glaciers in Greenland. However, these thresholds are determined **automatically** based on the distribution of termini from each satellite and each glacier. Therefore, we believe it is feasible to apply the method globally. We have clarified this in the text. Line 242.
I understand now. However, please indicate that the screening has not been tested on glaciers outside of Greenland.

13. Line 235 „We then repeat this screening procedure ten times to maintain the quality of the terminus product": What screening procedure is meant here exactly now? All three or just the one with large areas? And does the outcome change when the screening procedure is done several times? If yes, please explain why.
The pipeline of the screening module is shown in Figure 3. For the first time, the inter-quartile range quantifies thresholds based on the distribution of terminus features such as length, and we remove the outlier traces. Such removal changes the distribution of terminus features, and we will have new thresholds for the second time and detect new outliers. We keep doing this ten times or until we don't find any more outliers. We have clarified this in the text. Line 254.
Thank you.

14. Line 245 „Traditional uncertainty quantification for glacier terminus position is conducted by calculating the difference between manually picked termini and automatically-picked termini." This is not uncertainty quantification but an error assessment (see major concerns 1).
We agree with the reviewer's comments and have revised the related sentence. See our response to major concern 1.
Thank you.

15. Line 262 „instead of quantifying the uncertainties of terminus traces, [Hartmann et al.] use the multiple inferences of MC dropout as extra information to retrain the network. " This is not quite correct. Hartmann et al. use the model uncertainty on one specific input as additional information for a second network with dropout. This second network then outputs several predictions again from which uncertainties could be calculated - but instead, to make it more robust, the predictions are averaged to eliminate this uncertainty.
We thank the reviewer's comments and rephrase the sentence as "*Hartmann et al. (2021) applied MC dropout to glacier terminus delineation and built a two-stage approach. They used the uncertainty of the first network as additional information to train the second network. The multiple outputs of the second network are averaged to eliminate the uncertainty and get the final prediction.*" Line 283.
Thank you.

16. Line 267 „To strike a balance between computational cost and the reliability of the MC dropout, we randomly chose ten images from all the sensors and make three inferences for each of them": It is not quite clear to me. Are ten images of each sensor taken? „in total each glacier will have two measures of uncertainty" – So, also ten images of each glacier?
Our original description is somewhat misleading. For each glacier, we will randomly select ten images for each sensor, and we will have 50 images in total. Using the ten images from the same sensor, we conduct MC dropout to quantify one uncertainty for that sensor. We have two measures of uncertainty, one is from duplicate traces, and the other is from MC dropout. The ten images from the same sensor are only for quantifying MC dropout uncertainty. We rephrase the text as "*To

*strike a balance between computational cost and the reliability of the MC dropout, we randomly chose ten images from each of the five sensors and made three inferences for each of the images. Thus, in total, each glacier will have six measures of uncertainty: one from duplicate traces and the other five estimated by MC dropout for each sensor.*" Line 289.
Thank you.

17. The results do, at some points, not validate the conclusions. No correlation was calculated (or it was not stated in the manuscript), and even a correlation would not necessarily induce causality. Please rephrase the conclusions to hypotheses.

> 1. Line 307 „glaciers with less training data will have larger uncertainties and lower success rates"
> Rephrased as "*glaciers with less training data will probably have larger uncertainties and lower success rates*" Line 331.
> 2. Line 309 „since they have the highest spatial resolution"
> Rephrased as "*Among the five datasets used, Landsat-8 and Sentinel-2 have the lowest average uncertainties, probably because they have the highest spatial resolution.*" Line 333.
> 3. Line 310 onwards „The reasons for the Landsat-5 uncertainty are twofold [...]"
> Rephrased as: "*The reasons for the Landsat-5 uncertainty might be twofold.*" Line 335.
> 4. Line 314 „The higher uncertainty of Sentinel-1 images is due to its low image quality, coarse
> resolution, and the lower volume of training data derived from this sensor."
> Rephrased as "*The higher uncertainty of Sentinel-1 images could be due to its low image quality, coarse resolution, and the lower volume of training data derived from this sensor.*" Line 338.
>
> Thank you for addressing my points here. Please also go over the complete manuscript to search for further such claims (e.g., line 327 „However, the network does struggle to delineate termini in many wintertime Sentinel-1 images because of blurry boundaries and the lack of sufficient training data specifically using Sentinel-1 imagery"). There is some mix-up of results and discussion (e.g., line 325 „Such variations are largely caused by the uneven distribution of the training data—glaciers with more training data have higher success rates." Here again, causality is not proven. The authors **observe** that glaciers with more training data have higher success rates, and **conclude** that the uneven distribution of training data might cause this.) which should be addressed. Please separate plain results and conclusions into two sections (i.e., the conclusions should be moved from the results section to the discussion section).

18. Line 325 „the uncertainty from duplicate traces is more representative of Landsat-7 and Sentinel-2 than other datasets" - Is it not only representative of these two datasets, as it was only calculated for these?
For each glacier, we average the uncertainties from all duplicated traces and use the mean to represent the uncertainty of that glacier. Each glacier has one value of uncertainty from duplicate traces, and that value is more representative of Landsat-8 and Sentinel-2 as the value comes from these two satellites. We have added two sentences to make the description clear:
Line 274: "*For each glacier, we average the uncertainties from all duplicated traces and use the mean to represent the uncertainty of that glacier.*"
Line 291: "*Thus, in total, each glacier will have six measures of uncertainty: one from duplicate traces and the other five estimated by MC dropout for each sensor.*"
Still, my question stands: Is the uncertainty calculated with duplicate traces (and just this uncertainty, not the MC dropout one) not only representative of these two sensors, as it was only calculated for these? (Line 355)

19. Line 308 „Among the five datasets used, Landsat-8 and Sentinel-2 have the lowest average uncertainties": Please give the exact numbers here. A table showing the different values for different data subsets would be good.
The numbers are shown in Figure 8. The missing information is now added to the figure caption.
Thank you.

20. Line 355 „The metadata contains the date in YYYY-MM-DD, Glacier ID, source image satellite, and the uncertainty of each trace by averaging the two types of uncertainties provided“: I thought the uncertainties were not available for every single trace, as they were only calculated for some of them due to computational limitations? Please clarify.

Each glacier has six measures of uncertainty: one from duplicate traces and the other five estimated by MC dropout for each sensor. The uncertainty of each trace is estimated by averaging the uncertainty from duplicate traces and the MC dropout uncertainty of its satellite sensor. We added one sentence to clarify in Line 291: "*Thus, in total, each glacier will have six measures of uncertainty: one from duplicate traces and the other five estimated by MC dropout for each sensor.*"

Please additionally indicate in line 386 that the average is taken as the uncertainty measure for each of the glacier's traces.

Moreover, an additional question arises: In lines 293-296, the authors describe the procedure for how the dataset is generated. But how is it configured in the automatic pipeline if, say, it is used for glaciers in Antarctica? Will MC dropout uncertainty be calculated for each image, or how are images randomly chosen here?

Line 423: „additional training data will be required to improve the data quality“: or an improved network/pipeline.

Revised as suggested.

Thank you.

21. Line 434: „The pipeline can alert us of its failure based on the success rate within the screening module.“: With your limitation that the screening module might not provide valid results for glaciers with few training examples, this alert might not trigger.

When most of the results are of good quality, the terminus features, such as length, will have a Gaussianlike distribution, where most of the terminus lengths are within the thresholds determined by the screening module, and the success rate will be high. For glaciers with few training examples, their results might be of poor quality. In that case, the distribution of the terminus features will be relatively scattered because many terminus lengths could be unreasonably long or short. As a result, the screening module will detect many traces as outliers, and these glaciers will have a low success rate, which can be used as an alert. We have clarified the text to point this out more readily: " *The network's failure will result in many termini not passing the screening. The pipeline can use the low success rates to alert us to prepare more training data for the corresponding glaciers.*" Line 464.

As the thresholds of the screening module are calculated for each glacier individually based on the network's predictions, the screening module will not filter out wrong predictions of a glacier where only wrong predictions ever occur (e.g., due to very few training examples). I assume all three variables (terminus length, curvature, and enclosed area) would roughly be uniformly distributed (as the predictions would be somewhat random), which would lead to a smaller Q1 and a bigger Q3 and, therefore, to a lower $T\_L$ and a higher $T\_U$. This, in turn, reduces the amount of filtered-out predictions. Please correct me if I have an error in my logic.

22. Please revise the color scheme of your figures, as red and green should not appear in the same plot (https://www.the-cryosphere.net/submission.html#figurestables).

Thank you for bringing this to our attention. We have modified the color schemes of Figure 6, Figure 8, and Figure 11, and checked figures through https://www.color-blindness.com/coblis-color-blindnesssimulator/.

Please also do the same for the remaining figures (5, 7, 10).

23. Figure 9: Is the number on the bottom left the average? Moreover, it would be good to state between which sensors the duplicates were calculated in the description of the figure.

Yes. We have added the missing information in the figure captions.

Thank you.

24. Figure S5: Visually, this does not appear to be a linear relationship. Have you done a correlation test?

Yes, the uncertainties from duplicate traces and MC dropout do not show a linear relationship. The differences in the two types of uncertainties are caused by their quantification methods and source

images. We explained the details of the differences in the last paragraph of section 4.1, Data Quality.
I don't understand. Why does the caption then claim that there is a linear relationship?

**Specific Comments:**
1. Line 56 onwards: Heidler et al. 2022 (Deep Active Contour Models for Delineating Glacier Calving Fronts), Loebel et al. 2022 (Extracting glacier calving fronts by deep learning: the benefit of multispectral, topographic and textural input features), Gourmelon et al. 2022 (Calving fronts and where to find them: a benchmark dataset and methodology for automatic glacier calving front extraction from synthetic aperture radar imagery), and Davari et al. 2022 (Pixelwise Distance Regression for Glacier Calving Front Detection and Segmentation) are missing.
We thank the reviewer's comments, and all the missing citations are included.
Thank you.

2. Line 188: „Although TermPicks covers a range of conditions and brings great diversity to the training set, additional training data would improve the accuracy of the network in difficult situations." Please rephrase more cautiously (e.g., „... would presumably improve ..."), as you have no hard evidence that further training data would really improve the accuracy in this situation.
The sentence is revised as suggested.
Thank you.

3. Line 205: GPU -> GPUs
Revised as suggested.
Thank you.

4. Line 220 „With these three metrics, we calculate the lower (TL) and upper thresholds (TU) for each based on the inter-quartile range:" - The sentence structure is hard to follow. So, you compute the thresholds for each individual criterion?
Yes. The sentence is rephrased as "*For each of the three metrics, we calculate the lower (TL) and upper thresholds (TU) based on the inter-quartile range*:"
Thank you.

5. Line 417 „120 GB of GPU memory": I guess you mean 120GB RAM? There are only a 40GB and an 80GB A100 version as far as I know, and 4 (=number of GPUs) times 40 GB is already 160 GB.
I mean 120 GB of GPU memory. We have four A100 GPUs with a total memory of 160 GB, and we use 120 GB of memory. We didn't use all the GPU memory since we wanted the batch size to be a power of 2. When setting the batch size to 16, our network will need 120 GB of GPU memory.
I see, thank you.

6. Line 444: Remove the word „fully", as you still have some manual steps like defining the region of interest.
The region of interest is only manually defined once at the beginning, and it won't interrupt the pipeline for continuous producing termini. Also, the region of interest will be **automatically** defined by the intersection between the terminus and the flowline in the future. So, we still think our pipeline is fully automated.
OK.

7. Table 1 includes abbreviations that were not introduced.
The information in the last column is not necessary so it was removed.
Ok, thank you.

8. Figure S2: Please name the conditions in the figure's description as well, referencing (a) to (e).
The names of the conditions are included in the figure caption.

Thank you.

**Response to Reviewer 2's comments**

**General Comments:**

Presented in this manuscript is an automated data processing pipeline for extracting glacier termini positions, and the associated dataset that consists of data spanning 295 Greenlandic glaciers over period 1984-2021. The dataset consists of 278,239 glacier termini for 295 glaciers, and includes ice/ocean masks for the years 2018-2020. The pipeline consists of a Google Earth Engine based downloader , combined with a deep neural network to extract termini locations from the subsetted and preprocessed satellite imagery. The literature review covers most of the existing work in the field. The deep learning methodology also incorporates the greatest diversity of sensors (Landsat 5-8, Sentinel 1 & 2) and sensor types (both optical and SAR), which is a novel development. The methodology is quality controlled by assessing its performance on two uncertainty quantification metrics. In summary, the study represents a significant contribution to the cryosphere and scientific community, by providing a new glacial termini dataset for Greenland, and an automated deep learning based pipeline for automated glacial feature extraction. However, there are certain comments to be addressed regarding the dataset and the manuscript before acceptance at the editor's discretion, as detailed below.

We greatly appreciate the detailed review and constructive comments by Reviewer 2. We have made our best effort to revise the manuscript based on the referee's comments and suggestions. In the following, we made an item-by-item response to the specific comments by the referee.

**Major Comments:**

- A primary concern to be noted is the lack of certain validation metrics that are commonly used in works such as this. Previous studies use the same established validation metrics (average area/distance between predicted and observed termini, or Mean distance error) to ensure ease of comparison. This measure is used in existing works such as Mohajerani et al. (2019), Baumhoer et al. (2019), Cheng et al. (2021), Heidler et al. (2021), Gourmelon et al. (2022), Loebel et al. (2022), and specifically Zhang et al. (2019, 2021). The average uncertainty of 37m, which is calculated using the average distance between duplicate picks from Landsat-8 and Sentinel-2, is somewhat misleading given this context, and the lack of such mean distance error calculation with respect to the ground truth should be addressed. Use of existing validation sets (Cheng et al. (2021), TermPicks/Goliber et al. (2022), and specifically Gourmelon et al. (2022)) would be advisable, as this would allow a fair comparison of this method with existing studies on established measures.

  We agree with the reviewer that using average uncertainty to compare with the measure of uncertainty defined in other related works is somewhat misleading. Following our response to comments from Reviewer 1, we have built a test dataset by randomly choosing 100 traces from TermPicks and used the rest of the TermPicks dataset to train the network from scratch. After training the network, we apply it to the test dataset and quantify the mean distance error between the network's predictions and the manual delineations. The test error of the network is 79 meters, which is now used to compare with others.

- A related concern to be noted is the biases inherent in the chosen validation metrics. One validation metric (average distance between duplicate picks from Landsat-8 and Sentinel-2) is biased towards lower/better values, since it is only calculated on higher resolution images, and doesn't measure the method's performance with respect to manual delineated observations that function as the ground truth. Furthermore, this uncertainty quantification cannot be calculated across the entire dataset, so its use as a metric to gauge the quality of the dataset is questionable.

  We only use duplicated Landsat-8 and Sentinel-2 since (i) duplicate Sentinel-1 traces are used for the georeferencing offset, and (ii) Landsat-5 or -7 lacks overlap with other datasets. We have clarified this point in the text. We now build a test set to quantify the overall error by measuring the deviation between the network's predictions and manual delineations. Since this comment is similar to Major Concern 1 from Reviewer 1, we respond with the same comment as we did there: Quantifying error based on manual delineation involves a trade-off: the more representative the error is, the more manual effort it takes. Since we aim to produce as large a terminus dataset as possible (with a resulting 278,239 glacier termini), a highly representative error would require too much manual effort, which violates our primary objective to save manual effort. For this reason, we still keep the two automated ways to quantify the uncertainty of the terminus data. We agree that uncertainty and error are not the same.

I had not realized that there would be a bias towards better values, as Reviewer 2 here correctly states. This should be addressed somewhere in the text, even if the authors now additionally use the commonly used error metric on a test set, as this bias still exists for the uncertainty measure and should be handled with care.

- The data itself has a few issues that require reevaluation of the automated screening module. Within the provided dataset, there are fronts that are closed loops, make large spatio-temporal jumps, or are otherwise erroneous. Additionally, there is a non-negligible number of glaciers with termini that are cutoff by the boundaries of the ROI, which should be expanded and/or otherwise addressed.
  Without specific time/location identification of these issues, it is difficult to address this comment. Perhaps the reviewer is referring to Figure 3, which shows pre-screened results. Our aim with this figure is to demonstrate some of the issues that we built the screening module to detect. Regarding the ROIs, without a clear identification of the glaciers/times with these issues, it is hard to address this comment. Our ROIs are prepared manually at the beginning of the entire process, and we carefully choose the ROIs to make them cover the glacier termini over their entire image acquisition period.

- While the primary contributions of this study are the data processing pipeline and dataset, there is value in providing some analysis of the results, such as commenting on the general/regional area change trends (as shown for individual glaciers in the supplement, and to a degree in Figure 6), volume loss (when integrated with velocity datasets, though this may be out of scope), or correlations with temperatures/other measurements.
  The main objective of this study is to build a fully automated pipelined that can continuously produce terminus traces and generate a huge terminus trace dataset. We agree with the reviewer that scientific investigation is important and interesting. However, it is out of the scope of this study and will be accomplished in future works that leverage our data compilation.

- The integration of figures in the manuscript could be better handled. Specifically, few figures are referenced within the manuscript (6, 8, 9, and 10 being the exceptions).
  We merge Figure 3 and Figure 5 together as they both show the screening module. We believe that the rest of the figures in the manuscript have distinct and relevant purposes, and we have doublechecked that they are all referenced in the manuscript.

- It would be in the best interests of the community for the TermPicks derived training data to be released for ease of use for future projects.
  Thank you for this suggestion. The reference polygons and label polygons converted from TermPicks traces have been included in the AutoTerm dataset now (10.5281/zenodo.7527485).

- The training & pre/postprocessing of the network can be elaborated upon. The learning rate/regularization factors are less important/useful than information such as the optimizer used, number of epochs trained on, the total number of images trained on, loss function used, vectorization algorithm, and data augmentations used (i.e., if no data augmentations were used, why not, and if so, what were they).
  We add the missing information in Line 208: "*To train the network, we use binary cross entropy as the loss function and stochastic gradient descent method as the optimizer with an L2 regularization factor of $5\times10^{-4}$,as recommended by Zhang et al. (2021). Based on the learning rate in Chen et al. (2018b) and Zhang et al. (2021), we train the network with learning rates of $5\times10^{-3}$, $2\times10^{-3}$, and $1\times10^{-3}$, and choose $2\times10^{-3}$owing to its lowest validation loss.*"
  We adopt similar post-processing procedures with Zhang et al. (2019) that vectorize deep learning output to generate terminus traces. It is now added in Line 139. The information about data augmentation can be found in Line 199.

**Specific Comments:**

**P2 L58:** I would recommend adding Gourmelon et al. (2022) and Loebel et al. (2022) to this list.
We thank the reviewer for pointing this out and have added these references to the reference list.

**P3 L70-71, P7 L210:** There are automated verification steps in Cheng et al. (2021), which includes filtering out unconfident predictions from the DL classifier.

*We thank the reviewer for pointing that out. Cheng et al. (2021) has an automated data screening based on the deviations of two classifications of the network. We have changed the related text as: "Many previous DL methods applied to terminus delineation do not have quality control (Mohajerani et al., 2019; Zhang et al., 2019). Where it does exist, data screening has been simplistic and not automatically applied. For example, Zhang et al. (2021) only considers the complexity of the terminus shape and removes traces with abnormal complexity (which, in turn, requires a threshold to be established for each glacier), Baumhoer et al. (2019) only considers outliers that arise in a time series of terminus position change over time, and Gourmelon et al. (2022) remove the outliers based on terminus length. Cheng et al. (2021) however did design an automated data screening based on the deviation of two classifications from the network. Our screening module is based on using the physical properties of glacier termini." Line 224.*

**P8 L225:** Could a detail/edge preserving speckle filter be applied? Or other types of Sentinel-1 processing steps to reduce speckle noise?

*Considering the coarse resolution of the Sentinel-1 images, we did not apply the speckle filter to avoid blurry images. Also, glacier termini are still observable from the original Sentinel-1 images, even with speckle noise.*

**P11 L341:** Is there a limitation (such as spatial coverage gaps) restricting ice mask generation to 2018-2020, or could they be made for other years?

*They could be made for other years. For certain years, some glaciers might lack terminus traces, but there are no significant spatial coverage gaps in general. We only create updated masks annually beginning in 2018 to serve the ICESat-2 community needs for improved accuracy of laser returns during periods of extensive glacier terminus retreat. We have clarified this in Line 295.*

**P21 Figure 1:** The flowchart is a not straight forward to follow. Perhaps consider separating the training/inference flowcharts, or organizing it in a more linear fashion.

*The figure is composed of three parts: network training (black arrow), terminus inference (blue arrow), and longevity maintenance (red arrow). We chose a figure design to separate these procedures. Moreover, our figure was designed to make good use of the figure space. Based on this comment, we have revised the figure to make the figure clearer by highlighting the training and inference.*

**P26 Figure 6:** The color of the uncertainty bars and your results are the same (both are black). This makes the figure hard to interpret. Additionally, consider using colorblind friendly color schemes.

*We have changed the color schemes and removed the uncertainty bars in the figure, as this figure is mainly for demonstrating the improved temporal resolution of our results. We also modified the color scheme of Figure 8 and Figure 11 and checked the figure through https://www.color-blindness.com/coblis-color-blindnesssimulator/ following comments from Reviewer 1.*

**P31 Figure 11:** Are the uncertainty bars for all of GID164's picks the same size?

*Yes. The uncertainty bar is measured by using duplicate traces. Termini of the same glacier have the same uncertainty valued from duplicate traces. We added the description of the uncertainty bar in the figure caption. We have added two sentences to make the description clear:*
*Line 274: "For each glacier, we average the uncertainties from all duplicated traces and use the mean to represent the uncertainty of that glacier."*
*Line 291: "Thus, in total, each glacier will have six measures of uncertainty: one from duplicate traces and the other five estimated by MC dropout for each sensor."*

---

## Referee Report (RR2)

**General comments**

The authors' response addresses the questions and concerns raised by both reviewers, and integrates much feedback into the revised manuscript. Such revisions include the incorporation of standardized error metrics (i.e., the median distance deviation between predicted and manually measured termini is ~79m, which is slightly better but on par with existing networks). Other addressed concerns include: reproducibility via provision of the training data; restructuring of the manuscript; clarifications/integration of the figures within the text; elaborations on the training process; and the large number of specific/minor comments.

Reasonable explanations are given for comments that are not fully addressed. However, there are concerns regarding the dataset itself. Otherwise, the manuscript passes general, technical, and presentational criteria to a satisfactory degree.

After review of the author's responses, as well as the changes to the revised manuscript, I can recommend that this submission should be subject to technical corrections before acceptance, at the editor's discretion.

**Major comments**

1. Regarding previous concerns raised about the quality of the dataset (Reviewer 2 comment 3): The current automated quality control method does not eliminate the majority of errors visible in the as of now final published AutoTerm dataset (Version 3). The specific comments below cover a non-exhaustive list of specific GIDs with errors to address, and potential quality control algorithms/solutions to implement that may help address them.

2. In conjunction with the suggested quality control measures suggested in the specific comments below, perhaps consider that a semi-automated approach is in order. Consider defining and validating several transects/flowlines (either automatically, or manually) for each glacier, then ensure all valid termini pass through all transects/centerlines, without making large spatio-temporal jumps (use Otsu's thresholding to remove time series outliers when measuring any particular termini's advance/retreat metrics). Ensure movement along multiple flowline/transects is relatively consistent (even though the termini will move differently along different transects, the same general trends of advance and retreat should be consistent to within some tolerance, which may be determined empirically/tuned manually).

- Otherwise, ensure by eye that for each glacier there are no visual artifacts or boundary issues (as in the above images, where the termini are cut off by domain boundary, and are in need of correction).
- For the sake of verification/validation, consider producing graphs for all glaciers like Figure 10 was produced for GID 164, but for movement along multiple flowline/transects as well.
- While implementing all of these suggestions may be out of scope for this study, integrating any of them would still be beneficial for the quality of the AutoTerm dataset and the cryosphere community as a whole.

3. If possible, it would also helpful to see the actual classifications output by the AutoTerm neural network, and see how it performs against images with issues like sea-ice mélange, ice tongues, clouds, debris, data gaps (from image boundaries or otherwise).

- Ultimately, the AutoTerm methodology must either successfully detect the glacial termini under those conditions, or flag the detections as incorrect, as it will otherwise result in an impact the

quality of the final dataset. As is, the dataset is of course still very valuable, but further validation will help to improve ease of use and accessibility.

4. Termpicks data provides a standard metadata format that would be good to adhere to, and would provide information such as source image IDs, should users seek to manually verify the termini picks. Fields like Error could also be renamed to Uncertainty/Variance that would better describe the measure, to be more in line with the changes in the manuscript. This may be out of scope and is not necessary, but would be valuable if provided.

**Specific Comments**

[Figure]

1. GID53 & 54 - These glaciers seem to overlap. Has GID54 not been properly isolated from GID53's domain? Additionally, there are some apparent image artifacts that have persisted through the termini extraction process (see concertation of termini boundaries along centerline – is there a physical explanation for this besides the image boundaries ending within the domain, and getting missed by the automated quality control algorithm?). Furthermore, the highlighted picks should have been removed given inspection of this output data. Perhaps consider removing termini that go beyond the glacial boundaries or intrude too deeply into the ice mask.

[Figure]

2. GID85 - Note the additional erroneous fronts. This data needs to be validated before release. Large amounts of fronts in the ocean and inland indicate potential issues with sea ice and/or clouds. Similar to the above, remove any termini that intrude too deeply into the ocean mask.

[Figure]

3. GID 44 & 45 - Perhaps implement an additional automatic quality control metric, deviation from closest/mean termini. Any termini which deviates a lot from the average/mean/nearest termini should be flagged for exclusion. Note also that the apparent Error (uncertainty) metadata is low, despite other valid termini having higher Error (uncertainty) values.

[Figure]

4. GID 86 & GID 144 - Another quality control metric could account for the distance between points in the detected line (to detect large straight jumps), as well as detecting the number/magnitude of large turns in the termini lines. Similar to the above, be sure to eliminate any detections that intrude too far into the grounded ice mask.

[Figure]

[Figure]

5. GID 120, GID150-153 - While this glacial domain looks fine for the majority of the termini, there are still issues with the domain boundary cutoff and termini that should be removed along those boundaries. Consider checking the proximity of the termini to the domain boundaries, and eliminating them if they have significant portions that are close to the edges & differ from the other termini.

[Figure]

[Figure]

6. GID 262, 253, 210, 211, 214, 215, etc.- Overlapping and erroneous/boundary issue termini. While enlarging the domains and re-evaluating the termini may be out of scope, perhaps add commentary on potential solutions to this issue for future work (such as consolidating domains/GIDs and reprocessing, merging termini across GIDs from the existing dataset, or using the overlapping smaller inset/subset domains to cut away overlapping portions of the larger domain's termini).

---

## Referee Report (RR3)

**Font Color in this response**
The **black** color represents the first round of the reviewer's comments, the blue color represents the first round of the response, the green color represents the second round of the reviewer's comments and the yellow color represents the second round of the response. Purple represents the third round of the reviewer's comments.

**Major Concern 1: Evaluation Protocol**
The pipeline has not been properly tested, and hence, we can not yet rely on its output. In my understanding, the authors seem to confuse uncertainty estimation with error assessment. In line 245, they call the calculation of the difference between prediction and ground truth „uncertainty quantification". The authors then claim that comparing to manually picked traces „requires significant manual effort" because it would have to be redone, as „network accuracy likely varies over time as glaciers experience different conditions". Instead, the authors use two different uncertainty quantifications that do not rely on ground truth data. Calculating uncertainties is definitely useful, and the two used ways of calculating the effect of different sources of uncertainty (model inherent and input inherent) look very promising. However, calculating the uncertainty is no substitute for an error assessment. The authors themselves state in line 395: „if both duplicated traces are deviated from reality but are close to each other, the uncertainty would not represent the reality." It is, therefore, indispensable to calculate the deviation of the network's predictions to manually delineated ground truth traces on a test set that is independent of the train set. First, we need to know how well the network is performing at the moment before we apply it to new unseen data and afterward assess whether the network's performance degrades when new sensors are used or other conditions change (called domain shift in machine learning).

We agree with the reviewer that the difference between predictions and ground truth should be called "error", while the difference between duplicate traces should be taken as "uncertainty". We have identified places in the manuscript where this terminology may have been confused and have updated the text. In addition, we have performed a test of the network as follows. We randomly choose 100 traces from TermPicks as a test dataset and use the rest of the TermPicks data to train the network from scratch. After training the network, we apply it to a test dataset and quantify the deviation of the network's prediction to manual delineations in the test dataset. This reveals a test error of 79 meters, which is similar to previous authors (Mohajerani et al., 2019; Zhang et al., 2019; Baumhoer et al., 2019; Cheng et al., 2020). The description of this test is added to the manuscript in Line 214 and Line 316.

Thank you for performing this test.
Please add more information in section 3.3 about the evaluation on the test set (train-test split – which images were picked for the test set exactly – this information ensures reproducibility; how exactly the error metric is calculated; etc.). Moreover, a split of the error on the test set between sensors would give additional valuable insights.

We appreciate the reviewer's comments. The list of the test set is provided in Table S1, which can be found in Line 326. The method of quantifying test error is described in Line 225. We added a new table (Table S2) to show the test error among the five sensors.

Thank you. Please add the information that the test set is randomly chosen from TermPicks to the manuscript. Line 225 „*We measure the test error by calculating the averaged width of the enclosed area bounded by the TermPicks traces and the network predictions*" – How is width defined here? What happens when the prediction crosses the trace? Will that negate the error? I'm still not sure how exactly the error is calculated. A formula or figure would be helpful.

Quantifying error based on manual delineation involves a trade-off: the more representative the error is, the more manual effort it takes. Since we aim to produce as large a terminus dataset as possible (with a resulting 278,239 glacier termini), a highly representative error would require too much manual effort, which violates our primary objective to save manual effort. For this reason, we still keep the two automated ways to quantify the uncertainty of the terminus data. We agree that uncertainty and error are not the same.

I'm not sure that I am understanding the authors correctly, but in my understanding, this trade-off between the representativeness of the error and the manual effort is not correct. If the test set is small, it needs to be chosen with greater care such that the error will be representative, i.e., the test set should cover the possible variability of the data the network will see.

What I understand from the author's second sentence is: They trade off quality assessment for quantity. As the authors do provide not only the dataset but also advertise their pipeline for future use, the quality assessment needs to be thorough. Still, keeping the uncertainty quantification is a great bonus.

We agree with the reviewer. We now have a test set that contains 100 images to quantify the test error. To further assess the quality of the data, we keep the original two ways of uncertainty quantification.

Although the reviewer states that we cannot rely on our model output, even without the model test we have now performed, we believe our data to be reliable for the following reasons. First, our terminus traces match the remote sensing images (Fig. 4). Second, the time series of terminus variation are in agreement with both Termpicks and CALFIN (Fig. 5). Third, the time series of terminus variations show a clear seasonal signal (refer to the time series data described in section 4.4), which would not be revealed if our terminus traces are unreliable.

Fig. 4 shows only six example traces, and in my regard, checking all 278,239 termini visually manually is also some manual effort, as even if the quality of each trace could be checked in one second, checking all traces would still require at least 10 days. I do not know how many and which images the authors checked, making the assessment not reproducible and subjective. Fig. 5, on the other hand, is a very nice analysis and indicates that there is probably no systemic error in the produced data. Still, this is just a rough hint at the quality and can not replace the test on a test set, which I would like to thank the authors for now providing.

We appreciate the reviewer's comments. We didn't manually check the quality of each trace. Instead, we used the test error and automatically-estimated uncertainty to demonstrate the reliability of the data.

Additionally, an experiment should be conducted to determine whether and by how much the error between prediction and ground truth on the test set is reduced when the screening module is applied versus not applied. In this way, the effectiveness of the screening module can be demonstrated. The same holds for the upsampling of small images (it is not sufficient to visualize the results of one sample, as shown in Fig. 13).

The screening module belongs in post-processing, and is thus not related to network inference or training. Instead, the screening module is for detecting outliers in order to improve data quality. Fig. 3 and the red crosses in Fig. 5 demonstrate the effectiveness of our screening module. For these reasons, we did not see it necessary to validate the effectiveness of the screening module on a test set.

The screening module is part of the complete pipeline which the authors propose. Hence, its effectiveness should, in my regard, be demonstrated in a thorough way as well. This can be done by once using the trained pipeline with and once without the module and reporting the difference in the error metric. Moreover, please also report the differences in the error metric for each sensor, as different sensors are handled differently by the screening module.

We applied the screening module to the test set. For the network trained with TermPicks, the test error is 62 meters after the screening module and 79 before the screening module. The success rate is 90% for the network trained with the TermPicks and 46% for the network trained without the TermPicks.
We added two sentences to describe this:
Line 329: "The success rate of the test set is 90%, and the test error was reduced to 62 meters after the screening module."
Line 404: "That network has a test error of 315 meters and a success rate of 46%, while the network trained with TermPicks has a testing error of 79 meters and a success rate of 90%."
We added a new table (Table S2) to show the test error among the five sensors before and after the screening.

Thank you!

We agree with the reviewer that the test error is needed to demonstrate the effectiveness of upsampling as it is a pre-processing procedure. Thus, we have also conducted an upsampling test. For this test, we randomly select 36 images of five small glaciers to be a test set for size normalization. These images are not included in the training set for the independent evaluation of the size normalization's effectiveness. We add a new table to show the test error and uncertainty from duplicate traces with and without size normalization

(Table S2). The results show that size normalization can effectively reduce test error and uncertainty. The related description is now added in Line 429.

Great work! Thank you! Just the formulation „36 images of five small glaciers that are beyond the training set as the test set" is hard to follow. Please rewrite as done above.

We appreciate the reviewer's comments and rephrased the sentence as "We randomly select 36 images of five small glaciers as the test set for size normalization. These images are beyond the training set." Line 428

**Major Concern 2: Generalizability**
The pipeline has to be tested on out-of-sample data (i.e., glaciers not present in the training dataset) and data outside of Greenland to show generalizability to the global scope.
> 1. Line 451 „Owing to the transferability of deep learning, the entire pipeline has the potential to be applied to many other outlet glaciers around the world"
> 2. Line 135 „converting the TermPicks terminus data into a training dataset suitable for deep learning highly generalizes the network"

These claims have to be proven on such a test set. As most manually annotated traces available from related work are part of TermPicks and hence, have been used for training, another test set has to be used. For testing on SAR imagery, the dataset provided by Gourmelon et al. could, for example, be used, as it is not incorporated in TermPicks (except Jacobshaven, which probably has overlaps with TermPicks). However, test data for optical imagery might have to be created manually (e.g., from Antarctica or the Russian Arctic). At least, I am unaware of a dataset based on optical imagery that is not incorporated in TermPicks.

The importance of the size of training data in the deep learning field has been well demonstrated. For instance, Sun et al. (2017) showed that the network's performance increases logarithmically based on the volume of training data size. For this reason, we see no need to provide additional proof that our model has the ability to generalize.

The authors are correct that there is a relationship between dataset size and the performance of a network. This, however, does not mean that every network that is trained with a sufficiently large dataset will have adequate performance. This must be proven on a test set.

Table S1 shows the test errors before and after training with TermPicks traces. The test error was improved from 315 meters to 79 meters. The related description can be found in Line 404.

Thank you!

Further, our study focus is on Greenland alone – not on a global scale. We merely identify the potential for our model to be used at that scale. That said, out of interest we conducted an experiment in which we trained the network with only 1466 training examples prepared manually without including TermPicks. The test error of this network is 315 meters, which is much larger than 79 meters that we have after training with TermPicks. Such an improvement demonstrates the generalization improvements brought by TermPicks. The related description is added in Line 405.

It seems I have misunderstood what the authors meant by „generalizability". I assumed the meaning was how well the model would predict the termini of images from the target distribution (i.e., all glaciers with termini around the world). However, it seems the authors merely meant how well the model would predict the termini of images from the distribution of data used to train the model (i.e., glaciers that are included in the train set but from future time points given that no other variables significantly change).
I apologize for the confusion. With the evaluation on the test set, the generalizability is proven, and the additional test with TermPicks also nicely proves the sentence from line 135.
However, it has to be stated more clearly (and also in the abstract) that the pipeline is only tested for glaciers in Greenland and not on a global scale and, therefore, can only be applied to Greenland without further precautions!

No need to apologize at all! We appreciate the reviewer's comments as they have significantly improved the manuscript. We have rephrased the sentence in the abstract as "The pipeline has been tested on glaciers in Greenland with an error of 79 meters." Line 16.

Thank you!

**Major Concern 3: Comparability**
It is not possible to compare the calculated uncertainties of this manuscript to the errors calculated in related works, as done in, e.g., line 304 or line 379. Two totally different metrics are compared here, and studies have been conducted on different datasets. For a valid comparison, the exact same network/pipeline needs to be tested on different datasets, or different networks/pipelines have to be trained, optimized, and tested on the exact same data (a so-called benchmark dataset). Altering both the dataset and the network/pipeline introduces too many changes, and a changed performance could result from either the different dataset (for example, the test set might be easier, and therefore, the performance of an otherwise worse performing network would be better on this test set) or the different network/pipeline. Concludingly, the claimed improvements 1 and 2 (line 377 „1 increasing the generalization level of the deep learning network to enable more and better quality terminus predictions; 2 deploying size normalization to improve the accuracy of terminus delineation for small glaciers") are not proven.
One way to show the superior terminus prediction performance on SAR imagery could be the use of the benchmark dataset recently proposed by Gourmelon et al. (2022) (i.e., retraining the pipeline on the train set and evaluating it on the test set using the stated metrics). To the best of our knowledge, there is no equivalent benchmark dataset for optical imagery.

Now that we have performed an error estimate we can more accurately compare our network error to that from other studies (Line 315), and find that our network's performance is comparable with other studies, not superior or inferior.

Please ensure that this is reflected correctly everywhere in the manuscript (e.g., lines 403-407 suggest that your performance is superior).

Although our test error is comparable with other studies, we indeed improve the method regarding the generalization. Converting the TermPicks traces into the training set makes it more representative of the real world and makes the network more generalized, which is demonstrated by the test errors before and after including TermPicks.

I think it is unclear what is meant by „*the deep learning network*" (Line 404 and same in line 79) – I was assuming you meant previous deep learning models. From what I understand now, you mean your own model, which was improved by incorporating TermPicks into the train set. Please rephrase these sections, as like this it sounds like you would improve over previous models (which is not proven – actually, you just have indices that it performs on par).
Additionally, please state in section lines 326-328 that the test sets are different and that the test errors are calculated slightly differently (please see this quote from Cheng et al. 2021: „*The primary quality assessment method is the mean distance error (Mohajerani et al., 2019; Zhang et al., 2019; Baumhoer et al., 2019). Conceptually, this method resembles the numerical integration of the area between two curves, normalized by the average length of the curves (see Fig. 8a). Also referred to as the area over front (A/F) in literature, this method can also be seen as a generalization of the method of transects along arbitrarily oriented fronts (Mohajerani et al., 2019; Baumhoer et al., 2019). This metric is implemented by taking the mean–median of the distances between closest pixels in the predicted and manually delineated fronts.*").

We agree with the review that different networks should be trained and tested on the same benchmark dataset to determine the best. However, such a test is outside of the scope of this study. The objective of this study is not to demonstrate which deep learning network is better but to generate a terminus dataset with spatial coverage and temporal resolution that no prior study has with the addition of further automation in the production pipeline.
For generalization and size normalization, please refer to the response of Major Concern 3 and Major Concern 1, respectively.

With the evaluation on the test set, this is addressed satisfactorily.
I understand that a rigorously correct comparison is outside the scope of this study.

We appreciate the reviewer's comments.

**Major Concern 5: Structure of the manuscript**

The structure of the manuscript has to be improved upon. There is a mix-up between the training and inference of the pipeline, and some information is given twice at different positions in the manuscript. It is hard to tell when the authors write about the newly derived dataset in contrast to the dataset derived from TermPicks for training the network, e.g., in line 295 („We find an average success rate of 64%"), it is unclear on which dataset the success rate was calculated. I would suggest splitting the manuscript into two main parts as follows, but there could also be another better split-up:

    1. Training Pipeline: manual delineated dataset creation (TermPicks + additional manual annotations), neural network (architecture), network training (train-validation-test split, learning rate, number of trained epochs, etc.), screening module, error calculation, uncertainty estimation

    2. Inference Pipeline: new data acquisition + pre-processing, uncertainty estimation on this newly derived dataset, ice/ocean mask updates

We thank the reviewer for these comments. We decided to structure the manuscript following the order of data processing. We first collect remote sensing images and conduct preprocessing (section 3.1). Second, we generate the training dataset by converting the terminus traces in TermPicks into label polygons and pairing polygons with the remote sensing images. Third, we introduce the network structure and the training progress in section 3.3. Sections 3.4 to 3.7 are post-processing procedures after applying the well-trained network to make inferences on the 433,721 images collected via Google Earth Engine. That said, we notice that the original section titles may have caused some confusion. Thus, to address this comment we changed the title of Section 3.2 from "Generalizing the network" to "Generating training data from TermPicks", and the title of Section 3.3 from "Deep Learning Network" to "The Structure and Training of Deep Learning Network".

Furthermore, we have revised some of the text to make it clearer that we first perform network training and then we apply the well-trained network to all the images collected through Google Earth Engine to generate terminus positions. This is done in the last sentence in Section 3.3 as "*After the training, we apply the well-trained network to the test set for quantifying the test error and to all the images collected via GEE for generating the terminus dataset.*" Line 217.

The success rate is the percentage of terminus traces that pass the screen module. We add one sentence at the end of section 3.4: "*Finally, we estimate the success rate by calculating the percentage of the terminus traces that pass the screening module.*" Line 255.

Thank you. I think adding a paragraph like „The rest of the paper is structured as follows ..." (as is done above) to the end of the introduction would aid the reader.
Please also provide the success rate for the test set.

We thank the reviewer for the comments. We added one paragraph at the beginning of the method section as the confusion mainly comes from this section. The added paragraph is: "The structure of the method section follows the order of data processing. We first collect remote sensing images and conduct preprocessing (section 3.1). Second, we generate the training dataset by converting the terminus traces in TermPicks into label polygons and pairing polygons with the remote sensing images. Third, we introduce the network structure and the training progress in section 3.3. Finally, sections 3.4 to 3.7 are post-processing procedures after applying the well-trained network to make inferences on all the images collected via Google Earth Engine."

The success rate is 90% for the test set. We added one sentence in Line 329: "The success rate of the test set is 90%, and the test error was reduced to 62 meters after the screening module."

Actually, the confusion does not come solely from the method section. For example, section 2 – „input data" - input to what? The pipeline? I was under the impression that the model would just get remote sensing imagery as input and not additionally an ice/ocean mask?

In section 4.1, the authors first introduce the ‚success rate', which should, however, be introduced in the methods section.

We add one sentence at the end of section 3.4: "*Finally, we estimate the success rate by calculating the percentage of the terminus traces that pass the screening module.*" Line 255.

Thank you.

Moreover, paragraph line 188 to 195 should be moved to limitations.

Line 194 to 201 describe the limitations of the input imagery, not the limitations of our work. These points are discussed here because they explain why we needed to prepare additional training data. Therefore, we think this text fits better in Section 3.2, which describes the training data preparation.

I understand. Does your pipeline still have difficulties with these kinds of images? If so, please additionally add a few sentences about it in the limitations.

We add one sentence in Line 453: "Another limitation is that even though we include additional training data, the network might struggle with some challenging situations (Fig. S2)."

Should this not be investigated to prove the correctness of your produced dataset? Please check and report it in the manuscript.

**Major Comments:**

5. The normalization of image sizes is not clear to me. Small images are upsampled, but large images are not downsampled. Hence, do they still have different sizes? I would not call this normalization, then. Moreover, the authors extract patches afterward, so the input size is always equal anyways. Additionally, only showing one figure that shows an improvement for one trace is not sufficient evidence that this upsampling generally improves the delineation performance. Please show the improvement in numbers over a complete, independent test set (refer to major concerns 1 and 2).

Different glaciers are not of the same physical size and therefore don't have similar number of pixels that fall within their fjord walls. This can vary from 300 pixels to 3000 pixels. As a result, the deep learning input (image patch) may sometimes cover more than the entire glacier and may alternatively sometimes only cover a part of a glacier.

After normalization, all small glaciers will be covered by images with a width of about 1000 pixels, regardless of their original image width. In other words, the normalization makes glaciers appear to the deep learning network as if they had a similar physical size. We have attempted to clarify this in the text. Line 168.

Thank you, I understand now that you meant the physical size. So, what is the mean physical size of one pixel now? Moreover, how is upsampling performed? Linear interpolation or a nearest neighbor interpolation?

The normalization will make small glaciers appear to have a larger physical size without changing the physical size of a pixel. For example, if a glacier has a physical width of 1000 meters and is covered by an image with a width of 500 pixels, after normalization, the image width will be 1000 pixels and each pixel will still have the same resolution as before. Thus, the nominal physical width of the glacier becomes 2000 meters.

OK.

We use the cubic upsampling method.

Please add this information to the manuscript.

We agree with the reviewer that test error is needed to demonstrate the effectiveness of upsampling as it is a pre-processing procedure. We randomly select 36 images of five small glaciers that are beyond the training set as the test set for size normalization. We add a new table to show the test error and uncertainty from duplicate traces with and without size normalization (Table S2). The results show that size normalization can effectively reduce test error and uncertainty. The related description is now added in Line 428.

Thank you.

This is Table S3 now. Please correct.

6. Section 3.3:

4. „This network has been proven to have large learning capability, spatial transferability [...]": These are quite big claims based on a train set of two glaciers and a test set of one glacier that are all located in Greenland (Zhang et al., 2021).

The large learning capability has been demonstrated by the paper of DeepLabV3+ (Chen et al., 2018b). Zhang et al. (2021) applied the network to a glacier beyond the training set, showing the network's spatial transferability. In this work, we apply the network to 295 glaciers in Greenland and generate 278,239 glacier termini, of which 17,906 terminus traces are from the training set. This means that 94% of our results are beyond the training set, which further demonstrates the learning capability and spatial transferability of the network.

As this sentence only references Zhang et al. (2021), the authors are not reporting on the present study. Please call it „network architecture" and not „network" to make clear what you are referring to and add the reference to Chen et al., 2018b as well.

We thank the reviewer for the comments and revised the related sentence as "This network structure has been proven to have large learning capability, spatial transferability, and the capability of using multi-sensor remote sensing images (Zhang et al., 2021)." Line 213. We also added Chen et al., 2018b to the reference.

Network „architecture" is actually a technical term. Hence, please do not call it „structure". Moreover, please also add the reference (Chen et al., 2018b) after this sentence directly, as the large learning capability has been shown by it and not Zhang et al. (2021).

8. „The network training takes about a week": This is quite long and might be due to a sub-optimal learning rate. Please specify not only the training time but also the number of trained iterations over the complete augmented dataset. Also, specify your train-test-split and your metrics for evaluation (refer to major concern 1). Moreover, did you use an early stopping criterion? You might have overfitted during this long training time.

The long training time is caused by the large training dataset. We have 17906 training examples, and the augmentation increases the training set by a factor of four. The training takes more than 600,000 iterations. Among all the training data, we select 5% of images randomly as validation datasets to conduct early stopping. The training will be stopped when the validation error stops decreasing for 3 consecutive epochs. We add these details in Line 214.

Thank you very much. I assume by iteration, inputting one batch into the network and backpropagating its loss to update the weights is meant? Please instead provide the times the network has seen the train set („iterations over the train set"), which is normally defined as the number of epochs. If my assumption is correct, that would be (600,000 * 16) / (17906 * 4). Out of curiosity, is the checkpoint saved after each batch?

Yes, your understanding of the iteration is correct. The network training stopped at seven epochs. The training examples are counted by the number of images. The remote-sensing images will be split into image patches with identical sizes. In total, we have around 1,300,000 patches after augmentation. Therefore, the number of epochs is (600,000 * 16)/1300000≈7. The checkpoint will be saved after each epoch if the validation loss decreases.

Thank you!

11. Line 224 „percentile of the data range": Do you refer to the data range of the generated training data? Is this computed per glacier? Per satellite? The validity of these thresholds needs to be checked on an independent test set (refer to major concerns 1 and 2).

For each glacier, we will calculate the thresholds based on the termini from the same satellite. For instance, we will calculate thresholds for Sentinel-2 traces of GID2 and Landsat-8 traces of GID2 separately. We add one sentence in Line 242 to provide additional clarification: "*The thresholds are calculated automatically based on the results of the same glacier and same satellite.*"

Thank you. It is also now clear that you calculate these metrics on outputs of the network. So the screening can only be done if the pipeline is applied to a time series of images? Or do you store a rolling average of each glacier and each satellite in your pipeline?

Yes, the screening can be done if we apply the pipeline to a series of images and generate a bunch of termini.

The question was if the screening can only be done on time series or whether it is possible also when I just want one single trace from one satellite image.

The screening module belongs to the post-processing, which is not related to network inference or training. The screening module is for detecting outliers and maintaining the data quality. Fig. 3 and the red crosses in Fig. 5 demonstrate the effectiveness of our screening module. So, we did not validate the effectiveness of the screening module on a test set.

Fig. 3 and Fig. 5 are insufficient proof of the effectiveness. The screening module is part of the complete pipeline which the authors propose. Hence, its effectiveness should, in my regard, be demonstrated in a thorough way as well.

We applied the screening module to the test set. For the network trained with TermPicks, the test error is 62 meters after the screening module and 79 before the screening module. The success rate is 90% for the network trained with the TermPicks and 46% for the network trained without the TermPicks.

We added two sentences to describe this:

Line 329: "The success rate of the test set is 90%, and the test error was reduced to 62 meters after the screening module."

Line 404: "That network has a test error of 315 meters and a success rate of 46%, while the network trained with TermPicks has a testing error of 79 meters and a success rate of 90%."

We added a new table (Table S2) to show the test error among the five sensors before and after the screening.

Thank you!

12. Line 227 „For outliers in terminus length, we remove both the lower and upper thresholds (Eqns. 1 and 2) because we do not anticipate large changes in terminus length in either direction (bigger or smaller)." As far as I understood, these thresholds were calculated on data for Greenland. Hence, the optimal thresholds for, e.g., Antarctica, might completely deviate from the ones calculated for Greenland. This might hinder the global applicability of the pipeline (refer to major concerns 1 and 2). This should be added to the limitations.

We agree with the reviewer that these thresholds differ for Greenland and Antarctica glaciers. They are different among glaciers in Greenland. However, these thresholds are determined **automatically** based on the distribution of termini from each satellite and each glacier. Therefore, we believe it is feasible to apply the method globally. We have clarified this in the text. Line 242.

I understand now. However, please indicate that the screening has not been tested on glaciers outside of Greenland.

We add one sentence in Line 413: "Despite the success of the screening module in Greenland, further validation will be needed as applying it globally."

Thank you, however please add this in the limitations section.

17. The results do, at some points, not validate the conclusions. No correlation was calculated (or it was not stated in the manuscript), and even a correlation would not necessarily induce causality. Please rephrase the conclusions to hypotheses.

1. Line 307 „glaciers with less training data will have larger uncertainties and lower success rates"

Rephrased as "*glaciers with less training data will probably have larger uncertainties and lower success rates*" Line 331.

2. Line 309 „since they have the highest spatial resolution"

Rephrased as "*Among the five datasets used, Landsat-8 and Sentinel-2 have the lowest average uncertainties, probably because they have the highest spatial resolution.*" Line 333.

3. Line 310 onwards „The reasons for the Landsat-5 uncertainty are twofold [...]"

Rephrased as: "*The reasons for the Landsat-5 uncertainty might be twofold.*" Line 335.

4. Line 314 „The higher uncertainty of Sentinel-1 images is due to its low image quality, coarse

resolution, and the lower volume of training data derived from this sensor."

Rephrased as "*The higher uncertainty of Sentinel-1 images could be due to its low image quality, coarse resolution, and the lower volume of training data derived from this sensor.*" Line 338.

Thank you for addressing my points here. Please also go over the complete manuscript to search for further such claims (e.g., line 327 „However, the network does struggle to delineate termini in many wintertime Sentinel-1 images because of blurry boundaries and the lack of sufficient training data specifically using Sentinel-1 imagery"). There is some mix-up of results and discussion (e.g., line 325 „Such variations are largely caused by the uneven distribution of the training data—glaciers with more training data have higher success rates." Here again, causality is not proven. The authors **observe** that glaciers with

more training data have higher success rates, and **conclude** that the uneven distribution of training data might cause this.) which should be addressed. Please separate plain results and conclusions into two sections (i.e., the conclusions should be moved from the results section to the discussion section).

Thanks for the comments. We rephase the following sentences:
• Line 331 "Such variations could be caused by the uneven distribution of the training data---glaciers with more training data have higher success rates."
• Line 332 "However, the network does struggle to delineate termini in many wintertime Sentinel-1 images, probably because of blurry boundaries and the lack of sufficient training data specifically using Sentinel-1 imagery."
• Line 340: "The duplicate trace uncertainty varies between glaciers along with success rates might be because the training data is not evenly distributed for each glacier"
We move the third paragraph of section 4.1 to the discussion section (section 5.4 Difference of the two types of uncertainties).

Thank you.

18. Line 325 „the uncertainty from duplicate traces is more representative of Landsat-7 and Sentinel-2 than other datasets" - Is it not only representative of these two datasets, as it was only calculated for these?

For each glacier, we average the uncertainties from all duplicated traces and use the mean to represent the uncertainty of that glacier. Each glacier has one value of uncertainty from duplicate traces, and that value is more representative of Landsat-8 and Sentinel-2 as the value comes from these two satellites. We have added two sentences to make the description clear:

Line 274: "*For each glacier, we average the uncertainties from all duplicated traces and use the mean to represent the uncertainty of that glacier.*"

Line 291: "*Thus, in total, each glacier will have six measures of uncertainty: one from duplicate traces and the other five estimated by MC dropout for each sensor.*"

Still, my question stands: Is the uncertainty calculated with duplicate traces (and just this uncertainty, not the MC dropout one) not only representative of these two sensors, as it was only calculated for these? (Line 355)

The duplicate uncertainty is calculated from the Landsat-8 and Sentinel-2, but we use the value to represent the network's uncertainty on a certainty glacier. Therefore, such an uncertainty is more representative of Landsat-8 and Sentinel-2, and would be biased towards a lower value when representing the uncertainty of results obtained from other satellites. To clarify, we added on sentence in Line 465: "Since Landsat-8 and Sentinel-2 images have the highest resolution among the five satellites, using the duplicate uncertainty to represent the error of results obtained from other satellites would be biased towards lower values."

This must also be part of the limitations section, not only the „Difference of the two types of uncertainties" section.

20. Line 355 „The metadata contains the date in YYYY-MM-DD, Glacier ID, source image satellite, and the uncertainty of each trace by averaging the two types of uncertainties provided": I thought the uncertainties were not available for every single trace, as they were only calculated for some of them due to computational limitations? Please clarify.

Each glacier has six measures of uncertainty: one from duplicate traces and the other five estimated by MC dropout for each sensor. The uncertainty of each trace is estimated by averaging the uncertainty from duplicate traces and the MC dropout uncertainty of its satellite sensor. We added one sentence to clarify in Line 291: "*Thus, in total, each glacier will have six measures of uncertainty: one from duplicate traces and the other five estimated by MC dropout for each sensor.*"

Please additionally indicate in line 386 that the average is taken as the uncertainty measure for each of the glacier's traces.

Moreover, an additional question arises: In lines 293-296, the authors describe the procedure for how the dataset is generated. But how is it configured in the automatic pipeline if, say, it is used for glaciers in Antarctica? Will MC dropout uncertainty be calculated for each image, or how are images randomly chosen here?

We rephrase the sentence in Line 379: "The entire record of uncertainties is provided in a spreadsheet. Each glacier has six averaged uncertainty measures, including one from duplicate trace uncertainty and five from MC dropout uncertainties of different satellites."

Thanks.

We randomly choose ten images from each of the five sensors and make three inferences for each of the ten images. The related description can be found in Line 298. We combine For loop and "shuf -n 10" in Bash to automatically and randomly choose ten images for calculating MC dropout uncertainty.

Sorry, I think I did not make myself clear. Let me rephrase: When I apply your pipeline to a new glacier in Antarctica for one date and one sensor, will it calculate the MC dropout uncertainty for it?

21. Line 434: „The pipeline can alert us of its failure based on the success rate within the screening module.": With your limitation that the screening module might not provide valid results for glaciers with few training examples, this alert might not trigger.

When most of the results are of good quality, the terminus features, such as length, will have a Gaussianlike distribution, where most of the terminus lengths are within the thresholds determined by the screening module, and the success rate will be high. For glaciers with few training examples, their results might be of poor quality. In that case, the distribution of the terminus features will be relatively scattered because many terminus lengths could be unreasonably long or short. As a result, the screening module will detect many traces as outliers, and these glaciers will have a low success rate, which can be used as an alert. We have clarified the text to point this out more readily: " *The network's failure will result in many termini not passing the screening. The pipeline can use the low success rates to alert us to prepare more training data for the corresponding glaciers.*" Line 464.

As the thresholds of the screening module are calculated for each glacier individually based on the network's predictions, the screening module will not filter out wrong predictions of a glacier where only wrong predictions ever occur (e.g., due to very few training examples). I assume all three variables (terminus length, curvature, and enclosed area) would roughly be uniformly distributed (as the predictions would be somewhat random), which would lead to a smaller Q1 and a bigger Q3 and, therefore, to a lower $T\_L$ and a higher $T\_U$. This, in turn, reduces the amount of filtered-out predictions. Please correct me if I have an error in my logic.

Your understanding is mostly correct. It's just that when all three variables are uniformly distributed, even if we have a lower $T\_L$ and higher $T\_U$, we will filter out more predictions than normal cases as the results will have a concentrated distribution in the normal cases.

I'm sorry, I do not understand what you mean by normal cases here and what your difference between results and predictions is. Which concentrated distribution? Please explain, as my argument still stands, and I think this is important for both your dataset and future use of the pipeline.

22. Please revise the color scheme of your figures, as red and green should not appear in the same plot (https://www.the-cryosphere.net/submission.html#figurestables).

Thank you for bringing this to our attention. We have modified the color schemes of Figure 6, Figure 8, and Figure 11, and checked figures through https://www.color-blindness.com/coblis-color-blindnesssimulator/.

Please also do the same for the remaining figures (5, 7, 10).

We are sorry for the confusion. We have changed the color scheme for figures 5,7, and 10. We have merged the original figures 3 and 5. So, figures 6, 8, and 11 in response actually mean figures 5,7, and 10

Please revise all figures, as there are still several that show green and red in one plot.

24. Figure S5: Visually, this does not appear to be a linear relationship. Have you done a correlation test?

Yes, the uncertainties from duplicate traces and MC dropout do not show a linear relationship. The differences in the two types of uncertainties are caused by their quantification methods and source images. We explained the details of the differences in the last paragraph of section 4.1, Data Quality.

I don't understand. Why does the caption then claim that there is a linear relationship?

Our previous response is somewhat misleading. There is a linear relationship between the two uncertainties but they are not exactly the same. The linear relationship is more clear in Landsat-8 and Sentinel-2 but less obvious among Landsat-5, Landsat-7, and Sentinel-1. Since uncertainty values vary drastically across glaciers, we now use the natural logarithm to show the comparison

between the two types of uncertainties (see the updated Fig. S5). The correlation coefficient is 0.69 for Landsat-8 and Sentinel-2, and 0.43 for the rest three satellites.
Thank you.

**Response to Reviewer 2's comments**

**Major Comments:**

- A related concern to be noted is the biases inherent in the chosen validation metrics. One validation metric (average distance between duplicate picks from Landsat-8 and Sentinel-2) is biased towards lower/better values, since it is only calculated on higher resolution images, and doesn't measure the method's performance with respect to manual delineated observations that function as the ground truth. Furthermore, this uncertainty quantification cannot be calculated across the entire dataset, so its use as a metric to gauge the quality of the dataset is questionable. We only use duplicated Landsat-8 and Sentinel-2 since (i) duplicate Sentinel-1 traces are used for the georeferencing offset, and (ii) Landsat-5 or -7 lacks overlap with other datasets. We have clarified this point in the text. We now build a test set to quantify the overall error by measuring the deviation between the network's predictions and manual delineations. Since this comment is similar to Major Concern 1 from Reviewer 1, we respond with the same comment as we did there: Quantifying error based on manual delineation involves a trade-off: the more representative the error is, the more manual effort it takes. Since we aim to produce as large a terminus dataset as possible (with a resulting 278,239 glacier termini), a highly representative error would require too much manual effort, which violates our primary objective to save manual effort. For this reason, we still keep the two automated ways to quantify the uncertainty of the terminus data. We agree that uncertainty and error are not the same.
I had not realized that there would be a bias towards better values, as Reviewer 2 here correctly states. This should be addressed somewhere in the text, even if the authors now additionally use the commonly used error metric on a test set, as this bias still exists for the uncertainty measure and should be handled with care.
We thank the reviewer for the comment and added one sentence in Line 465: "Since Landsat-8 and Sentinel-2 images have the highest resolution among the five satellites, using the duplicate uncertainty to represent the error of results obtained from other satellites would be biased towards lower values."
Sorry, I should have been clearer. This must also be part of the limitations section, not only the „Difference of the two types of uncertainties" section.

---

## Referee Report (RR4)

**General Comments:**

The authors' response addresses the questions and comments raised by all reviewers, and integrates the feedback into the revised manuscript.

Specific revisions include detailed responses and technical corrections to issues in the dataset, and integration of suggested quality control measures into the methodology. The authors acceptably address reviewer concerns, and the added text is free from syntactical errors.

After reviewal of the author's responses, as well as the changes to the revised manuscript & associated datset, I can recommend that this submission should be accepted, at the editor's discretion.

**Specific Comments:**

N/A

---

## Author Response (AR2)

**Response to Reviewer 1's comments**

**Font Color in this response**

The **black** color represents the first round of the reviewer's comments, the **blue** color represents the first round of the response, the **green** color represents the second round of the reviewer's comments and the **yellow** color represents the second round of the response.

**Major Concern 1: Evaluation Protocol**

The pipeline has not been properly tested, and hence, we can not yet rely on its output. In my understanding, the authors seem to confuse uncertainty estimation with error assessment. In line 245, they call the calculation of the difference between prediction and ground truth „uncertainty quantification". The authors then claim that comparing to manually picked traces „requires significant manual effort" because it would have to be redone, as „network accuracy likely varies over time as glaciers experience different conditions". Instead, the authors use two different uncertainty quantifications that do not rely on ground truth data. Calculating uncertainties is definitely useful, and the two used ways of calculating the effect of different sources of uncertainty (model inherent and input inherent) look very promising. However, calculating the uncertainty is no substitute for an error assessment. The authors themselves state in line 395: „if both duplicated traces are deviated from reality but are close to each other, the uncertainty would not represent the reality." It is, therefore, indispensable to calculate the deviation of the network's predictions to manually delineated ground truth traces on a test set that is independent of the train set. First, we need to know how well the network is performing at the moment before we apply it to new unseen data and afterward assess whether the network's performance degrades when new sensors are used or other conditions change (called domain shift in machine learning).

We agree with the reviewer that the difference between predictions and ground truth should be called "error", while the difference between duplicate traces should be taken as "uncertainty". We have identified places in the manuscript where this terminology may have been confused and have updated the text. In addition, we have performed a test of the network as follows. We randomly choose 100 traces from TermPicks as a test dataset and use the rest of the TermPicks data to train the network from scratch. After training the network, we apply it to a test dataset and quantify the deviation of the network's prediction to manual delineations in the test dataset. This reveals a test error of 79 meters, which is similar to previous authors (Mohajerani et al., 2019; Zhang et al., 2019; Baumhoer et al., 2019; Cheng et al., 2020). The description of this test is added to the manuscript in Line 214 and Line 316.

Thank you for performing this test. Please add more information in section 3.3 about the evaluation on the test set (train-test split – which images were picked for the test set exactly – this information ensures reproducibility; how exactly the error metric is calculated; etc.). Moreover, a split of the error on the test set between sensors would give additional valuable insights.

We appreciate the reviewer's comments. The list of the test set is provided in Table S1, which can be found in Line 326. The method of quantifying test error is described in Line 225. We added a new table (Table S2) to show the test error among the five sensors.

Quantifying error based on manual delineation involves a trade-off: the more representative the error is, the more manual effort it takes. Since we aim to produce as large a terminus dataset as possible (with a resulting

278,239 glacier termini), a highly representative error would require too much manual effort, which violates our primary objective to save manual effort. For this reason, we still keep the two automated ways to quantify the uncertainty of the terminus data. We agree that uncertainty and error are not the same.

I'm not sure that I am understanding the authors correctly, but in my understanding, this trade-off between the representativeness of the error and the manual effort is not correct. If the test set is small, it needs to be chosen with greater care such that the error will be representative, i.e., the test set should cover the possible variability of the data the network will see.

What I understand from the author's second sentence is: They trade off quality assessment for quantity. As the authors do provide not only the dataset but also advertise their pipeline for future use, the quality assessment needs to be thorough. Still, keeping the uncertainty quantification is a great bonus.

We agree with the reviewer. We now have a test set that contains 100 images to quantify the test error. To further assess the quality of the data, we keep the original two ways of uncertainty quantification.

Although the reviewer states that we cannot rely on our model output, even without the model test we have now performed, we believe our data to be reliable for the following reasons. First, our terminus traces match the remote sensing images (Fig. 4). Second, the time series of terminus variation are in agreement with both Termpicks and CALFIN (Fig. 5). Third, the time series of terminus variations show a clear seasonal signal (refer to the time series data described in section 4.4), which would not be revealed if our terminus traces are unreliable.

Fig. 4 shows only six example traces, and in my regard, checking all 278,239 termini visually manually is also some manual effort, as even if the quality of each trace could be checked in one second, checking all traces would still require at least 10 days. I do not know how many and which images the authors checked, making the assessment not reproducible and subjective. Fig. 5, on the other hand, is a very nice analysis and indicates that there is probably no systemic error in the produced data. Still, this is just a rough hint at the quality and can not replace the test on a test set, which I would like to thank the authors for now providing.

We appreciate the reviewer's comments. We didn't manually check the quality of each trace. Instead, we used the test error and automatically-estimated uncertainty to demonstrate the reliability of the data.

Additionally, an experiment should be conducted to determine whether and by how much the error between prediction and ground truth on the test set is reduced when the screening module is applied versus not applied. In this way, the effectiveness of the screening module can be demonstrated. The same holds for the upsampling of small images (it is not sufficient to visualize the results of one sample, as shown in Fig. 13).

The screening module belongs in post-processing, and is thus not related to network inference or training. Instead, the screening module is for detecting outliers in order to improve data quality. Fig. 3 and the red crosses in Fig. 5 demonstrate the effectiveness of our screening module. For these reasons, we did not see it necessary to validate the effectiveness of the screening module on a test set.

The screening module is part of the complete pipeline which the authors propose. Hence, its effectiveness should, in my regard, be demonstrated in a thorough way as well. This can be done by once using the trained pipeline with and once without the module and reporting the difference in the error metric. Moreover, please also report the differences in the error metric for each sensor, as different sensors are handled differently by the screening module.

We applied the screening module to the test set. For the network trained with TermPicks, the test error is 62 meters after the screening module and 79 before the screening module. The success rate is 90% for the network trained with the TermPicks and 46% for the network trained without the TermPicks.

We added two sentences to describe this:

Line 329: "*The success rate of the test set is 90%, and the test error was reduced to 62 meters after the screening module.*"

Line 404: "*That network has a test error of 315 meters and a success rate of 46%, while the network trained with TermPicks has a testing error of 79 meters and a success rate of 90%.*"

We added a new table (Table S2) to show the test error among the five sensors before and after the screening.

We agree with the reviewer that the test error is needed to demonstrate the effectiveness of upsampling as it is a pre-processing procedure. Thus, we have also conducted an upsampling test. For this test, we randomly select 36 images of five small glaciers to be a test set for size normalization. These images are not included in the training set for the independent evaluation of the size normalization's effectiveness. We add a new table to show the test error and uncertainty from duplicate traces with and without size normalization (Table S2). The results show that size normalization can effectively reduce test error and uncertainty. The related description is now added in Line 429.

Great work! Thank you! Just the formulation „36 images of five small glaciers that are beyond the training set as the test set" is hard to follow. Please rewrite as done above.

We appreciate the reviewer's comments and rephrased the sentence as "*We randomly select 36 images of five small glaciers as the test set for size normalization. These images are beyond the training set.*" Line 428

**Major Concern 2: Generalizability**

The pipeline has to be tested on out-of-sample data (i.e., glaciers not present in the training dataset) and data outside of Greenland to show generalizability to the global scope.

1. Line 451 „Owing to the transferability of deep learning, the entire pipeline has the potential to be applied to many other outlet glaciers around the world"

2. Line 135 „converting the TermPicks terminus data into a training dataset suitable for deep learning highly generalizes the network"

These claims have to be proven on such a test set. As most manually annotated traces available from related work are part of TermPicks and hence, have been used for training, another test set has to be used. For testing on SAR imagery, the dataset provided by Gourmelon et al. could, for example, be used, as it is not incorporated in TermPicks (except Jacobshaven, which probably has overlaps with TermPicks). However, test data for optical imagery might have to be created manually (e.g., from Antarctica or the Russian Arctic). At least, I am unaware of a dataset based on optical imagery that is not incorporated in TermPicks.

The importance of the size of training data in the deep learning field has been well demonstrated. For instance, Sun et al. (2017) showed that the network's performance increases logarithmically based on the volume of training data size. For this reason, we see no need to provide additional proof that our model has the ability to generalize.

The authors are correct that there is a relationship between dataset size and the performance of a network. This, however, does not mean that every network that is trained with a sufficiently large dataset will have adequate performance. This must be proven on a test set.

Table S1 shows the test errors before and after training with TermPicks traces. The test error was improved from 315 meters to 79 meters. The related description can be found in Line 404.

Further, our study focus is on Greenland alone – not on a global scale. We merely identify the potential for our model to be used at that scale. That said, out of interest we conducted an experiment in which we trained the network with only 1466 training examples prepared manually without including TermPicks. The test error of this network is 315 meters, which is much larger than 79 meters that we have after training with TermPicks. Such an improvement demonstrates the generalization improvements brought by TermPicks. The related description is added in Line 405.

It seems I have misunderstood what the authors meant by „generalizability". I assumed the meaning was how well the model would predict the termini of images from the target distribution (i.e., all glaciers with termini around the world). However, it seems the authors merely meant how well the model would predict the termini of images from the distribution of data used to train the model (i.e., glaciers that are included in the train set but from future time points given that no other variables significantly change).

I apologize for the confusion. With the evaluation on the test set, the generalizability is proven, and the additional test with TermPicks also nicely proves the sentence from line 135. However, it has to be stated more clearly (and also in the abstract) that the pipeline is only tested for glaciers in Greenland and not on a global scale and, therefore, can only be applied to Greenland without further precautions!

No need to apologize at all! We appreciate the reviewer's comments as they have significantly improved the manuscript. We have rephrased the sentence in the abstract as "*The pipeline has been tested on glaciers in Greenland with an error of 79 meters.*" Line 16.

**Major Concern 3: Comparability**

It is not possible to compare the calculated uncertainties of this manuscript to the errors calculated in related works, as done in, e.g., line 304 or line 379. Two totally different metrics are compared here, and studies have been conducted on different datasets. For a valid comparison, the exact same network/pipeline needs to be tested on different datasets, or different networks/pipelines have to be trained, optimized, and tested on the exact same data (a so-called benchmark dataset). Altering both the dataset and the network/pipeline introduces too many changes, and a changed performance could result from either the different dataset (for

example, the test set might be easier, and therefore, the performance of an otherwise worse performing network would be better on this test set) or the different network/pipeline. Concludingly, the claimed improvements 1 and 2 (line 377 „1 increasing the generalization level of the deep learning network to enable more and better quality terminus predictions; 2 deploying size normalization to improve the accuracy of terminus delineation for small glaciers") are not proven. One way to show the superior terminus prediction performance on SAR imagery could be the use of the benchmark dataset recently proposed by Gourmelon et al. (2022) (i.e., retraining the pipeline on the train set and evaluating it on the test set using the stated metrics). To the best of our knowledge, there is no equivalent benchmark dataset for optical imagery.

Now that we have performed an error estimate we can more accurately compare our network error to that from other studies (Line 315), and find that our network's performance is comparable with other studies, not superior or inferior.

Please ensure that this is reflected correctly everywhere in the manuscript (e.g., lines 403-407 suggest that your performance is superior).

Although our test error is comparable with other studies, we indeed improve the method regarding the generalization. Converting the TermPicks traces into the training set makes it more representative of the real world and makes the network more generalized, which is demonstrated by the test errors before and after including TermPicks.

We agree with the review that different networks should be trained and tested on the same benchmark dataset to determine the best. However, such a test is outside of the scope of this study. The objective of this study is not to demonstrate which deep learning network is better but to generate a terminus dataset with spatial coverage and temporal resolution that no prior study has with the addition of further automation in the production pipeline.

For generalization and size normalization, please refer to the response of Major Concern 3 and Major Concern 1, respectively.

With the evaluation on the test set, this is addressed satisfactorily. I understand that a rigorously correct comparison is outside the scope of this study.

We appreciate the reviewer's comments.

**Major Concern 4: Reproducibility**

Please make your complete assembled training data (including the satellite imagery) publicly available, as only in this way the reproducibility of the results is guaranteed. Moreover, please also provide the manually created reference polygons for each glacier.

All our remote sensing images are freely available on Google Earth Engine. The code for collecting the data is also available on GitHub. TermPicks is also a publicly available dataset. The reference polygons and label polygons converted from TermPicks traces have been included in the AutoTerm dataset now (10.5281/zenodo.7527485).

Thank you for addressing this issue. Recreating a dataset might have produced some slightly different data, which might again have influenced the training of a network.

We appreciate the reviewer's comments.

**Major Concern 5: Structure of the manuscript**

The structure of the manuscript has to be improved upon. There is a mix-up between the training and inference of the pipeline, and some information is given twice at different positions in the manuscript. It is hard to tell when the authors write about the newly derived dataset in contrast to the dataset derived from TermPicks for training the network, e.g., in line 295 („We find an average success rate of 64%"), it is unclear on which dataset the success rate was calculated. I would suggest splitting the manuscript into two main parts as follows, but there could also be another better split-up:

1. Training Pipeline: manual delineated dataset creation (TermPicks + additional manual annotations), neural network (architecture), network training (train-validation-test split, learning rate, number of trained epochs, etc.), screening module, error calculation, uncertainty estimation 2. Inference Pipeline: new data acquisition + pre-processing, uncertainty estimation on this newly derived dataset, ice/ocean mask updates

We thank the reviewer for these comments. We decided to structure the manuscript following the order of data processing. We first collect remote sensing images and conduct preprocessing (section 3.1). Second, we generate the training dataset by converting the terminus traces in TermPicks into label polygons and pairing polygons with the remote sensing images. Third, we introduce the network structure and the training progress in section 3.3. Sections 3.4 to 3.7 are post-processing procedures after applying the well- trained network to make inferences on the 433,721 images collected via Google Earth Engine. That said, we notice that the original section titles may have caused some confusion. Thus, to address this comment we changed the title of Section 3.2 from "Generalizing the network" to "Generating training data from TermPicks", and the title of Section 3.3 from "Deep Learning Network" to "The Structure and Training of Deep Learning Network".

Furthermore, we have revised some of the text to make it clearer that we first perform network training and then we apply the well-trained network to all the images collected through Google Earth Engine to generate terminus positions. This is done in the last sentence in Section 3.3 as "*After the training, we apply the well-trained network to the test set for quantifying the test error and to all the images collected via GEE for generating the terminus dataset.*" Line 217.

The success rate is the percentage of terminus traces that pass the screen module. We add one sentence at the end of section 3.4: "*Finally, we estimate the success rate by calculating the percentage of the terminus traces that pass the screening module.*" Line 255.

Thank you. I think adding a paragraph like „The rest of the paper is structured as follows ..." (as is done above) to the end of the introduction would aid the reader.

Please also provide the success rate for the test set.

We thank the reviewer for the comments. We added one paragraph at the beginning of the method section as the confusion mainly comes from this section. The added paragraph is: "*The structure of the method section follows the order of data processing. We first collect remote sensing images and conduct preprocessing (section 3.1). Second, we generate the training dataset by converting the terminus traces in TermPicks into label polygons and pairing polygons with the remote sensing images. Third, we introduce the network structure and the training progress in section 3.3. Finally, sections 3.4 to 3.7 are post-processing procedures after applying the well-trained network to make inferences on all the images collected via Google Earth Engine.*"

The success rate is 90% for the test set. We added one sentence in Line 329: "*The success rate of the test set is 90%, and the test error was reduced to 62 meters after the screening module.*"

In section 4.1, the authors first introduce the ‚success rate', which should, however, be introduced in the methods section.

We add one sentence at the end of section 3.4: "*Finally, we estimate the success rate by calculating the percentage of the terminus traces that pass the screening module.*" Line 255.

Thank you.

Moreover, paragraph line 188 to 195 should be moved to limitations.

Line 194 to 201 describe the limitations of the input imagery, not the limitations of our work. These points are discussed here because they explain why we needed to prepare additional training data. Therefore, we think this text fits better in Section 3.2, which describes the training data preparation.

I understand. Does your pipeline still have difficulties with these kinds of images? If so, please additionally add a few sentences about it in the limitations.

We add one sentence in Line 453: "*Another limitation is that even though we include additional training data, the network might struggle with some challenging situations (Fig. S2).*"

An explanation of the two uncertainty measures, as given in lines 319 to 323, should be moved to further at the beginning of the manuscript.

The principle of the two uncertainty measures is described fully in Methods (Section 3.6). Lines 344 to 355 explains the uncertainty results and focus on why these two uncertainties are different. Therefore, we believe lines 344 to 355 fit better in the Results section.

I understand. Thank you.

**Major Comments:**

1. It is unclear to me whether the name „AutoTerm" refers to the automated pipeline or the derived dataset, or                                                                                                                                                   both.
We thank the reviewer for pointing this out. We think it is a good idea to use AutoTerm reflect both the data and pipeline. We have changed the title of the manuscript to: "AutoTerm: an automated pipeline for glacier terminus extraction using machine learning and a "big data" repository of Greenland glacier termini."

Yes, thank you.

2. The title of the manuscript does not mention the automated pipeline, which is, in my humble regard, the most significant contribution. Hence, I'd argue for a more suitable title, e.g. AutoTerm: an automated Google Earth Engine pipeline for glacier terminus extraction and „big data" repository of Greenland glacier termini

We thank the reviewer's suggestion. We have changed the title of the manuscript to: "AutoTerm: an automated pipeline for glacier terminus extraction using machine learning and a "big data" repository of Greenland glacier termini."

Thank you.

3. It needs to be clarified whether the region of interest that has to be defined for each new glacier has to be a polygon like in figure 2 or whether it can simply be a bounding box.

Each region of interest is a bounding box. The polygon in Figure 2 is a reference polygon only for converting TermPicks traces into a training label polygon, which is described in the Figure caption. We have updated the text where needed to clarify this point. Line 146.

Thank you.

4. Line 163 onwards: „This allows glaciers with various natural sizes to have a similar image size in computer vision, which largely decreases the complexity of delineating glacier terminus." This statement (the second part of it) needs more explanation or a reference. We rephrased the text here and added a reference: "*We then normalize the image size, which is commonly adopted in the computer vision field for better capturing object features with various physical sizes (Xu et al., 2017).*" Line 167.

Thank you.

5. The normalization of image sizes is not clear to me. Small images are upsampled, but large images are

not downsampled. Hence, do they still have different sizes? I would not call this normalization, then. Moreover, the authors extract patches afterward, so the input size is always equal anyways. Additionally, only showing one figure that shows an improvement for one trace is not sufficient evidence that this upsampling generally improves the delineation performance. Please show the improvement in numbers over a complete, independent test set (refer to major concerns 1 and 2).

Different glaciers are not of the same physical size and therefore don't have similar number of pixels that fall within their fjord walls. This can vary from 300 pixels to 3000 pixels. As a result, the deep learning input (image patch) may sometimes cover more than the entire glacier and may alternatively sometimes only cover a part of a glacier.

After normalization, all small glaciers will be covered by images with a width of about 1000 pixels, regardless of their original image width. In other words, the normalization makes glaciers appear to the deep learning network as if they had a similar physical size. We have attempted to clarify this in the text. Line 168.

Thank you, I understand now that you meant the physical size. So, what is the mean physical size of one pixel now? Moreover, how is upsampling performed? Linear interpolation or a nearest neighbor interpolation?

The normalization will make small glaciers appear to have a larger physical size without changing the physical size of a pixel. For example, if a glacier has a physical width of 1000 meters and is covered by an image with a width of 500 pixels, after normalization, the image width will be 1000 pixels and each pixel

will still have the same resolution as before. Thus, the nominal physical width of the glacier becomes 2000 meters.

We use the cubic upsampling method.

We agree with the reviewer that test error is needed to demonstrate the effectiveness of upsampling as it is a pre-processing procedure. We randomly select 36 images of five small glaciers that are beyond the training set as the test set for size normalization. We add a new table to show the test error and uncertainty from duplicate traces with and without size normalization (Table S2). The results show that size normalization can effectively reduce test error and uncertainty. The related description is now added in Line 428.

Thank you.

6. Section 3.3:

1. „encoder-decoder structure [...] can obtain sharp object boundaries": Actually, an encoder decoder structure without skip connections would most probably not recover any details and, therefore, no sharp object boundaries. In Chen et al. (2018), they use a more sophisticated method to obtain the sharp boundaries: „A fully connected CRF [conditional random field] is then applied to refine the segmentation result and better capture the object boundaries."

We realize that there is a mistake in our citation. The original DeepLab use the CRF to refine the segmentation results (Chen et al., 2018a, DeepLab: Semantic Image Segmentation with Deep Convolutional Nets, Atrous Convolution and Fully Connected CRFs). Later on, the same author improved the DeepLab and developed DeepLabV3+ (Chen et al., 2018b; Encoder-Decoder with Atrous Separable Convolution for Semantic Image Segmentation), which is the network we use in this manuscript. The DeepLabV3+ adds a simple yet effective decoder module to refine the segmentation results especially along object boundaries (Chen et al., 2018b).

I see, thank you for correcting your reference.

2. „atrous convolution [...] senses multi-scale contextual information": It is not the atrous convolutions alone that make recognition of multi-scale contextual information possible, but the combination of atrous convolutions in ASPP. "Atrous convolution allows us to explicitly control the resolution at which feature responses are computed within Deep Convolutional Neural Networks. It also allows us to effectively enlarge the field of view of filters to incorporate larger context without increasing the number of parameters or the amount of computation. Second, we propose atrous spatial pyramid pooling (ASPP) to robustly segment objects at multiple scales." (Chen et al., 2018)

We thank the reviewer's comments and changed "atrous convolution" to "atrous spatial pyramid pooling". Thank you.

3. „multi-scale contextual information [...] [is] helpful for our task since [...] we integrate remote sensing datasets with different spatial resolutions": Multi-scale refers to how many pixels a neuron is able to see (effective receptive field) and not how much square meters one pixel can see. Hence, multi-scale contextual information helps when the calving front covers many versus only a few pixels. Thus, it helps only indirectly with different spatial resolutions of the dataset.

We agree with the reviewer's comment. Multi-scale refers to the network's ability to sense various portions of the images but not sense images with different resolutions. We rephrase the sentence as

"*Sharp boundaries can improve delineation accuracy, and sensing multi-scale information helps indirectly when we integrate remote sensing datasets with different spatial resolutions.*" Line 205 Thank you.

4. „This network has been proven to have large learning capability, spatial transferability [...]": These are quite big claims based on a train set of two glaciers and a test set of one glacier that are all located in Greenland (Zhang et al., 2021).

The large learning capability has been demonstrated by the paper of DeepLabV3+ (Chen et al., 2018b). Zhang et al. (2021) applied the network to a glacier beyond the training set, showing the network's spatial transferability. In this work, we apply the network to 295 glaciers in Greenland and generate 278,239 glacier termini, of which 17,906 terminus traces are from the training set. This means that 94% of our results are beyond the training set, which further demonstrates the learning capability and spatial transferability of the network.

As this sentence only references Zhang et al. (2021), the authors are not reporting on the present study. Please call it „network architecture" and not „network" to make clear what you are referring to and add the reference to Chen et al., 2018b as well.

We thank the reviewer for the comments and revised the related sentence as "*This network structure has been proven to have large learning capability, spatial transferability, and the capability of using multi-sensor remote sensing images (Zhang et al., 2021).*" Line 213. We also added Chen et al., 2018b to the reference.

5. „The network is trained with a learning rate of 0.005 [...] as recommended by (Zhang et al., 2021)": The optimal learning rate for training is highly dependent on the dataset as well as on the batch size (not just the model). Hence, the learning rate has to be treated as a hyperparameter, which has to be optimized on a validation set (not the test set). A sub-optimal learning rate can lead to significantly longer training times until convergence or no convergence at all.

We agree with the reviewer and have tried to train the network with learning rates of $5\times10^{-3}$, $2\times10^{-3}$, and $1\times10^{-3}$. The validation losses for these three learning rates are 0.023, 0.020, and 0.024, respectively. Overall, the validation losses are measured using binary cross entropy and are similar to each other. We chose the learning rate ($2\times10^{-3}$) with the lowest validation error. We rephrased the sentence as: "*Based on the learning rate in Chen et al. (2018b) and Zhang et al. (2021), we train the network with learning rates of $5\times10^{-3}$, $2\times10^{-3}$, and $1\times10^{-3}$, and choose $2\times10^{-3}$owing to its lowest validation loss.*" Line 210.

Thank you very much.

6. „we choose the largest batch size (16)": This should be „largest possible batch size (16) on an A100 GPU with 40/80 GB GPU memory". Please specify whether your A100s have 40 or 80 GB GPU memory. The sentence is rephrased as follows: "*we choose the largest possible batch size (16) on four A100 GPUs with 160 GB GPU memory in total. We set the batch size to a power of two to take full advantage of GPU processing (Kandel and Castelli, 2020)*" Line 212.

Thank you very much.

7. What exactly is meant by "maximize our computational power" in line 204?

It means using as much GPU memory as possible. The sentence is now rephrased as *"we choose the largest possible batch size (16) on four A100 GPUs with 160 GB GPU memory in total."* Line 212. Thank you.

8. „The network training takes about a week": This is quite long and might be due to a sub-optimal learning rate. Please specify not only the training time but also the number of trained iterations over the complete augmented dataset. Also, specify your train-test-split and your metrics for evaluation (refer to major concern 1). Moreover, did you use an early stopping criterion? You might have overfitted during this long training time.

The long training time is caused by the large training dataset. We have 17906 training examples, and the augmentation increases the training set by a factor of four. The training takes more than 600,000 iterations. Among all the training data, we select 5% of images randomly as validation datasets to conduct early stopping. The training will be stopped when the validation error stops decreasing for 3 consecutive epochs. We add these details in Line 214.

Thank you very much. I assume by iteration, inputting one batch into the network and backpropagating its loss to update the weights is meant? Please instead provide the times the network has seen the train set („iterations over the train set"), which is normally defined as the number of epochs. If my assumption is correct, that would be (600,000 * 16) / (17906 * 4).

Out of curiosity, is the checkpoint saved after each batch?

Yes, your understanding of the iteration is correct. The network training stopped at seven epochs. The training examples are counted by the number of images. The remote-sensing images will be split into image patches with identical sizes. In total, we have around 1,300,000 patches after augmentation. Therefore, the number of epochs is (600,000 * 16)/1300000≈7.

The checkpoint will be saved after each epoch if the validation loss decreases.

7. Line 210 „do not have any quality control": At least Cheng et al. have manual control. So, maybe rephrase it to „do not have any automated quality control".

We thank the reviewer for pointing that out. Cheng et al. (2021) have an automated data screening based on the deviations of two classifications of the network. We have changed the related text as: *"Many previous DL methods applied to terminus delineation do not have quality control (Mohajerani et al., 2019; Zhang et al., 2019). Where it does exist, data screening has been simplistic and not automatically applied. For example, Zhang et al. (2021) only considers the complexity of the terminus shape and removes traces with abnormal complexity (which, in turn, requires a threshold to be established for each glacier), Baumhoer et al. (2019) only considers outliers that arise in a time series of terminus position change over time, and Gourmelon et al. (2022) remove the outliers based on terminus length. Cheng et al. (2021) however did design an automated data screening based on the deviation of two classifications from the network. Our screening module is based on using the physical properties of glacier termini."* Line 224. Thank you.

8. Line 215 onwards: Please mention that the screening builds on top of existing works here (Zhang et al. 2021 – terminus curvature screening, Baumhoer et al. 2019 – time series outliers, Gourmelon et al. 2022 –

removal of too short termini predictions), but goes one step further, i.e., doesn't use any manual intervention or prior knowledge of the data.

The related sentences are revised as follows: "*Based on the previous works (Baumhoer et al. 2019 , Zhang et al. 2021, Gourmelon et al. 2022), we develop an automated screening module that forgoes any manual intervention or prior knowledge of the data.*" Line 232.

Thank you.

9. Line 217 „Terminus length is determined by the sum of the piece-wise length along an individual terminus trace". Please explain in more detail. This, at least for me, is hard to understand. Each terminus trace is an ordered set of points. The length is the sum of the length between the two closest points. The description refers to how we calculate the terminus length, which might be confusing to readers. Terminus length is just the physical length of the glacier terminus. As a result of this confusion, we decided to remove this sentence.

Thank you.

10. Line 218 „Terminus curvature is computed between two adjacent points for each point along the terminus and then an average is taken for each terminus trace." This is also not completely clear to me. I think an equation would help.

We have rephrased the related text as: "*Terminus curvature is computed among every three adjacent points along the terminus based on Peijin Zhang's work (https://github.com/peijin94/PJCurvature), and then an average is taken for each terminus traces.*" Line 235.

Thank you.

11. Line 224 „percentile of the data range": Do you refer to the data range of the generated training data? Is this computed per glacier? Per satellite? The validity of these thresholds needs to be checked on an independent      test      set      (refer      to      major      concerns      1      and      2). For each glacier, we will calculate the thresholds based on the termini from the same satellite. For instance, we will calculate thresholds for Sentinel-2 traces of GID2 and Landsat-8 traces of GID2 separately. We add one sentence in Line 242 to provide additional clarification: "*The thresholds are calculated automatically based on the results of the same glacier and same satellite.*"

Thank you. It is also now clear that you calculate these metrics on outputs of the network. So the screening can only be done if the pipeline is applied to a time series of images? Or do you store a rolling average of each glacier and each satellite in your pipeline?

Yes, the screening can be done if we apply the pipeline to a series of images and generate a bunch of termini.

The screening module belongs to the post-processing, which is not related to network inference or training. The screening module is for detecting outliers and maintaining the data quality. Fig. 3 and the red crosses in Fig. 5 demonstrate the effectiveness of our screening module. So, we did not validate the effectiveness of the screening module on a test set.

Fig. 3 and Fig. 5 are insufficient proof of the effectiveness. The screening module is part of the complete pipeline which the authors propose. Hence, its effectiveness should, in my regard, be demonstrated in a thorough way as well.

We applied the screening module to the test set. For the network trained with TermPicks, the test error is 62 meters after the screening module and 79 before the screening module. The success rate is 90% for the network trained with the TermPicks and 46% for the network trained without the TermPicks.

We added two sentences to describe this:

Line 329: "*The success rate of the test set is 90%, and the test error was reduced to 62 meters after the screening module.*"

Line 404: "*That network has a test error of 315 meters and a success rate of 46%, while the network trained with TermPicks has a testing error of 79 meters and a success rate of 90%.*"

We added a new table (Table S2) to show the test error among the five sensors before and after the screening.

12. Line 227 „For outliers in terminus length, we remove both the lower and upper thresholds (Eqns. 1 and 2) because we do not anticipate large changes in terminus length in either direction (bigger or smaller)." As far as I understood, these thresholds were calculated on data for Greenland. Hence, the optimal thresholds for, e.g., Antarctica, might completely deviate from the ones calculated for Greenland. This might hinder the global applicability of the pipeline (refer to major concerns 1 and 2). This should be added to the limitations.

We agree with the reviewer that these thresholds differ for Greenland and Antarctica glaciers. They are different among glaciers in Greenland. However, these thresholds are determined **automatically** based on the distribution of termini from each satellite and each glacier. Therefore, we believe it is feasible to apply the method globally. We have clarified this in the text. Line 242.

I understand now. However, please indicate that the screening has not been tested on glaciers outside of Greenland.

We add one sentence in Line 413: "*Despite the success of the screening module in Greenland, further validation will be needed as applying it globally.*"

13. Line 235 „We then repeat this screening procedure ten times to maintain the quality of the terminus product": What screening procedure is meant here exactly now? All three or just the one with large areas? And does the outcome change when the screening procedure is done several times? If yes, please explain why.

The pipeline of the screening module is shown in Figure 3. For the first time, the inter-quartile range quantifies thresholds based on the distribution of terminus features such as length, and we remove the outlier traces. Such removal changes the distribution of terminus features, and we will have new thresholds for the second time and detect new outliers. We keep doing this ten times or until we don't find any more outliers. We have clarified this in the text. Line 254.

Thank you.

14. Line 245 „Traditional uncertainty quantification for glacier terminus position is conducted by calculating the difference between manually picked termini and automatically-picked termini." This is not uncertainty quantification but an error assessment (see major concerns 1). We agree with the reviewer's comments and have revised the related sentence. See our response to major concern 1.

Thank you.

15. Line 262 „instead of quantifying the uncertainties of terminus traces, [Hartmann et al.] use the multiple inferences of MC dropout as extra information to retrain the network. " This is not quite correct. Hartmann et al. use the model uncertainty on one specific input as additional information for a second network with dropout. This second network then outputs several predictions again from which uncertainties could be calculated - but instead, to make it more robust, the predictions are averaged to eliminate this uncertainty.

We thank the reviewer's comments and rephrase the sentence as "*Hartmann et al. (2021) applied MC dropout to glacier terminus delineation and built a two-stage approach. They used the uncertainty of the first network as additional information to train the second network. The multiple outputs of the second network are averaged to eliminate the uncertainty and get the final prediction.*" Line 283.

Thank you.

16. Line 267 „To strike a balance between computational cost and the reliability of the MC dropout, we randomly chose ten images from all the sensors and make three inferences for each of them": It is not quite clear to me. Are ten images of each sensor taken? „in total each glacier will have two measures of uncertainty" – So, also ten images of each glacier?

Our original description is somewhat misleading. For each glacier, we will randomly select ten images for each sensor, and we will have 50 images in total. Using the ten images from the same sensor, we conduct MC dropout to quantify one uncertainty for that sensor. We have two measures of uncertainty, one is from duplicate traces, and the other is from MC dropout. The ten images from the same sensor are only for quantifying MC dropout uncertainty. We rephrase the text as "*To*

*strike a balance between computational cost and the reliability of the MC dropout, we randomly chose ten images from each of the five sensors and made three inferences for each of the images. Thus, in total, each glacier will have six measures of uncertainty: one from duplicate traces and the other five estimated by MC dropout for each sensor.*" Line 289.

Thank you.

17. The results do, at some points, not validate the conclusions. No correlation was calculated (or it was not stated in the manuscript), and even a correlation would not necessarily induce causality. Please rephrase the conclusions to hypotheses.

1. Line 307 „glaciers with less training data will have larger uncertainties and lower success rates" Rephrased as "*glaciers with less training data will probably have larger uncertainties and lower success rates*" Line 331.

2. Line 309 „since they have the highest spatial resolution"

Rephrased as "*Among the five datasets used, Landsat-8 and Sentinel-2 have the lowest average uncertainties, probably because they have the highest spatial resolution.*" Line 333. 3. Line 310 onwards „The reasons for the Landsat-5 uncertainty are twofold [...]" Rephrased as: "*The reasons for the Landsat-5 uncertainty might be twofold.*" Line 335. 4. Line 314 „The higher uncertainty of Sentinel-1 images is due to its low image quality, coarse

resolution, and the lower volume of training data derived from this sensor."

Rephrased as "*The higher uncertainty of Sentinel-1 images could be due to its low image quality, coarse resolution, and the lower volume of training data derived from this sensor.*" Line 338.

Thank you for addressing my points here. Please also go over the complete manuscript to search for further such claims (e.g., line 327 „However, the network does struggle to delineate termini in many wintertime Sentinel-1 images because of blurry boundaries and the lack of sufficient training data specifically using Sentinel-1 imagery"). There is some mix-up of results and discussion (e.g., line 325 „Such variations are largely caused by the uneven distribution of the training data—glaciers with more training data have higher success rates." Here again, causality is not proven. The authors **observe** that glaciers with more training data have higher success rates, and **conclude** that the uneven distribution of training data might cause this.) which should be addressed. Please separate plain results and conclusions into two sections (i.e., the conclusions should be moved from the results section to the discussion section).

Thanks for the comments. We rephase the following sentences:

- Line 331 "*Such variations could be caused by the uneven distribution of the training data---glaciers with more training data have higher success rates.*"
- Line 332 "*However, the network does struggle to delineate termini in many wintertime Sentinel-1 images, probably because of blurry boundaries and the lack of sufficient training data specifically using Sentinel-1 imagery.*"
- Line 340: "*The duplicate trace uncertainty varies between glaciers along with success rates might be because the training data is not evenly distributed for each glacier*"

We move the third paragraph of section 4.1 to the discussion section (section 5.4 Difference of the two types of uncertainties).

18. Line 325 „the uncertainty from duplicate traces is more representative of Landsat-7 and Sentinel-2 than other datasets" - Is it not only representative of these two datasets, as it was only calculated for these? For each glacier, we average the uncertainties from all duplicated traces and use the mean to represent the uncertainty of that glacier. Each glacier has one value of uncertainty from duplicate traces, and that value is more representative of Landsat-8 and Sentinel-2 as the value comes from these two satellites. We have added two sentences to make the description clear:

Line 274: "*For each glacier, we average the uncertainties from all duplicated traces and use the mean to represent the uncertainty of that glacier.*"

Line 291: "*Thus, in total, each glacier will have six measures of uncertainty: one from duplicate traces and the other five estimated by MC dropout for each sensor.*"

Still, my question stands: Is the uncertainty calculated with duplicate traces (and just this uncertainty, not the MC dropout one) not only representative of these two sensors, as it was only calculated for these? (Line 355)

The duplicate uncertainty is calculated from the Landsat-8 and Sentinel-2, but we use the value to represent the network's uncertainty on a certainty glacier. Therefore, such an uncertainty is more representative of Landsat-8 and Sentinel-2, and would be biased towards a lower value when representing the uncertainty of results obtained from other satellites. To clarify, we added on sentence in Line 465: "*Since Landsat-8 and Sentinel-2 images have the highest resolution among the five satellites, using the duplicate uncertainty to represent the error of results obtained from other satellites would be biased towards lower values.*"

19. Line 308 „Among the five datasets used, Landsat-8 and Sentinel-2 have the lowest average uncertainties": Please give the exact numbers here. A table showing the different values for different data subsets would be good.

The numbers are shown in Figure 8. The missing information is now added to the figure caption.

Thank you.

20. Line 355 „The metadata contains the date in YYYY-MM-DD, Glacier ID, source image satellite, and the uncertainty of each trace by averaging the two types of uncertainties provided": I thought the uncertainties were not available for every single trace, as they were only calculated for some of them due to computational limitations? Please clarify.

Each glacier has six measures of uncertainty: one from duplicate traces and the other five estimated by MC dropout for each sensor. The uncertainty of each trace is estimated by averaging the uncertainty from duplicate traces and the MC dropout uncertainty of its satellite sensor. We added one sentence to clarify in Line 291: "*Thus, in total, each glacier will have six measures of uncertainty: one from duplicate traces and the other five estimated by MC dropout for each sensor.*"

Please additionally indicate in line 386 that the average is taken as the uncertainty measure for each of the glacier's traces. Moreover, an additional question arises: In lines 293-296, the authors describe the procedure for how the dataset is generated. But how is it configured in the automatic pipeline if, say, it is used for glaciers in Antarctica? Will MC dropout uncertainty be calculated for each image, or how are images randomly chosen here?

We rephrase the sentence in Line 379: "*The entire record of uncertainties is provided in a spreadsheet. Each glacier has six averaged uncertainty measures, including one from duplicate trace uncertainty and five from MC dropout uncertainties of different satellites.*"

We randomly choose ten images from each of the five sensors and make three inferences for each of the ten images. The related description can be found in Line 298. We combine For loop and "shuf -n 10" in Bash to automatically and randomly choose ten images for calculating MC dropout uncertainty.

Line 423: „additional training data will be required to improve the data quality": or an improved network/pipeline.
Revised as suggested.

Thank you.

21. Line 434: „The pipeline can alert us of its failure based on the success rate within the screening module.": With your limitation that the screening module might not provide valid results for glaciers with few training examples, this alert might not trigger.

When most of the results are of good quality, the terminus features, such as length, will have a Gaussian like distribution, where most of the terminus lengths are within the thresholds determined by the screening module, and the success rate will be high. For glaciers with few training examples, their results might be of poor quality. In that case, the distribution of the terminus features will be relatively scattered because many terminus lengths could be unreasonably long or short. As a result, the screening module will detect many traces as outliers, and these glaciers will have a low success rate, which can be used as an alert. We have clarified the text to point this out more readily: " *The network's failure will result in many termini not passing the screening. The pipeline can use the low success rates to alert us to prepare more training data for the corresponding glaciers.*" Line 464.

As the thresholds of the screening module are calculated for each glacier individually based on the network's predictions, the screening module will not filter out wrong predictions of a glacier where only wrong predictions ever occur (e.g., due to very few training examples). I assume all three variables (terminus length, curvature, and enclosed area) would roughly be uniformly distributed (as the predictions would be somewhat random), which would lead to a smaller Q1 and a bigger Q3 and, therefore, to a lower T_L and a higher T_U. This, in turn, reduces the amount of filtered- out predictions. Please correct me if I have an error in my logic.

Your understanding is mostly correct. It's just that when all three variables are uniformly distributed, even if we have a lower T_L and higher T_U, we will filter out more predictions than normal cases as the results will have a concentrated distribution in the normal cases.

22. Please revise the color scheme of your figures, as red and green should not appear in the same plot (https://www.the-cryosphere.net/submission.html#figurestables).
Thank you for bringing this to our attention. We have modified the color schemes of Figure 6, Figure 8, and Figure 11, and checked figures through https://www.color-blindness.com/coblis- color-blindnesssimulator/.

Please also do the same for the remaining figures (5, 7, 10).

We are sorry for the confusion. We have changed the color scheme for figures 5,7, and 10. We have merged the original figures 3 and 5. So, figures 6, 8, and 11 in response actually mean figures 5,7, and 10.

23. Figure 9: Is the number on the bottom left the average? Moreover, it would be good to state between which sensors the duplicates were calculated in the description of the figure.
Yes. We have added the missing information in the figure captions.

Thank you.

24. Figure S5: Visually, this does not appear to be a linear relationship. Have you done a correlation test?
Yes, the uncertainties from duplicate traces and MC dropout do not show a linear relationship. The differences in the two types of uncertainties are caused by their quantification methods and source images. We explained the details of the differences in the last paragraph of section 4.1, Data Quality.
I don't understand. Why does the caption then claim that there is a linear relationship?

Our previous response is somewhat misleading. There is a linear relationship between the two uncertainties but they are not exactly the same. The linear relationship is more clear in Landsat-8 and Sentinel-2 but less obvious among Landsat-5, Landsat-7, and Sentinel-1. Since uncertainty values vary drastically across glaciers, we now use the natural logarithm to show the comparison between the two types of uncertainties (see the updated Fig. S5). The correlation coefficient is 0.69 for Landsat-8 and Sentinel-2, and 0.43 for the rest three satellites.

**Specific Comments:**

1. Line 56 onwards: Heidler et al. 2022 (Deep Active Contour Models for Delineating Glacier Calving Fronts), Loebel et al. 2022 (Extracting glacier calving fronts by deep learning: the benefit of multispectral, topographic and textural input features), Gourmelon et al. 2022 (Calving fronts and where to find them: a benchmark dataset and methodology for automatic glacier calving front extraction from synthetic aperture radar imagery), and Davari et al. 2022 (Pixelwise Distance Regression for Glacier Calving Front Detection and Segmentation) are missing.

We thank the reviewer's comments, and all the missing citations are included.

Thank you.

2. Line 188: „Although TermPicks covers a range of conditions and brings great diversity to the training set, additional training data would improve the accuracy of the network in difficult situations." Please rephrase more cautiously (e.g., „... would presumably improve ..."), as you have no hard evidence that further training data would really improve the accuracy in this situation.

The sentence is revised as suggested.

Thank you.

3. Line 205: GPU -> GPUs

Revised as suggested.

Thank you.

4. Line 220 „With these three metrics, we calculate the lower (TL) and upper thresholds (TU) for each based on the inter-quartile range:" - The sentence structure is hard to follow. So, you compute the thresholds for                          each                          individual                          criterion?
Yes. The sentence is rephrased as "*For each of the three metrics, we calculate the lower (TL) and upper thresholds (TU) based on the inter-quartile range:*"

Thank you.

5. Line 417 „120 GB of GPU memory": I guess you mean 120GB RAM? There are only a 40GB and an 80GB A100 version as far as I know, and 4 (=number of GPUs) times 40 GB is already 160 GB.
I mean 120 GB of GPU memory. We have four A100 GPUs with a total memory of 160 GB, and we use 120 GB of memory. We didn't use all the GPU memory since we wanted the batch size to be a power of 2. When setting the batch size to 16, our network will need 120 GB of GPU memory.

I see, thank you.

6. Line 444: Remove the word „fully", as you still have some manual steps like defining the region of interest.
The region of interest is only manually defined once at the beginning, and it won't interrupt the pipeline for continuous producing termini. Also, the region of interest will be **automatically** defined by the intersection between the terminus and the flowline in the future. So, we still think our pipeline is fully automated.
OK.

7. Table 1 includes abbreviations that were not introduced.

The information in the last column is not necessary so it was removed.

Ok, thank you.

8. Figure S2: Please name the conditions in the figure's description as well, referencing (a) to (e).

The names of the conditions are included in the figure caption.

Thank you.

**Response to Reviewer 2's comments**

**Major Comments:**

- A related concern to be noted is the biases inherent in the chosen validation metrics. One validation metric (average distance between duplicate picks from Landsat-8 and Sentinel-2) is biased towards lower/better values, since it is only calculated on higher resolution images, and doesn't measure the method's performance with respect to manual delineated observations that function as the ground truth. Furthermore, this uncertainty quantification cannot be calculated across the entire dataset, so its use as a metric to gauge the quality of the dataset is questionable.

  We only use duplicated Landsat-8 and Sentinel-2 since (i) duplicate Sentinel-1 traces are used for the georeferencing offset, and (ii) Landsat-5 or -7 lacks overlap with other datasets. We have clarified this point in the text. We now build a test set to quantify the overall error by measuring the deviation between the network's predictions and manual delineations. Since this comment is similar to Major Concern 1 from Reviewer 1, we respond with the same comment as we did there: Quantifying error based on manual delineation involves a trade-off: the more representative the error is, the more manual effort it takes. Since we aim to produce as large a terminus dataset as possible (with a resulting 278,239 glacier termini), a highly representative error would require too much manual effort, which violates our primary objective to save manual effort. For this reason, we still keep the two automated ways to quantify the uncertainty of the terminus data. We agree that uncertainty and error are not the same.

I had not realized that there would be a bias towards better values, as Reviewer 2 here correctly states. This should be addressed somewhere in the text, even if the authors now additionally use the commonly used

error metric on a test set, as this bias still exists for the uncertainty measure and should be handled with care.

We thank the reviewer for the comment and added one sentence in Line 465: "*Since Landsat-8 and Sentinel-2 images have the highest resolution among the five satellites, using the duplicate uncertainty to represent the error of results obtained from other satellites would be biased towards lower values.*"

**General comments**

The authors' response addresses the questions and concerns raised by both reviewers, and integrates much feedback into the revised manuscript. Such revisions include the incorporation of standardized error metrics (i.e., the median distance deviation between predicted and manually measured termini is ~79m, which is slightly better but on par with existing networks). Other addressed concerns include: reproducibility via provision of the training data; restructuring of the manuscript; clarifications/integration of the figures within the text; elaborations on the training process; and the large number of specific/minor comments.

Reasonable explanations are given for comments that are not fully addressed. However, there are concerns regarding the dataset itself. Otherwise, the manuscript passes general, technical, and presentational criteria to a satisfactory degree.

After review of the author's responses, as well as the changes to the revised manuscript, I can recommend that this submission should be subject to technical corrections before acceptance, at the editor's discretion.

*We greatly appreciate the careful inspection of the data and the constructive suggestions about the data screening by Reviewer 2. We have made our best effort to revise the manuscript and improve the data quality based on the referee's comments. In the following, we made an item-by-item response to the specific comments by the referee.*

**Major comments**

1. Regarding previous concerns raised about the quality of the dataset (Reviewer 2 comment 3): The current automated quality control method does not eliminate the majority of errors visible in the as of now final published AutoTerm dataset (Version 3). The specific comments below cover a non-exhaustive list of specific GIDs with errors to address, and potential quality control algorithms/solutions to implement that may help address them.

*We thank the reviewer for the careful inspection of our products. We have conducted the following things to improve the data quality:*

- *Refine the glacier domain to truncate the termini and avoid them intruding into the ocean and land.*
- *For results from Landsat-5 and Landsat-7, we use the threshold derived from Landsat-8 because Landsat-7 is affected by SLC-off, and Landsat-5 has a small number of images. The related description is added in Line 251: "Since the Sentinel-1 images suffer from speckle noise, Landsat-7 is affected by SLC-off, and Landsat-5 has a small number of images, the results generated from these satellites are of relatively poor quality compared to the other datasets, or the obtained thresholds are inappropriate. Therefore, we calculate the thresholds based on results from Landsat-8 and Sentinel-2 and apply them uniformly to all remaining datasets."*

- Keep the termini with a single intersection with the glacier center flowline. Please see the details in response to Comment 2.

In response to the specific comments, we show the updated termini of selected glaciers.

2. In conjunction with the suggested quality control measures suggested in the specific comments below, perhaps consider that a semi-automated approach is in order. Consider defining and validating several transects/flowlines (either automatically, or manually) for each glacier, then ensure all valid termini pass through all transects/centerlines, without making large spatio-temporal jumps (use Otsu's thresholding to remove time series outliers when measuring any particular termini's advance/retreat metrics). Ensure movement along multiple flowline/transects is relatively consistent (even though the termini will move differently along different transects, the same general trends of advance and retreat should be consistent to within some tolerance, which may be determined empirically/tuned manually).

• Otherwise, ensure by eye that for each glacier there are no visual artifacts or boundary issues (as in the above images, where the termini are cut off by domain boundary, and are in need of correction).

• For the sake of verification/validation, consider producing graphs for all glaciers like Figure 10 was produced for GID 164, but for movement along multiple flowline/transects as well.

• While implementing all of these suggestions may be out of scope for this study, integrating any of them would still be beneficial for the quality of the AutoTerm dataset and the cryosphere community as a whole.

We agree with the reviewer that ensuring all the termini pass through the central flowline is a necessary step of screening. We calculate the number of intersections between each glacier terminus and its flowline and only consider the termini with a single intersection as valid. The related description is added in

> Line 240: "*The outliers are quantified in three different categories: 1) the number of intersections between terminus and glacier flowline, 2) terminus length, 3) terminus curvature, and 4) the abnormally large area enclosed by the two temporally closest termini.*"

> Line 245: "*We will only keep termini that have a single intersection with the glacier flowline.*"

We previously used movements of termini along the central flowline to calculate the time series of the terminus variations and detect the outliers. However, this method can only determine whether there are anomalies at the intersections of the glacier termini and the flowlines. Therefore, we use the area enclosed by the two temporally closest termini. That area can be taken as the first derivative of terminus variations. If the area is too large, it will be detected as an outlier to avoid large jumps in time series. We use the inter-quartile range to automatically detect the outliers, which is similar to Otsu's thresholding method suggested by the reviewer.

3. If possible, it would also helpful to see the actual classifications output by the AutoTerm neural network, and see how it performs against images with issues like sea-ice mélange, ice tongues, clouds, debris, data gaps (from image boundaries or otherwise).

• Ultimately, the AutoTerm methodology must either successfully detect the glacial termini under those conditions, or flag the detections as incorrect, as it will otherwise result in an impact the quality of the final dataset. As is, the dataset is of course still very valuable, but further validation will help to improve ease of use and accessibility.

We didn't save the classifications but only the final shapefiles of the glacier termini, as the classification files consume much storage. The network can have a good performance in some of the difficult situations (Fig. 4), but not always. Figure R1 is the classification output of the results in Figure 4c.

[Figure]

**Figure R1**. The classification output of the result in Figure 4c.

The screening module also helps to detect glacier termini that are of poor quality. However, the screening module is not perfect, and therefore, we also depend on future community feedback about our products to assist in identifying issues not caught by our screening module. We have acknowledged this in Line 485.

4. Termpicks data provides a standard metadata format that would be good to adhere to, and would provide information such as source image IDs, should users seek to manually verify the termini picks. Fields like Error could also be renamed to Uncertainty/Variance that would better describe the measure, to be more in line with the changes in the manuscript. This may be out of scope and is not necessary, but would be valuable if provided.

For the satellites that we use, one satellite only captured one image of a glacier on a certain day. Therefore, we provide the date and the source satellite of each terminus traces for users to track the source satellite images. We agree with the reviewer and changed the field Error to Uncertainty in shapefiles.

**Specific Comments**

Images of the relevant GIDs are provided in the PDF version of this comment.

[Figure]

1. GID53 & 54 - These glaciers seem to overlap. Has GID54 not been properly isolated from GID53's domain? Additionally, there are some apparent image artifacts that have persisted through the termini extraction process (see concertation of termini boundaries along centerline – is there a physical explanation for this besides the image boundaries ending within the domain, and getting missed by the automated quality control algorithm?). Furthermore, the highlighted picks should have been removed given inspection of this output data. Perhaps consider removing termini that go beyond the glacial boundaries or intrude too deeply into the ice mask.

Figure R2 shows the termini for GID53 and GID54. The part in the red box of Fig. R2 might be the overlap part mentioned by the reviewer, which is not considered in our glacier inventory and truncated out from the termini of GID53.

[Figure]

**Figure R2.** Glacier termini of GID53 (pink) and GID54 (Green).

[Figure]

2. GID85 - Note the additional erroneous fronts. This data needs to be validated before release. Large amounts of fronts in the ocean and inland indicate potential issues with sea ice and/or clouds. Similar to the above, remove any termini that intrude too deeply into the ocean mask.

We have refined the glacier domine to truncate the glacier termini. This glacier has a long terminus, which makes it difficult to accurately delineate the full terminus. The uncertainty of this glacier is high, larger than 900 meters.

[Figure]

**Figure R3.** Glacier termini of GID85.

[Figure]

3. GID 44 & 45 - Perhaps implement an additional automatic quality control metric, deviation from closest/mean termini. Any termini which deviates a lot from the average/mean/nearest termini should be flagged for exclusion. Note also that the apparent Error (uncertainty) metadata is low, despite other valid termini having higher Error (uncertainty) values.

Figure R4 shows the updated termini for GID44 and GID45. The uncertainties are low because most of the termini are of good quality.

[Figure]

**Figure R4.** Glacier termini of GID44 (brown) and GID45 (green).

[Figure]

[Figure]

4. GID 86 & GID 144 - Another quality control metric could account for the distance between points in the detected line (to detect large straight jumps), as well as detecting the number/magnitude of large turns in the termini lines. Similar to the above, be sure to eliminate any detections that intrude too far into the grounded ice mask.

Figure R5 shows the termini of GID86. This glacier has a floating tongue, which makes the terminus delineation difficult. When a certain number of results are problematic, the threshold obtained from the interquartile range will be inappropriate. Therefore, there are still some traces that are not captured by the screening, which is reflected in its high uncertainty (826 meters).

[Figure]

**Figure R5.** Glacier termini of GID86.

[Figure]

**Figure R6.** Glacier termini of GID144.

[Figure]

[Figure]

5. GID 120, GID150-153 - While this glacial domain looks fine for the majority of the termini, there are still issues with the domain boundary cutoff and termini that should be removed along those boundaries. Consider checking the proximity of the termini to the domain boundaries, and eliminating them if they have significant portions that are close to the edges & differ from the other termini.

We refine the glacier domains for GID120, and GID150-GID154. GID152, GID153, and GID154 were previously one glacier (Kangerlussuaq). As that glacier retreated, it diverged into three tributaries and exposed new glacier termini (GID153 and GID154).

[Figure]

**Figure R7.** Glacier termini of GID120.

[Figure]

**Figure R8.** Glacier termini of GID150 (yellow) and GID151(red).

[Figure]

**Figure R9.** Glacier termini of GID152 (purple), GID153(pink) and GID154 (green).

[Figure]

6. GID 262, 253, 210, 211, 214, 215, etc.- Overlapping and erroneous/boundary issue termini. While enlarging the domains and re-evaluating the termini may be out of scope, perhaps add commentary on potential solutions to this issue for future work (such as consolidating domains/GIDs and reprocessing, merging termini across GIDs from the existing dataset, or using the overlapping smaller inset/subset domains to cut away overlapping portions of the larger domain's termini).

The overlap issues of these glaciers are because they were previously one glacier. As a glacier retreats, it diverges into several tributaries and exposes new glacier termini. Our previous terminus products include both the old part and the newly exposed part, which causes the overlapping issue. We refined the glacier domains and isolated termini for each glacier.

[Figure]

**Figure R10.** Glacier termini of GID210 (pink) and GID211(orange).

[Figure]

**Figure R11.** Glacier termini of GID214 (blue) and GID215(purple).

[Figure]

**Figure R12.** Glacier termini of GID253 (brown) and GID254(yellow).

[Figure]

**Figure R13.** Glacier termini of GID262.

---

## Author Response (AR3)

**Response to Review 1**

We greatly appreciate the detailed and thorough review by Reviewer 1 throughout the entire review progress.

**Font Color in this response**

The **yellow** color represents the second round of the response. The **black** represents the third round of the reviewer's comments. The **blue** color represents the third round of the response.

1. Thank you. Please add the information that the test set is randomly chosen from TermPicks to the manuscript. Line 225 „*We measure the test error by calculating the averaged width of the enclosed area bounded by the TermPicks traces and the network predictions*" – How is width defined here? What happens when the prediction crosses the trace? Will that negate the error? I'm still not sure how exactly the error is calculated. A formula or figure would be helpful.

   The information that the test set is randomly chosen is described in Line 221: "*From TermPicks traces, we randomly select 100 traces as the test set and take the rest into the training set.*"

   The averaged width means the enclosed area bounded by the TermPicks traces and the network predictions divided by the length of the TermPicks traces. If there is a cross, we calculated the area on both sides of the cross and add them up.

   We rephrase the sentence in Line 225 as: "*We measure the test error by calculating the enclosed area bounded by the TermPicks traces and the network predictions divided by the length of the TermPicks traces. In the case where there are crosses between a* TermPicks trace *and a network prediction, we calculate the area on both sides of the crosses and then add them together.*" We choose not to an equation in the manuscript as the revised text explains itself.

2. I think it is unclear what is meant by „the deep learning network" (Line 404 and same in line 79) – I was assuming you meant previous deep learning models. From what I understand now, you mean your own model, which was improved by incorporating TermPicks into the train set. Please rephrase these sections, as like this it sounds like you would improve over previous models (which is not proven – actually, you just have indices that it performs on par).

   We agree with the reviewer and changed "network" to "model" throughout the manuscript.

3. Additionally, please state in section lines 326-328 that the test sets are different and that the test errors are calculated slightly differently (please see this quote from Cheng et al. 2021: „The primary quality assessment method is the mean distance error (Mohajerani et al., 2019; Zhang et al., 2019; Baumhoer et al., 2019). Conceptually, this method resembles the numerical integration of the area between two curves, normalized by the average length of the curves (see Fig. 8a). Also referred to as the area over front (A/F) in literature, this method can also be seen as a generalization of the method of transects along arbitrarily oriented fronts (Mohajerani et al., 2019; Baumhoer et al., 2019). This metric is implemented by taking the mean–median of the distances between closest pixels in the predicted and manually delineated fronts.").

   We have rephrased the sentence as: "*Our averaged test error is 79 meters (Table S1), which is comparable to previous studies where errors range from 33 to 108 m (Mohajerani et al., 2019;*

*Zhang et al., 2019; Baumhoer et al., 2019; Cheng et al., 2020), although the test set and the way of calculating test error are slightly different".*

4. Actually, the confusion does not come solely from the method section. For example, section 2 – „input data" - input to what? The pipeline? I was under the impression that the model would just get remote sensing imagery as input and not additionally an ice/ocean mask?

   It is the input data of the pipeline. The model would just need remote sensing imagery to produce glacier termini. We need a reference ice/ocean mask to update the mask using the newly produced termini. The name of the section 2 is now changed to "Input Data of the Pipeline".

5. We add one sentence in Line 453: "Another limitation is that even though we include additional training data, the network might struggle with some challenging situations (Fig. S2)."

   Should this not be investigated to prove the correctness of your produced dataset? Please check and report it in the manuscript.

   We have calculated the uncertainty of the terminus traces, which reflect such a limitation. Our screening module can detect some of the wrong picks caused by this limitation. We also mentioned in Line 487 that: "*We depend on future community feedback about our products to assist in identifying issues not caught by our screening module. This is because the massive amount of data precludes the ability to guarantee the quality of each individual trace.*"

6. We use the cubic upsampling method.

   Please add this information to the manuscript.

   We rephrase the sentence as: "*Specifically, we upsample small images (image width less than 1000 pixels) by an integer value using cubic interpolation so that their widths are just over 1000 pixels.*"

7. This is Table S3 now. Please correct.

   Corrected in the manuscript.

8. Network „architecture" is actually a technical term. Hence, please do not call it „structure". Moreover, please also add the reference (Chen et al., 2018b) after this sentence directly, as the large learning capability has been shown by it and not Zhang et al. (2021).

   We agree with the reviewer and changed "network structure" to "network architecture" throughout the manuscript and add the reference (Chen et al., 2018) right after the sentence: "*This network architecture has been proven to have large learning capability (Chen et al., 2018), spatial transferability, and the capability of using multi-sensor remote sensing images (Zhang et al., 2021).*"

9. The question was if the screening can only be done on time series or whether it is possible also when I just want one single trace from one satellite image.

   The screening will not be able to work effectively if there is only one trace as the screening method is essentially an outlier-detection method. However, this limitation is not a concern in the context of glacier termini extraction because we will typically have a wealth of remote sensing images as well as terminus data for screening purposes.

10. We add one sentence in Line 413: "Despite the success of the screening module in Greenland, further validation will be needed as applying it globally."

Thank you, however please add this in the limitations section.

We have moved this sentence to the limitation section, Line 454: "*The fourth limitation is that further validation will be needed as applying it globally despite the success of the screening module in Greenland.*"

11. The duplicate uncertainty is calculated from the Landsat-8 and Sentinel-2, but we use the value to represent the network's uncertainty on a certainty glacier. Therefore, such an uncertainty is more representative of Landsat-8 and Sentinel-2, and would be biased towards a lower value when representing the uncertainty of results obtained from other satellites. To clarify, we added on sentence in Line 465: "Since Landsat-8 and Sentinel-2 images have the highest resolution among the five satellites, using the duplicate uncertainty to represent the error of results obtained from other satellites would be biased towards lower values."

This must also be part of the limitations section, not only the „Difference of the two types of uncertainties" section.

We added on sentence in Line 455: "*The fifth limitation comes from the biased value of duplicate uncertainty as Landsat-8 and Sentinel-2 images have the highest resolution among the five satellites.*"

12. We randomly choose ten images from each of the five sensors and make three inferences for each of the ten images. The related description can be found in Line 298. We combine For loop and "shuf -n 10" in Bash to automatically and randomly choose ten images for calculating MC dropout uncertainty.

Sorry, I think I did not make myself clear. Let me rephrase: When I apply your pipeline to a new glacier in Antarctica for one date and one sensor, will it calculate the MC dropout uncertainty for it?

Yes, it will.

13. Your understanding is mostly correct. It's just that when all three variables are uniformly distributed, even if we have a lower T_L and higher T_U, we will filter out more predictions than normal cases as the results will have a concentrated distribution in the normal cases. I'm sorry, I do not understand what you mean by normal cases here and what your difference between results and predictions is. Which concentrated distribution? Please explain, as my argument still stands, and I think this is important for both your dataset and future use of the pipeline.

In the normal case, the distribution will have a Gaussian-like distribution (concentrated distribution, Figure R1). And you are correct, when variables (length, smoothness, and enclosed area) are evenly distributed, the screening module will fail to detect the wrong picks. Sorry that we made a mistake in the previous response.

The limitation of the screening module is described in Line 458 : "*The second limitation is caused by our assumption that the screening module provides high-quality results. This assumption rests on the choice of thresholds defined by the interquartile range in the screening module. Thus, when*

*most results for a glacier are not credible, the screening module might not be able to clean the results because the random distribution of the terminus attributes leads to improper thresholds.*"

However, the low success rate can still be used to indicate poorly performing glaciers, although it may not serve as a warning for all of them. To compensate for this limitation, we will use both uncertainty and success rate to indicate poorly performing glaciers. We rephrased the sentence in Line 480 as: "*The network's failure will result in many termini not passing the screening and high uncertainty. The pipeline can use the low success rates and high uncertainty to alert us to prepare more training data for the corresponding glaciers.*"

[Figure]

Figure R1. One example of the terminus length distribution in the normal case. The two bars indicate the lower T_L and higher T_U, respectively.

14. Please revise all figures, as there are still several that show green and red in one plot.

    We have changed the green color to dark green and checked the figure through https://www.color-blindness.com/coblis- color-blindnesssimulator/.

15. We thank the reviewer for the comment and added one sentence in Line 465: "Since Landsat-8 and Sentinel-2 images have the highest resolution among the five satellites, using the duplicate uncertainty to represent the error of results obtained from other satellites would be biased towards lower values."

Sorry, I should have been clearer. This must also be part of the limitations section, not only the „Difference of the two types of uncertainties" section.

*We added on sentence in Line 455: "The fifth limitation comes from the biased value of duplicate uncertainty as Landsat-8 and Sentinel-2 images have the highest resolution among the five satellites."*

**Response to Review 2**

**General Comments:**

The authors' response addresses the questions and comments raised by all reviewers, and integrates the feedback into the revised manuscript.

Specific revisions include detailed responses and technical corrections to issues in the dataset, and integration of suggested quality control measures into the methodology. The authors acceptably address reviewer concerns, and the added text is free from syntactical errors.

After review of the author's responses, as well as the changes to the revised manuscript & associated dataset, I can recommend that this submission should be accepted, at the editor's discretion.

*We greatly appreciate the careful inspection of the data and the constructive suggestions by Reviewer 2 throughout the entire review progress.*